# Learning multi-modal generative models with permutation-invariant encoders and tighter variational objectives

**Marcel Hirt**                                                                 *marcelandre.hirt@ntu.edu.sg*
*School of Social Sciences*
*Nanyang Technological University*
*Singapore*

**Domenico Campolo**                                                              *d.campolo@ntu.edu.sg*
*School of Mechanical and Aerospace Engineering*
*Nanyang Technological University*
*Singapore*

**Victoria Leong**                                                          *VictoriaLeong@ntu.edu.sg*
*School of Social Sciences*
*Nanyang Technological University*
*Singapore*

**Juan-Pablo Ortega**                                                    *juan-pablo.ortega@ntu.edu.sg*
*School of Physical and Mathematical Sciences*
*Nanyang Technological University*
*Singapore*

**Reviewed on OpenReview:** *https://openreview.net/forum?id=lM4nHnxGfL*

## Abstract

Devising deep latent variable models for multi-modal data has been a long-standing theme in machine learning research. Multi-modal Variational Autoencoders (VAEs) have been a popular generative model class that learns latent representations that jointly explain multiple modalities. Various objective functions for such models have been suggested, often motivated as lower bounds on the multi-modal data log-likelihood or from information-theoretic considerations. To encode latent variables from different modality subsets, Product-of-Experts (PoE) or Mixture-of-Experts (MoE) aggregation schemes have been routinely used and shown to yield different trade-offs, for instance, regarding their generative quality or consistency across multiple modalities. In this work, we consider a variational objective that can tightly approximate the data log-likelihood. We develop more flexible aggregation schemes that avoid the inductive biases in PoE or MoE approaches by combining encoded features from different modalities based on permutation-invariant neural networks. Our numerical experiments illustrate trade-offs for multi-modal variational objectives and various aggregation schemes. We show that our variational objective and more flexible aggregation models can become beneficial when one wants to approximate the true joint distribution over observed modalities and latent variables in identifiable models.

## 1  Introduction

Multi-modal data sets where each sample has features from distinct sources have grown in recent years. For example, multi-omics data such as genomics, epigenomics, transcriptomics, and metabolomics can provide a

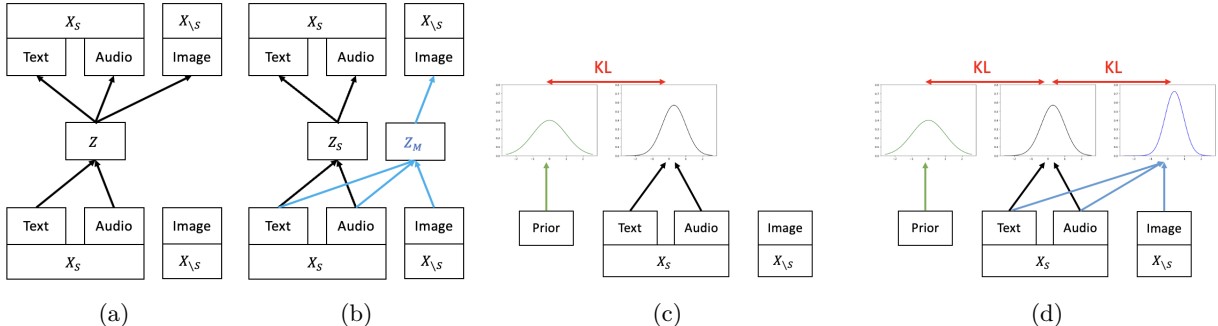

(a)        (b)        (c)        (d)

Figure 1: Reconstruction or cross-prediction of modalities in 1a and 1b for a mixture-based bound and our objective, respectively. The mixture-based bound resorts to a single latent variable $Z \sim q_\phi(\cdot|x_\mathcal{S})$ that encodes information from a modality subset $x_\mathcal{S}$ and is trained to reconstruct the conditioning modalities $x_\mathcal{S}$, as well as to predict the masked modalities $x_{\backslash\mathcal{S}}$. Our objective relies on two latent variables $Z_\mathcal{S} \sim q_\phi(\cdot|x_\mathcal{S})$ and $Z_\mathcal{M} \sim q_\phi(\cdot|x_\mathcal{S}, x_{\backslash\mathcal{S}})$, where $Z_\mathcal{S}$ is learned to reconstruct all its conditioning modalities, with $Z_\mathcal{M}$ learned to reconstruct the remaining modalities. KL regularization terms in 1c and 1d for a mixture-based bound and our objective, respectively. The mixture-based bound aims to minimize the KL divergence between the encoding distribution given a modality subset $x_\mathcal{S}$ and a prior distribution. Our objective additionally aims to minimize the KL divergence between the encoding distribution given all modalities relative to the encoding distribution of a modality subset $x_\mathcal{S}$.

more comprehensive understanding of biological systems if multiple modalities are analyzed in an integrative framework (Argelaguet et al., 2018; Lee and van der Schaar, 2021; Minoura et al., 2021). In neuroscience, multi-modal integration of neural activity and behavioral data can help to learn latent neural dynamics (Zhou and Wei, 2020; Schneider et al., 2023). However, annotations or labels in such data sets are often rare, making unsupervised or semi-supervised generative approaches particularly attractive as such methods can be used in these settings to (i) generate data, such as missing modalities, and (ii) learn latent representations that are useful for down-stream analyzes or that are of scientific interest themselves. The availability of heterogeneous data for different modalities promises to learn generalizable representations that can capture shared content across multiple modalities in addition to modality-specific information. A promising class of weakly-supervised generative models is multi-modal VAEs (Suzuki et al., 2016; Wu and Goodman, 2019; Shi et al., 2019; Sutter et al., 2021) that combine information across modalities in an often-shared low-dimensional latent representation. A common route for learning the parameters of latent variable models is via maximization of the marginal data likelihood with various lower bounds thereof, as suggested in previous work.

**Setup.** We consider a set of $M$ random variables $\{X_1, \ldots, X_M\}$ with empirical density $p_d$, where each random variable $X_s$, $s \in \mathcal{M} = \{1, \ldots, M\}$, can be used to model a different data modality taking values in $\mathsf{X}_s$. With some abuse of notation, we write $X = \{X_1, \ldots, X_M\}$ and for any subset $\mathcal{S} \subset \mathcal{M}$, we set $X = (X_\mathcal{S}, X_{\backslash\mathcal{S}})$ for two partitions of the random variables into $X_\mathcal{S} = \{X_s\}_{s\in\mathcal{S}}$ and $X_{\backslash\mathcal{S}} = \{X_s\}_{s\in\mathcal{M}\backslash\mathcal{S}}$. We pursue a latent variable model setup, analogous to uni-modal VAEs (Kingma and Ba, 2014; Rezende et al., 2014). For a latent variable $Z \in \mathsf{Z}$ with prior density $p_\theta(z)$, we posit a joint generative model[1] $p_\theta(z, x) = p_\theta(z) \prod_{s=1}^M p_\theta(x_s|z)$, where $p_\theta(x_s|z)$ is commonly referred to as the decoding distribution for modality $s$. Observe that all modalities are independent given the latent variable $z$ shared across all modalities. However, one can introduce modality-specific latent variables by making sparsity assumptions for the decoding distribution. Intuitively, this conditional independence assumption means that the latent variable $Z$ captures all unobserved factors shared by the modalities.

---

[1]We usually denote random variables using upper-case letters, and their realizations by the corresponding lower-case letter. We assume throughout that $\mathsf{Z} = \mathbb{R}^D$, and that $p_\theta(z)$ is a Lebesgue density, although the results can be extended to more general settings such as discrete random variables $Z$ with appropriate adjustments, for instance, regarding the gradient estimators.

**Multi-modal variational bounds and mutual information.** Popular approaches to train multi-modal models are based on a mixture-based variational bound (Daunhawer et al., 2022; Shi et al., 2019) given by $\mathcal{L}^{\mathrm{Mix}}(\theta, \phi, \beta) = \int \rho(S) \mathcal{L}_{\mathcal{S}}^{\mathrm{Mix}}(x, \theta, \phi, \beta) \mathrm{d}\mathcal{S}$, where

$$\mathcal{L}_{\mathcal{S}}^{\mathrm{Mix}}(x, \theta, \phi, \beta) = \int q_\phi(z|x_{\mathcal{S}}) \left[\log p_\theta(x|z)\right] \mathrm{d}z - \beta \mathsf{KL}(q_\phi(z|x_{\mathcal{S}})|p_\theta(z)) \tag{1}$$

and $\rho$ is some distribution on the power set $\mathcal{P}(\mathcal{M})$ of $\mathcal{M}$ and $\beta > 0$. For $\beta = 1$, one obtains the bound $\mathcal{L}_{\mathcal{S}}^{\mathrm{Mix}}(x, \theta, \phi, \beta) \leq \log p_\theta(x)$. However, as shown in Daunhawer et al. (2022), there is a gap between the variational bound and the log-likelihood given by the conditional entropies that cannot be reduced even for flexible encoding distributions. More precisely, it holds that

$$\int p_d(x) \log p_\theta(x) \mathrm{d}x \geq \int p_d(x) \mathcal{L}^{\mathrm{Mix}}(x, \theta, \phi, 1) \mathrm{d}x + \int \rho(\mathcal{S}) \mathcal{H}(p_d(X_{\setminus \mathcal{S}}|X_{\mathcal{S}})) \mathrm{d}\mathcal{S},$$

where $\mathcal{H}(p_d(X_{\setminus \mathcal{S}}|X_{\mathcal{S}}))$ is the entropy of the conditional data distributions. Intuitively, in (1), one tries to reconstruct or predict all modalities from incomplete information using only the modalities $\mathcal{S}$, which leads to learning an inexact, average prediction (Daunhawer et al., 2022). In particular, it cannot reliably predict modality-specific information that is not shared with other modality subsets, as measured by the conditional entropies $\mathcal{H}(p_d(X_{\setminus \mathcal{S}}|X_{\mathcal{S}}))$. For an illustration, see Figure 1a, where the latent variable $Z$ encodes information from a text and audio modality and is tasked to both reconstruct the text and audio modalities and to predict an unobserved image. Information that is specific to the image modality only thus cannot be recovered. A related observation has been made for Masked AutoEncoders (He et al., 2022) that correspond to the limiting case $\beta \to 0$, where the cross-reconstruction objective leads to learning features that are invariant to the masking of modalities (Kong and Zhang, 2023) and allows for the recovery of latent variables that represent maximally shared information between the unmasked and masked modality (Kong et al., 2023).

We will illustrate that maximizing $\mathcal{L}_{\mathcal{S}}^{\mathrm{Mix}}$ can be interpreted as the information-theoretic objective of

$$\text{maximizing } \left\{ \hat{\mathrm{I}}_{q_\phi}^{\mathrm{lb}}(X, Z_{\mathcal{S}}) - \beta \hat{\mathrm{I}}_{q_\phi}^{\mathrm{ub}}(X_{\mathcal{S}}, Z_{\mathcal{S}}) \right\}, \tag{2}$$

where $\hat{\mathrm{I}}_q^{\mathrm{ub}}$ and $\hat{\mathrm{I}}_q^{\mathrm{lb}}$ are variational upper, respectively, lower bounds of the corresponding mutual information $I_q(X, Y) = \int q(x, y) \log \frac{q(x,y)}{q(x)q(y)} \mathrm{d}x \mathrm{d}y$ of random variables $X$ and $Y$ having marginal and joint densities $q$. In order to emphasize that the latent variable $Z$ is conditional on $X_{\mathcal{S}}$ under the encoding density $q_\phi$, we write $Z_{\mathcal{S}}$ instead of $Z$. Variations of (1) have been suggested (Sutter et al., 2020), such as by replacing the prior density $p_\theta$ in the KL-term by a weighted product of the prior density $p_\theta$ and the uni-modal encoding distributions $q_\phi(z|x_s)$, for all $s \in \mathcal{M}$. Likewise, the multi-view variational information bottleneck approach developed in Lee and van der Schaar (2021) for predicting $X_{\setminus \mathcal{S}}$ given $X_{\mathcal{S}}$ can be interpreted as maximizing $\hat{\mathrm{I}}_{q_\phi}^{\mathrm{lb}}(X_{\setminus \mathcal{S}}, Z_{\mathcal{S}}) - \beta \hat{\mathrm{I}}_{q_\phi}^{\mathrm{ub}}(X_{\mathcal{S}}, Z_{\mathcal{S}})$. Hwang et al. (2021) suggested a related bound that aims to maximize the reduction of total correlation of $X$ when conditioned on a latent variable. Similar bounds have been suggested in Sutter et al. (2020) and Suzuki et al. (2016) by considering different KL-regularisation terms; see also Suzuki and Matsuo (2022). Shi et al. (2020) add a contrastive estimate $\hat{\mathrm{I}}_{p_\theta}$ of the point-wise mutual information to the maximum likelihood objective and minimize $-\log p_\theta(x) - \beta \hat{\mathrm{I}}_{p_\theta}(x_{\mathcal{S}}, x_{\setminus \mathcal{S}})$. Optimizing variational bounds of different mutual information terms such as (2) yield latent representations that have different trade-offs in terms of either (i) reconstruction or (ii) cross-prediction of multi-modal data from a rate-distortion viewpoint (Alemi et al., 2018).

**Multi-modal aggregation schemes.** To optimize the variational bounds above or to allow for flexible conditioning at test time, we need to learn encoding distributions $q_\phi(z|x_{\mathcal{S}})$ for any $\mathcal{S} \in \mathcal{P}(\mathcal{M})$. The typical aggregation schemes that are scalable to a large number of modalities are based on a choice of uni-modal encoding distributions $q_{\phi_s}(z|x_s)$ for any $s \in \mathcal{M}$, which are then used to define the multi-modal encoding distributions as follows:

- Mixture of Experts (MoE), see Shi et al. (2019),

$$q_\phi^{\mathrm{MoE}}(z|x_{\mathcal{S}}) = \frac{1}{|\mathcal{S}|} \sum_{s \in \mathcal{S}} q_{\phi_s}(z|x_s).$$

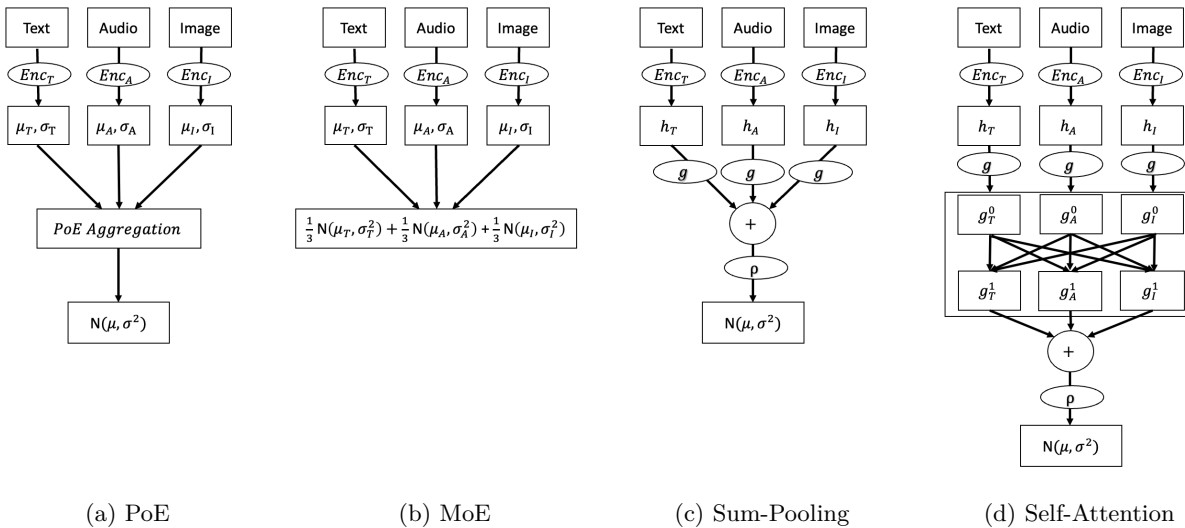

(a) PoE      (b) MoE      (c) Sum-Pooling      (d) Self-Attention

Figure 2: Illustration of multi-modal aggregation schemes. All encoding schemes first apply modality-specific encoders to each individual modality. A PoE model (2a) aggregates the outputs from the modality-specific encoders into a single Gaussian distribution that results from a multiplication of the corresponding uni-modal Gaussian densities. An MoE model (2b) assumes an equally weighted Gaussian mixture distribution comprised of the uni-modal Gaussian densities. Our new aggregation schemes allow for learning permutation-invariant fusion models: A Sum-Pooling or Deep Set model (2c) applies the same function $g$ to the encoded features $h_s$, $s \in \{T, A, I\}$, before summing them up and using a non-linear projection $\rho$ to the parameters of a Gaussian distribution. A Self-Attention model (2d) differs from the Sum-Pooling approach by applying self-attention layers or transformer layers before summing up the features, thereby accounting for pairwise interactions between the encoded modalities. Our newly introduced schemes allow for encoding only a modality subset by using standard masking operations.

- Product of Experts (PoE), see Wu and Goodman (2018),

$$q_\phi^{\mathrm{PoE}}(z|x_\mathcal{S}) \propto p_\theta(z) \prod_{s \in \mathcal{S}} q_{\phi_s}(z|x_s).$$

Figure 2a-2b illustrates these previously considered aggregation schemes. While these schemes do not require learning the aggregation function, we introduce aggregation schemes that, as illustrated in 2c-2d, involve learning additional neural network parameters that specify a learnable aggregation function.

**Contributions.** This paper contributes (i) a new variational objective as an approximation of a lower bound on the multi-modal log-likelihood (LLH). We avoid a limitation of mixture-based bounds (1), which may not provide tight lower bounds on the joint LLH if there is considerable modality-specific variation (Daunhawer et al., 2022), even for flexible encoding distributions. The novel variational objective contains a lower bound of the marginal LLH $\log p_\theta(x_\mathcal{S})$ and a term approximating the conditional $\log p_\theta(x_{\backslash \mathcal{S}}|x_\mathcal{S})$ for any choice of $\mathcal{S} \in \mathcal{P}(\mathcal{M})$, provided that we can learn a flexible multi-modal encoding distribution. This paper then contributes (ii) new multi-modal aggregation schemes that yield more expressive multi-modal encoding distributions when compared to MoEs or PoEs. These schemes are motivated by the flexibility of permutation-invariant (PI) architectures such as DeepSets (Zaheer et al., 2017) or attention models (Vaswani et al., 2017; Lee et al., 2019). We illustrate that these innovations (iii) are beneficial when learning identifiable models, aided by using flexible prior and encoding distributions consisting of mixtures, and (iv) yield higher LLH in experiments.

**Further related work.** Canonical Correlation Analysis (Hotelling, 1936; Bach and Jordan, 2005) is a classical approach for multi-modal data that aims to find projections of two modalities by maximally correlating

and has been extended to include more than two modalities (Archambeau and Bach, 2008; Tenenhaus and Tenenhaus, 2011) or to allow for non-linear transformations (Akaho, 2001; Hardoon et al., 2004; Wang et al., 2015; Karami and Schuurmans, 2021). Probabilistic CCA can also be seen as multi-battery factor analysis (MBFA) (Browne, 1980; Klami et al., 2013), wherein a shared latent variable models the variation common to all modalities with modality-specific latent variables capturing the remaining variation. Likewise, latent factor regression or classification models (Stock and Watson, 2002) assume that observed features and response are driven jointly by a latent variable. Vedantam et al. (2018) considered a tiple-ELBO for two modalities, while Sutter et al. (2021) introduced a generalized variational bound that involves a summation over all modality subsets. A series of work has developed multi-modal VAEs based on shared and private latent variables (Wang et al., 2016; Lee and Pavlovic, 2021; Lyu and Fu, 2022; Lyu et al., 2021; Vasco et al., 2022; Palumbo et al., 2023). Tsai et al. (2019a) proposed a hybrid generative-discriminative objective and minimized an approximation of the Wasserstein distance between the generated and observed multi-modal data. Joy et al. (2021) consider a semi-supervised setup of two modalities that requires no explicit multi-modal aggregation function. Extending the Info-Max principle (Linsker, 1988), maximizing mutual information $I_q(g_1(X_1), g(X_2)) \le I_q((X_1, X_2), (Z_1, Z_2))$ based on representations $Z_s = g_s(X_s)$ for modality-specific encoders $g_s$ from two modalities has been a motivation for approaches based on (symmetrized) contrastive objectives (Tian et al., 2020; Zhang et al., 2022c; Daunhawer et al., 2023) such as InfoNCE (Oord et al., 2018; Poole et al., 2019; Wang and Isola, 2020) as a variational lower bound on the mutual information between $Z_1$ and $Z_2$. Recent work (Bounoua et al., 2023; Bao et al., 2023) considered score-based diffusion models on auto-encoded private latent variables.

## 2 A tighter variational objective with arbitrary modality masking

For $\mathcal{S} \subset \mathcal{M}$ and $\beta > 0$, we define

$$\mathcal{L}_\mathcal{S}(x_\mathcal{S}, \theta, \phi, \beta) = \int q_\phi(z|x_\mathcal{S}) \left[\log p_\theta(x_\mathcal{S}|z)\right] \mathrm{d}z - \beta \mathsf{KL}(q_\phi(z|x_\mathcal{S})|p_\theta(z)). \tag{3}$$

This is simply a standard variational lower bound (Jordan et al., 1999; Blei et al., 2017) restricted to the subset $\mathcal{S}$ for $\beta = 1$, and therefore $\mathcal{L}_\mathcal{S}(x_\mathcal{S}, \theta, \phi, 1) \le \log p_\theta(x_\mathcal{S})$. One can express the variational bound in information-theoretic (Alemi et al., 2018) terms as

$$\int p_d(x_\mathcal{S})\mathcal{L}(x_\mathcal{S})\mathrm{d}x_\mathcal{S} = -D_\mathcal{S} - \beta R_\mathcal{S}$$

for the rate

$$R_\mathcal{S} = \int p_d(x_\mathcal{S})\mathsf{KL}(q_\phi(z|x_\mathcal{S})|p_\theta(z))\mathrm{d}x_\mathcal{S}$$

measuring the information content that is encoded from the observed modalities in $\mathcal{S}$ by $q_\phi$ into the latent representation, and the distortion

$$D_\mathcal{S} = -\int p_d(x_\mathcal{S})q_\phi(z|x_\mathcal{S})\log p_\theta(x_\mathcal{S}|z)\mathrm{d}z\mathrm{d}x_\mathcal{S}$$

given as the negative reconstruction log-likelihood of the modalities in $\mathcal{S}$. While the latent variable $Z_\mathcal{S}$ that is encoded via $q_\phi(z|x_\mathcal{S})$ from $X_\mathcal{S}$ can be tuned via the choice of $\beta > 0$ to tradeoff compression and reconstruction of all modalities in $\mathcal{S}$ jointly, it does not explicitly optimize for cross-modal prediction of modalities not in $\mathcal{S}$. Indeed, the mixture-based variational bound differs from the above decomposition exactly by an additional cross-modal prediction or cross-distortion term

$$D^c_{\backslash \mathcal{S}} = -\int p_d(x)q_\phi(z_\mathcal{S}|x_\mathcal{S})\log p_\theta(x_{\backslash \mathcal{S}}|z_\mathcal{S})\mathrm{d}z_\mathcal{S}\mathrm{d}x,$$

thereby explicitly optimizing for both self-reconstruction of a modality subset and cross-modal prediction within a single objective:

$$\int p_d(\mathrm{d}x)\mathcal{L}^{\mathrm{Mix}}_\mathcal{S}(x) = -D_\mathcal{S} - D^c_{\backslash \mathcal{S}} - \beta R_\mathcal{S}. \tag{4}$$

Instead of adding an explicit cross-modal prediction term, we consider an additional variational objective with a second latent variable $Z_{\mathcal{M}}$ that encodes all observed modalities $X = X_{\mathcal{M}}$ and tries to reconstruct the remaining modality subset. Unlike the latent variable $Z_{\mathcal{S}}$ in (1) and (3) that can only encode incomplete information using the modalities $\mathcal{S}$, this second latent variable $Z_{\mathcal{M}}$ can encode modality-specific information from all observed modalities, thereby avoiding the averaging prediction in the mixture-based bound, see Figure 1b.

Ideally, we may like to consider an additional variational objective that lower bounds the conditional log-likelihood $\log p_\theta(x_{\backslash \mathcal{S}}|x_{\mathcal{S}})$ so that maximizing the sum of both bounds maximizes a lower bound of the multi-modal log-likelihood $\log p_\theta(x_{\mathcal{S}}, x_{\backslash \mathcal{S}}) = \log p_\theta(x_{\mathcal{S}}) + \log p_\theta(x_{\backslash \mathcal{S}}|x_{\mathcal{S}})$. To motivate such a conditional variational objective, note that maximizing the variational bound (3) with infinite capacity encoders yields $q_\phi(z|x_{\mathcal{S}}) = p_\theta(z|x_{\mathcal{S}})$. This suggests replacing the intractable posterior $p_\theta(z|x_{\mathcal{S}})$ with the encoder $q_\phi(z|x_{\mathcal{S}})$ for the probabilistic model when conditioned on $x_{\mathcal{S}}$. A variational objective under this replacement then becomes

$$\mathcal{L}_{\backslash \mathcal{S}}(x, \theta, \phi, \beta) = \int q_\phi(z|x) \left[ \log p_\theta(x_{\backslash \mathcal{S}}|z) \right] \mathrm{d}z - \beta \mathsf{KL}(q_\phi(z|x)|q_\phi(z|x_{\mathcal{S}})). \tag{5}$$

However, the above only approximates a lower bound on the conditional log-likelihood $\log p_\theta(x_{\backslash \mathcal{S}}|x_{\mathcal{S}})$, and $\mathcal{L}_{\backslash \mathcal{S}}$ is a lower bound only under idealized conditions. We will make these approximations more precise in Section 2.1, where we illustrate how these bounds yield to a matching of different distributions in the latent or data space, while Section 2.2 provides an information-theoretic interpretation of these variational objectives. The introduction of a second latent variable leads to two KL regularization terms, see Figure 1d, that do not satisfy a triangle inequality $\mathsf{KL}(q_\phi(z|x)|q_\phi(z|x_{\mathcal{S}})) + \mathsf{KL}(q_\phi(z|x_{\mathcal{S}})|p_\theta(z)) \not\geq \mathsf{KL}(q_\phi(z|x)|p_\theta(z))$.

In summary, for some fixed density $\rho$ on $\mathcal{P}(\mathcal{M})$, we suggest to maximize the overall bound

$$\mathcal{L}(x, \theta, \phi, \beta) = \int \rho(\mathcal{S}) \left[ \mathcal{L}_{\mathcal{S}}(x_{\mathcal{S}}, \theta, \phi, \beta) + \mathcal{L}_{\backslash \mathcal{S}}(x, \theta, \phi, \beta) \right] \mathrm{d}\mathcal{S},$$

with respect to $\theta$ and $\phi$, which is a generalization of the bound suggested in Wu and Goodman (2019) to an arbitrary number of modalities. This bound can be optimized using standard Monte Carlo techniques, for example, by computing unbiased pathwise gradients (Kingma and Ba, 2014; Rezende et al., 2014; Titsias and Lázaro-Gredilla, 2014) using the reparameterization trick. For variational families such as Gaussian mixtures[2], one can employ implicit reparameterization (Figurnov et al., 2018). It is straightforward to adapt variance reduction techniques such as ignoring the scoring term of the multi-modal encoding densities for pathwise gradients (Roeder et al., 2017), see Algorithm 1 in Appendix H for pseudo-code. Nevertheless, a scalable approach requires an encoding technique that allows to condition on any masked modalities with a computational complexity that does not increase exponentially in $M$. We will analyse scalable model architectures in Section 3.

## 2.1   Multi-modal distribution matching

Likelihood-based learning approaches aim to match the model distribution $p_\theta(x)$ to the true data distribution $p_d(x)$. Variational approaches achieve this by matching in the latent space the encoding distribution to the true posterior as well as maximizing a tight lower bound on $\log p_\theta(x)$, see Rosca et al. (2018). These types of analyses have proved useful for uni-modal VAEs as they can provide some insights as to why VAEs may lead to worse generative sample quality compared to other generative models such as GANs (Goodfellow et al., 2014) or may fail to learn useful latent representations (Zhao et al., 2019; Dieng et al., 2019). We show similar results for the multi-modal variational objectives. This suggests that limitations from uni-modal VAEs also affect multi-modal VAEs, but also that previous attempts to address these shortcomings in uni-modal VAEs may benefit multi-modal VAEs. In particular, mismatches between the prior and the aggregated prior for uni-modal VAEs that result in poor unconditional generation have a natural counterpart for cross-modal generations with multi-modal VAEs that may potentially be reduced using more flexible conditional prior distributions, see Remark 3, or via adding additional mutual information regularising terms (Zhao et al.,

---

[2]For MoE aggregation schemes, Shi et al. (2019) considered a stratified ELBO estimator as well as a tighter bound based on importance sampling, see also Morningstar et al. (2021), that we do not pursue here for consistency with other aggregation schemes that can likewise be optimized based on importance sampling ideas.

2019), see Remark 4. Given these results, it is neither surprising that multi-modal diffusion models such as Bounoua et al. (2023); Bao et al. (2023) yield improved sample quality, nor that sample quality can be improved by augmenting multi-modal VAEs with diffusion models (Pandey et al., 2022; Palumbo et al., 2024).

We consider the densities

$$p_\theta(z, x) = p_\theta(z)p_\theta(x_\mathcal{S}|z)p_\theta(x_{\backslash\mathcal{S}}|z)$$

and

$$q_\phi(z_\mathcal{S}, x) = p_d(x_\mathcal{S})q_\phi(z_\mathcal{S}|x_\mathcal{S}).$$

The latter is the encoding path comprising the encoding density $q_\phi$ conditioned on $x_\mathcal{S}$ and the empirical density $p_d$. We write

$$q_{\phi,\mathcal{S}}^{\mathrm{agg}}(z) = \int p_d(x_\mathcal{S})q_\phi(z|x_\mathcal{S})\mathrm{d}x_\mathcal{S}$$

for the aggregated prior (Makhzani et al., 2016; Hoffman and Johnson, 2016; Tomczak and Welling, 2017) restricted on modalities from $\mathcal{S}$ and $q^\star(x_\mathcal{S}|z) = q_\phi(x_\mathcal{S}, z)/q_\phi^{\mathrm{agg}}(z)$ and likewise consider its conditional version,

$$q_{\phi,\backslash\mathcal{S}}^{\mathrm{agg}}(z|x_\mathcal{S}) = \int p_d(x_{\backslash\mathcal{S}}|x_\mathcal{S})q_\phi(z|x)\mathrm{d}x_{\backslash\mathcal{S}}$$

for an aggregated encoder conditioned on $x_\mathcal{S}$. We provide a multi-modal ELBO surgery, summarized in Proposition 1 below. It implies that maximizing $\int p_d(x_\mathcal{S})\mathcal{L}_\mathcal{S}(x_\mathcal{S}, \theta, \phi)\mathrm{d}x_\mathcal{S}$ drives

1. the joint inference distribution $q_\phi(z, x_\mathcal{S}) = p_d(x_\mathcal{S})q_\phi(z|x_\mathcal{S})$ of the $\mathcal{S}$ submodalities to the joint generative distribution $p_\theta(z, x_\mathcal{S}) = p_\theta(z)p_\theta(x_\mathcal{S}|z)$ and
2. the generative marginal $p_\theta(x_\mathcal{S})$ to its empirical counterpart $p_d(x_\mathcal{S})$.

Analogously, maximizing $\int p_d(x_{\backslash\mathcal{S}}|x_\mathcal{S})\mathcal{L}_{\backslash\mathcal{S}}(x, \theta, \phi)\mathrm{d}x_{\backslash\mathcal{S}}$ drives, for fixed $x_\mathcal{S}$,

1. the distribution $p_d(x_{\backslash\mathcal{S}}|x_\mathcal{S})q_\phi(z|x)$ to the distribution $p_\theta(x_{\backslash\mathcal{S}}|z)q_\phi(z|x_\mathcal{S})$ and
2. the conditional $p_\theta(x_{\backslash\mathcal{S}}|x_\mathcal{S})$ to its empirical counterpart $p_d(x_{\backslash\mathcal{S}}|x_\mathcal{S})$.

Furthermore, it shows that maximizing $\mathcal{L}_{\backslash\mathcal{S}}(x, \theta, \phi)$ minimizes a Bayes-consistency matching term $\mathsf{KL}(q_{\phi,\backslash\mathcal{S}}^{\mathrm{agg}}(z|x_\mathcal{S})|q_\phi(z|x_\mathcal{S}))$ for the multi-modal encoders where a mismatch can yield poor cross-generation, as an analog of the prior not matching the aggregated posterior leading to poor unconditional generation, see Remark 4.

**Proposition 1 (Marginal and conditional distribution matching)** *For any $\mathcal{S} \in \mathcal{P}(\mathcal{M})$, we have*

$$\int p_d(x_\mathcal{S})\mathcal{L}_\mathcal{S}(x_\mathcal{S}, \theta, \phi)\mathrm{d}x_\mathcal{S} + \mathcal{H}(p_d(x_\mathcal{S}))$$

$$= -\mathsf{KL}(q_\phi(z, x_\mathcal{S})|p_\theta(z, x_\mathcal{S})) \tag{$\mathsf{ZX}_{\mathrm{marginal}}$}$$

$$= -\mathsf{KL}(p_d(x_\mathcal{S})|p_\theta(x_\mathcal{S})) - \int p_d(x_\mathcal{S})\mathsf{KL}(q_\phi(z|x_\mathcal{S})|p_\theta(z|x_\mathcal{S}))\mathrm{d}x_\mathcal{S} \tag{$\mathsf{X}_{\mathrm{marginal}}$}$$

$$= -\mathsf{KL}(q_{\phi,\mathcal{S}}^{agg}(z)|p_\theta(z)) - \int q_{\phi,\mathcal{S}}^{agg}(z)\mathsf{KL}(q^\star(x_\mathcal{S}|z)|p_\theta(x_\mathcal{S}|z))\mathrm{d}x_\mathcal{S}, \tag{$\mathsf{Z}_{\mathrm{marginal}}$}$$

*where $q^\star(x_\mathcal{S}|z) = q_\phi(x_\mathcal{S}, z)/q_\phi^{agg}(z)$. Moreover, for fixed $x_\mathcal{S}$,*

$$\int p_d(x_{\backslash\mathcal{S}}|x_\mathcal{S})\mathcal{L}_{\backslash\mathcal{S}}(x, \theta, \phi)\mathrm{d}x_{\backslash\mathcal{S}} + \mathcal{H}(p_d(x_{\backslash\mathcal{S}}|x_\mathcal{S}))$$

$$= -\mathsf{KL}\left(q_\phi(z|x)p_d(x_{\backslash\mathcal{S}}|x_\mathcal{S})|p_\theta(x_{\backslash\mathcal{S}}|z)q_\phi(z|x_\mathcal{S})\right) \tag{$\mathsf{ZX}_{\mathrm{conditional}}$}$$

$$= -\mathsf{KL}(p_d(x_{\backslash\mathcal{S}}|x_\mathcal{S})|p_\theta(x_{\backslash\mathcal{S}}|x_\mathcal{S})) \tag{$\mathsf{X}_{\mathrm{conditional}}$}$$

$$\quad - \int p_d(x_{\backslash\mathcal{S}}|x_\mathcal{S})\left(\mathsf{KL}(q_\phi(z|x)|p_\theta(z|x)) - \int q_\phi(z|x)\log\frac{q_\phi(z|x_\mathcal{S})}{p_\theta(z|x_\mathcal{S})}\mathrm{d}z\right)\mathrm{d}x_{\backslash\mathcal{S}}$$

$$= -\mathsf{KL}(q_{\phi,\backslash\mathcal{S}}^{agg}(z|x_\mathcal{S})|q_\phi(z|x_\mathcal{S})) - \int q_{\phi,\backslash\mathcal{S}}^{agg}(z|x_\mathcal{S})\left(\mathsf{KL}(q^\star(x_{\backslash\mathcal{S}}|z, x_\mathcal{S})|p_\theta(x_{\backslash\mathcal{S}}|z))\right)\mathrm{d}z, \tag{$\mathsf{Z}_{\mathrm{conditional}}$}$$

*where* $q^\star(x_{\backslash S}|z, x_S) = q_\phi(z, x_{\backslash S}|x_S)/q_{\phi, \backslash S}^{agg}(z|x_S) = p_d(x_{\backslash S}|x_S)q_\phi(z|x)/q_{\phi, \backslash S}^{agg}(z|x_S).$

If $q_\phi(z|x_S)$ approximates $p_\theta(z|x_S)$ exactly, Proposition 1 implies that $\mathcal{L}_{\backslash S}(x, \theta, \phi)$ is a lower bound of $\log p_\theta(x_{\backslash S}|x_S)$. More precisely, we obtain the following log-likelihood approximation.

**Corollary 2 (Multi-modal log-likelihood approximation)** *For any modality mask $\mathcal{S}$, we have*

$$\int p_d(x) \left[ \mathcal{L}_S(x_S, \theta, \phi, 1) + \mathcal{L}_{\backslash S}(x, \theta, \phi, 1) \right] \mathrm{d}x - \int p_d(x) \left[ \log p_\theta(x) \right] \mathrm{d}x$$

$$= - \int p_d(x_S) \left[ \mathsf{KL}(q_\phi(z|x_S)|p_\theta(z|x_S)) \right] \mathrm{d}x - \int p_d(x) \left[ \mathsf{KL}(q_\phi(z|x)|p_\theta(z|x)) \right] \mathrm{d}x$$

$$+ \int p_d(x)q_\phi(z|x) \left[ \log \frac{q_\phi(z|x_S)}{p_\theta(z|x_S)} \right] \mathrm{d}z\mathrm{d}x.$$

**Proof** This follows from ($\mathsf{X}_{\mathrm{marginal}}$) and ($\mathsf{X}_{\mathrm{conditional}}$). ∎

Our approach recovers meta-learning with (latent) Neural processes (Garnelo et al., 2018b) when one optimizes only $\mathcal{L}_{\backslash S}$ with $\mathcal{S}$ determined by context-target splits, cf. Appendix B. Our analysis implies that $\mathcal{L}_S + \mathcal{L}_{\backslash S}$ is an approximation of a lower bound on the multi-modal log-likelihood that becomes tight for infinite-capacity encoders so that $q_\phi(z|x_S) = p_\theta(z|x_S)$ and $q_\phi(z|x) = p_\theta(z|x)$, see Remarks 3 and 5 for details.

**Remark 3 (Log-Likelihood approximation and Empirical Bayes)** The term

$$\int p_d(x)q_\phi(z|x) \left[ \log \frac{q_\phi(z|x_S)}{p_\theta(z|x_S)} \right] \mathrm{d}z\mathrm{d}x$$

arising in Corollary 2 and in ($\mathsf{X}_{\mathrm{conditional}}$) is not necessarily negative. Analogous to other variational approaches for learning conditional distributions such as latent Neural processes, our bound becomes an approximation of a lower bound. Note that $\mathcal{L}_S$ is maximized when $q_\phi(z|x_S) = p_\theta(z|x_S)$, see ($\mathsf{X}_{\mathrm{marginal}}$), which implies a lower bound in Corollary 2 of

$$\int p_d(x) \left[ \mathcal{L}_S(x_S, \theta, \phi, 1) + \mathcal{L}_{\backslash S}(x, \theta, \phi, 1) \right] \mathrm{d}x = \int p_d(x) \left[ \log p_\theta(x) - \mathsf{KL}(q_\phi(z|x)|p_\theta(z|x)) \right] \mathrm{d}x.$$

We can re-write the conditional expectation of $\mathcal{L}_{\backslash S}$ for any fixed $x_S$ as

$$\int p_d(x_{\backslash S}|x_S)\mathcal{L}_{\backslash S}(x, \theta, \phi, 1)\mathrm{d}x_{\backslash S} = \int p_d(x_{\backslash S}|x_S)q_\phi(z|x) \log p_\theta(x_{\backslash S}|z)\mathrm{d}z\mathrm{d}x_{\backslash S} + p_d(x_{\backslash S}|x_S)\mathcal{H}(q_\phi(z|x))\mathrm{d}x_{\backslash S}$$

$$+ \int q_{\phi, \backslash S}^{\mathrm{agg}}(z|x_S) \log q_\phi(z|x_S)\mathrm{d}z.$$

Whenever $q_\phi(z|x_S)$ can be learned independently from $q_\phi(z|x)$, the above is maximized for

$$q_\phi(z|x_S) = \int p_d(x_{\backslash S}|x_S)q_\phi(z|x)\mathrm{d}x_{\backslash S} = q_{\phi, \backslash S}^{\mathrm{agg}}(z|x_S).$$

From a different perspective, we can consider an Empirical Bayes viewpoint (Robbins, 1992; Wang et al., 2019b) wherein one chooses the hyperparameters of the (conditional) prior so that it maximizes an approximation of the conditional log-likelihood $\log p_\theta(x_{\backslash S}|x_S)$. The conditional prior $p_\vartheta^\star(z|x_S)$ in the corresponding conditional ELBO term

$$\mathcal{L}_{\backslash S}^\star(x, \theta, \phi, \vartheta, \beta) = \int q_\phi(z|x) \left[ \log p_\theta(x_{\backslash S}|z) \right] \mathrm{d}z - \beta\mathsf{KL}(q_\phi(z|x)|p_\vartheta^\star(z|x_S)). \tag{6}$$

can thus be seen as a learned prior having the parameter $\vartheta = \vartheta(\mathcal{D})$ that is learned by maximizing the above variational approximation of $\log p_\theta(x_{\backslash S}|x_S)$ over $x \sim p_d$ for $\beta = 1$, and as such depends on the

empirical multi-modal dataset $\mathcal{D}$. While the aggregated prior $q_{\phi,\backslash\mathcal{S}}^{\text{agg}}(z|x_\mathcal{S})$ is the optimal learned prior when maximizing $\mathcal{L}_{\backslash\mathcal{S}}$, this choice can lead to overfitting. Moreover, computation of the aggregated prior, or sparse approximations thereof, such as variational mixture of posteriors prior (VampPrior) in Tomczak and Welling (2017), are challenging in the conditional setup. Previous constructions in this direction (Joy et al., 2021) for learning priors for bi-modal data only considered unconditional versions, wherein pseudo-samples are not dependent on some condition $x_\mathcal{S}$. While our permutation-invariant architectures introduced below may be used for flexibly parameterizing conditional prior distributions $p_\vartheta^\star(z|x_\mathcal{S})$ as a function of $x_\mathcal{S}$ with a model that is different from the encoding distributions $q_\phi(z|x_\mathcal{S})$, we contend ourselves with choosing the same model for both the conditional prior and the encoding distribution, $p_\vartheta^\star(z|x_\mathcal{S}) = q_\phi(z|x_\mathcal{S})$. Note that the encoding distribution then features both bounds, which encourages learning encoding distributions that perform well as conditional priors and as encoding distributions. In the ideal scenario where both the generative and inference models have the flexibility to satisfy $p_d(x_{\backslash\mathcal{S}}|x_\mathcal{S}) = p_\theta(x_{\backslash\mathcal{S}}|x_\mathcal{S})$, $q_\phi(z|x_\mathcal{S}) = p_\theta(z|x_\mathcal{S})$ and $q_\phi(z|x_\mathcal{S}) = p_\theta(z|x_\mathcal{S})$, then the optimal conditional prior distribution is

$$q_{\phi,\backslash\mathcal{S}}^{\text{agg}}(z|x_\mathcal{S}) = \int p_\theta(x_{\backslash\mathcal{S}}|x_\mathcal{S})p_\theta(z|x)\mathrm{d}x_{\backslash\mathcal{S}} = \int p_\theta(x_{\backslash\mathcal{S}}|x_\mathcal{S})\frac{p_\theta(z|x_\mathcal{S})p_\theta(x_{\backslash\mathcal{S}}|x_\mathcal{S})}{p_\theta(x_{\backslash\mathcal{S}}|x_\mathcal{S})}\mathrm{d}x_{\backslash\mathcal{S}} = p_\theta(z|x_\mathcal{S}) = q_\phi(z|x_\mathcal{S}).$$

**Remark 4 (Prior-hole problem and Bayes or conditional consistency)** In the uni-modal setting, the mismatch between the prior and the aggregated prior can be large and can lead to poor unconditional generative performance because this would lead to high-probability regions under the prior that have not been trained due to their small mass under the aggregated prior (Hoffman and Johnson, 2016; Rosca et al., 2018). Equation ($\mathsf{Z}_{\text{marginal}}$) extends this to the multi-modal case, and we expect that unconditional generation can be poor if this mismatch is large. Moreover, ($\mathsf{Z}_{\text{conditional}}$) extends this conditioned on some modality subset, and we expect that cross-generation for $x_{\backslash\mathcal{S}}$ conditional on $x_\mathcal{S}$ can be poor if the mismatch between $q_{\phi,\backslash\mathcal{S}}^{\text{agg}}(z|x_\mathcal{S})$ and $q_\phi(z|x_\mathcal{S})$ is large for $x_\mathcal{S} \sim p_d$, because high-probability regions under $q_\phi(z|x_\mathcal{S})$ will not have been trained - via optimizing $\mathcal{L}_{\backslash\mathcal{S}}(x)$ - to model $x_{\backslash\mathcal{S}}$ conditional on $x_\mathcal{S}$, due to their small mass under $q_{\phi,\backslash\mathcal{S}}^{\text{agg}}(z|x_\mathcal{S})$. The mismatch will vanish when the encoders are consistent and correspond to a single Bayesian model where they approximate the true posterior distributions. A potential approach to reduce this mismatch may be to include as a regulariser the divergence between them that can be optimized by likelihood-free techniques, such as the Maximum-Mean Discrepancy (Gretton et al., 2006), as in Zhao et al. (2019) for uni-modal or unconditional models. For the mixture-based bound, the same distribution mismatch affects unconditional generation, while both the training and generative sampling distribution is $q_\phi(z|x_\mathcal{S})$ for cross-generation.

**Remark 5 (Variational gap for mixture-based bounds)** Corollary 2 shows that the variational objective can become a tight bound in the limiting case where the encoding distributions approximate the true posterior distributions. A similar result does not hold for the mixture-based multi-modal bound. Moreover, our bound can be tight for an arbitrary number of modalities in the limiting case of infinite-capacity encoders. In contrast, Daunhawer et al. (2022) show that for mixture-based bounds, this variational gap increases with each additional modality if the new modality is 'sufficiently diverse', even for infinite-capacity encoders.

**Remark 6 (Optimization, multi-task learning and the choice of $\rho$)** For simplicity, we have chosen to sample $\mathcal{S} \sim \rho$ in our experiments via the hierarchical construction $\gamma \sim \mathcal{U}(0,1)$, $m_j \sim \text{Bern}(\gamma)$ iid for all $j \in [M]$ and setting $\mathcal{S} = \{s \in [M] \colon m_j = 1\}$. The distribution $\rho$ for masking the modalities can be adjusted to accommodate various weights for different modality subsets. Indeed, (2) can be seen as a linear scalarization of a multi-task learning problem (Fliege and Svaiter, 2000; Sener and Koltun, 2018). We aim to optimize a loss vector $(\mathcal{L}_\mathcal{S} + \mathcal{L}_{\backslash\mathcal{S}})_{\mathcal{S}\subset\mathcal{M}}$, where the gradients for each $\mathcal{S} \subset \mathcal{M}$ can point in different directions, making it challenging to minimize the loss for all modalities simultaneously. Consequently, Javaloy et al. (2022) used multi-task learning techniques (e.g., as suggested in Chen et al. (2018); Yu et al. (2020)) for adjusting the gradients in mixture-based VAEs. Such improved optimization routines are orthogonal to our approach. Similarly, we do not analyze optimization issues such as initializations and training dynamics that have been found challenging for multi-modal learning (Wang et al., 2020; Huang et al., 2022).

## 2.2 Information-theoretic perspective

Beyond generative modeling, $\beta$-VAEs (Higgins et al., 2017) have been popular for representation learning and data reconstruction. Alemi et al. (2018) suggest learning a latent representation that achieves certain mutual information with the data based on upper and lower variational bounds of the mutual information. A Legendre transformation thereof recovers the $\beta$-VAE objective and allows a trade-off between information content or rate versus reconstruction quality or distortion. We show that the proposed variational objective gives rise to an analogous perspective for multiple modalities. Recall that the mutual information on the inference path[3] is given by

$$\mathrm{I}_{q_\phi}(X_\mathcal{S}, Z_\mathcal{S}) = \int q_\phi(x_\mathcal{S}, z_\mathcal{S}) \log \frac{q_\phi(x_\mathcal{S}, z_\mathcal{S})}{p_d(x_\mathcal{S}) q_{\phi,\mathcal{S}}^{\mathrm{agg}}(z_\mathcal{S})} \mathrm{d}z_\mathcal{S} \mathrm{d}x_\mathcal{S},$$

can be bounded by standard (Barber and Agakov, 2004; Alemi et al., 2016; 2018) lower and upper bounds:

$$\mathcal{H}_\mathcal{S} - D_\mathcal{S} \le \mathcal{H}_\mathcal{S} - D_\mathcal{S} + \Delta_1 = \mathrm{I}_{q_\phi}(X_\mathcal{S}, Z_\mathcal{S}) = R_\mathcal{S} - \Delta_2 \le R_\mathcal{S}, \tag{7}$$

with $\Delta_1, \Delta_2 \ge 0$ and $\mathcal{H}_\mathcal{S} \le R_\mathcal{S} + D_\mathcal{S}$. For details, see Appendix C. Consequently, by tuning $\beta$, we can vary upper and lower bounds of $\mathrm{I}_{q_\phi}(X_\mathcal{S}, Z_\mathcal{S})$ to tradeoff between compressing and reconstructing $X_\mathcal{S}$.

To arrive at a similar interpretation for the conditional bound $\mathcal{L}_{\backslash\mathcal{S}}$ that involves the conditional mutual information

$$\mathrm{I}_{q_\phi}(X_{\backslash\mathcal{S}}, Z_\mathcal{M} | X_\mathcal{S}) = \int p_d(x_\mathcal{S}) \mathsf{KL}(p_d(x_{\backslash\mathcal{S}}, z_\mathcal{M} | x_\mathcal{S})) | p_d(x_{\backslash\mathcal{S}} | x_\mathcal{S}) q_{\phi,\backslash\mathcal{S}}^{\mathrm{agg}}(z_\mathcal{M} | x_\mathcal{S})) \mathrm{d}x_\mathcal{S}$$

recalling that $q_{\phi,\backslash\mathcal{S}}^{\mathrm{agg}}(z_\mathcal{M} | x_\mathcal{S}) = \int p_d(x_{\backslash\mathcal{S}} | x_\mathcal{S}) q_\phi(z_\mathcal{M} | x) \mathrm{d}x_{\backslash\mathcal{S}}$, we set

$$R_{\backslash\mathcal{S}} = \int p_d(x) \mathsf{KL}(q_\phi(z | x) | q_\phi(z | x_\mathcal{S})) \mathrm{d}x$$

for a conditional or cross rate. Similarly, set

$$D_{\backslash\mathcal{S}} = -\int p_d(x) q_\phi(z | x) \log p_\theta(x_{\backslash\mathcal{S}} | z) \mathrm{d}z \mathrm{d}x.$$

One obtains the following bounds, see Appendix C.

**Lemma 7 (Variational bounds on the conditional mutual information)** *It holds that*

$$-\int \mathcal{L}_{\backslash\mathcal{S}}(x, \theta, \phi, \beta) p_d(\mathrm{d}x) = D_{\backslash\mathcal{S}} + \beta R_{\backslash\mathcal{S}}$$

*and for $\Delta_{\backslash\mathcal{S},1}, \Delta_{\backslash\mathcal{S},2} \ge 0$,*

$$\mathcal{H}_{\backslash\mathcal{S}} - D_{\backslash\mathcal{S}} + \Delta_{\backslash\mathcal{S},1} = \mathrm{I}_{q_\phi}(X_{\backslash\mathcal{S}}, Z_\mathcal{M} | X_\mathcal{S}) = R_{\backslash\mathcal{S}} - \Delta_{\backslash\mathcal{S},2}.$$

Consequently, by tuning $\beta$, we can vary upper and lower bounds of $\mathrm{I}_{q_\phi}(X_{\backslash\mathcal{S}}, Z_\mathcal{M} | X_\mathcal{S})$ to tradeoff between compressing relative to $q_\phi(\cdot | x_\mathcal{S})$ and reconstructing $X_{\backslash\mathcal{S}}$. Using the chain rules for entropy, we obtain that the suggested bound can be seen as a relaxation of bounds on marginal and conditional mutual information.

**Corollary 8 (Lagrangian relaxation)** *It holds that*

$$\mathcal{H} - D_\mathcal{S} - D_{\backslash\mathcal{S}} \le \mathrm{I}_{q_\phi}(X_\mathcal{S}, Z_\mathcal{S}) + \mathrm{I}_{q_\phi}(X_{\backslash\mathcal{S}}, Z_\mathcal{M} | X_\mathcal{S}) \le R_\mathcal{S} + R_{\backslash\mathcal{S}}$$

*and maximizing $\mathcal{L}$ for fixed $\beta = \frac{\partial(D_\mathcal{S} + D_{\backslash\mathcal{S}})}{\partial(R_\mathcal{S} + R_{\backslash\mathcal{S}})}$ minimizes the rates $R_\mathcal{S} + R_{\backslash\mathcal{S}}$ and distortions $D_\mathcal{S} + D_{\backslash\mathcal{S}}$.*

---

[3]We include the conditioning modalities as an index for the latent variable $Z$ when the conditioning set is unclear.

**Remark 9 (Mixture-based variational bound)** We show in Appendix C, see also Daunhawer et al. (2022), that

$$\mathcal{H}_\mathcal{M} - D_\mathcal{S} - D_\mathcal{S}^c \leq \mathcal{H}_\mathcal{M} - D_\mathcal{S} - D_\mathcal{S}^c + \Delta_1' = \mathrm{I}_{q_\phi}(X_\mathcal{M}, Z_\mathcal{S}),$$

where $\Delta_1' = \int q_\phi^{\mathrm{agg}}(z)\mathsf{KL}(q^\star(x|z)|p_\theta(x|z))\mathrm{d}z > 0$. Consequently, $\mathcal{H}_\mathcal{M} - D_\mathcal{S} - D_\mathcal{S}^c$ is a variational lower bound, while $R_\mathcal{S}$ is a variational upper bound on $\mathrm{I}_{q_\phi}(X_\mathcal{M}, Z_\mathcal{S})$, which establishes (2). Maximizing the mixture-based bound thus corresponds to encoding a single latent variable $Z_\mathcal{S}$ that maximizes the reconstruction of all modalities while at the same time being maximally compressive relative to the prior.

**Remark 10 (Optimal variational distributions)** Consider the annealed likelihood $\tilde{p}_{\beta,\theta}(x_\mathcal{S}|z) \propto p_\theta(x_\mathcal{S}|z)^{1/\beta}$ as well as the adjusted posterior $\tilde{p}_{\beta,\theta}(z|x_\mathcal{S}) \propto \tilde{p}_{\beta,\theta}(x_\mathcal{S}|z)p_\theta(z)$. The minimum of the bound $\int p_d(\mathrm{d}x)\mathcal{L}_\mathcal{S}(x)$ is attained at any $x_\mathcal{S}$ for the variational density

$$q^\star(z|x_\mathcal{S}) \propto \exp\left(\frac{1}{\beta}\left[\log p_\theta(x_\mathcal{S}|z) + \beta \log p_\theta(z)\right]\right) \propto \tilde{p}_{\beta,\theta}(z|x_\mathcal{S}), \tag{8}$$

see also Huang et al. (2020) and Remark 18. Similarly, if (8) holds, then it is readily seen that the minimum of the bound $\int p_d(\mathrm{d}x)\mathcal{L}_{\setminus \mathcal{S}}(x)$ is attained at any $x$ for the variational density $q^\star(z|x) = \tilde{p}_{\beta,\theta}(z|x)$. In contrast, as shown in Appendix 19, the optimal variational density for the mixture-based (1) multi-modal bound is attained at

$$q^\star(z|x_\mathcal{S}) \propto \tilde{p}_{\beta,\theta}(z|x_\mathcal{S})\exp\left(\int p_d(x_{\setminus \mathcal{S}}|x_\mathcal{S})\log \tilde{p}_{\beta,\theta}(x_{\setminus \mathcal{S}}|z)\mathrm{d}x_{\setminus \mathcal{S}}\right).$$

The optimal variational density for the mixture-based bound thus tilts the posterior distribution to points that achieve higher cross-modal predictions.

## 3 Permutation-invariant modality encoding

Optimizing the above multi-modal bounds requires learning variational densities with different conditioning sets. We write $h_{s,\varphi}\colon \mathsf{X}_s \mapsto \mathbb{R}^{D_E}$ for some modality-specific feature function. We recall the following multi-modal encoding functions suggested in previous work where usually $h_{s,\varphi}(x_s) = \left[\mu_{s,\varphi}(x_s)^\top, \mathrm{vec}(\Sigma_{s,\varphi}(x_s))^\top\right]^\top$ with $\mu_{s,\varphi}$ and $\Sigma_{s,\varphi}$ being the mean, respectively the (often diagonal) covariance, of a uni-modal encoder of modality $s$. Accommodating more complex variational families, such as mixture distributions for the uni-modal encoding distributions, can be more challenging for these approaches.

- MoE: $q_\varphi^{\mathrm{MoE}}(z|x_\mathcal{S}) = \frac{1}{|\mathcal{S}|}\sum_{s\in\mathcal{S}} q_\mathcal{N}(z|\mu_{s,\varphi}(x_s), \Sigma_{s,\varphi}(x_s))$, where $q_\mathcal{N}(z|\mu, \Sigma)$ is a Gaussian density with mean $\mu$ and covariance $\Sigma$.

- PoE: $q_\varphi^{\mathrm{PoE}}(z|x_\mathcal{S}) = \frac{1}{\mathcal{Z}}p_\theta(z)\prod_{s\in\mathcal{S}} q_\mathcal{N}(z|\mu_{s,\varphi}(x_s), \Sigma_{s,\varphi}(x_s))$, for some $\mathcal{Z} \in \mathbb{R}$. For Gaussian priors $p_\theta(z) = q_\mathcal{N}(z|\mu_\theta, \Sigma_\theta)$ with mean $\mu_\theta$ and covariance $\Sigma_\theta$, the multi-modal distribution $q_\varphi^{\mathrm{PoE}}(z|x_\mathcal{S})$ is Gaussian with mean

$$(\mu_\theta\Sigma_\theta + \sum_{s\in\mathcal{S}}\mu_{s,\varphi}(x_s)\Sigma_{s,\varphi}(x_s))(\Sigma_{1,\theta}^{-1} + \sum_{s\in\mathcal{S}}\Sigma_{s,\varphi}(x_s)^{-1})^{-1}$$

  and covariance

$$(\Sigma_{1,\theta}^{-1} + \sum_{s\in\mathcal{S}}\Sigma_{s,\varphi}(x_s)^{-1})^{-1}.$$

- Mixture of Product of Experts (MoPoE), see Sutter et al. (2021),

$$q_\phi^{\mathrm{MoPoE}}(z|x_\mathcal{M}) = \frac{1}{2^M}\sum_{x_\mathcal{S}\in\mathcal{P}(x_\mathcal{M})} q_\phi^{\mathrm{PoE}}(z|x_\mathcal{S}).$$

### 3.1 Learnable permutation-invariant aggregation schemes

We aim to learn a more flexible aggregation scheme under the constraint that the encoding distribution is invariant (Bloem-Reddy and Teh, 2020) with respect to the ordering of encoded features of each modality. Put differently, for all $(H_s)_{s \in \mathcal{S}} \in \mathbb{R}^{|\mathcal{S}| \times D_E}$ and all permutations $\pi \in \mathbb{S}_{\mathcal{S}}$ of $\mathcal{S}$, we assume that the conditional distribution is $\mathbb{S}_{\mathcal{S}}$-invariant, i.e. $q'_\vartheta(z|h) = q'_\vartheta(z|\pi \cdot h)$ for all $z \in \mathbb{R}^D$, where $\pi$ acts on $H = (H_s)_{s \in \mathcal{S}}$ via $\pi \cdot H = (H_{\pi(s)})_{s \in \mathcal{S}}$. We set $q_\phi(z|x_{\mathcal{S}}) = q'_\vartheta(z|h_{s,\varphi}(x_s)_{s \in \mathcal{S}})$, $\phi = (\varphi, \vartheta)$ and remark that the encoding distribution is not invariant with respect to the modalities, but becomes only invariant after applying modality-specific encoder functions $h_{s,\varphi}$. Observe that such a constraint is satisfied by the aggregation schemes above for $h_{s,\varphi}$ being the uni-modal encoders.

A variety of invariant (or equivariant) functions along with their approximation properties have been considered previously, see for instance Santoro et al. (2017); Zaheer et al. (2017); Qi et al. (2017); Lee et al. (2019); Segol and Lipman (2019); Murphy et al. (2019); Maron et al. (2019); Sannai et al. (2019); Yun et al. (2019); Bruno et al. (2021); Wagstaff et al. (2022); Zhang et al. (2022b); Li et al. (2022); Bartunov et al. (2022), and applied in different contexts such as meta-learning (Edwards and Storkey, 2016; Garnelo et al., 2018b; Kim et al., 2018; Hewitt et al., 2018; Giannone and Winther, 2022), reinforcement learning (Tang and Ha, 2021; Zhang et al., 2022a) or generative modeling of (uni-modal) sets (Li et al., 2018; 2020; Kim et al., 2021; Biloš and Günnemann, 2021; Li and Oliva, 2021). We can use such constructions to parameterize more flexible encoding distributions. Indeed, the results from Bloem-Reddy and Teh (2020) imply that for an exchangable sequence $H_{\mathcal{S}} = (H_s)_{s \in \mathcal{S}} \in \mathbb{R}^{|\mathcal{S}| \times D_E}$ and random variable $Z$, the distribution $q'(z|h_{\mathcal{S}})$ is $\mathbb{S}_{\mathcal{S}}$-invariant if and only if there is a measurable function[4] $f^\star \colon [0,1] \times \mathcal{M}(\mathbb{R}^{D_E}) \to \mathbb{R}^D$ such that

$$(H_{\mathcal{S}}, Z) \stackrel{\text{a.s.}}{=} (H_{\mathcal{S}}, f^\star(\Xi, \mathbb{M}_{H_{\mathcal{S}}})), \text{ where } \Xi \sim \mathcal{U}[0,1] \text{ and } \Xi \perp\!\!\!\perp H_{\mathcal{S}}$$

with $\mathbb{M}_{H_{\mathcal{S}}}(\cdot) = \sum_{s \in \mathcal{S}} \delta_{H_s}(\cdot)$ being the empirical measure of $h_{\mathcal{S}}$, which retains the values of $h_{\mathcal{S}}$, but discards their order. For variational densities from a location-scale family such as a Gaussian or Laplace distribution, we find it more practical to consider a different reparameterization in the form $Z = \mu(h_{\mathcal{S}}) + \sigma(h_{\mathcal{S}}) \odot \Xi$, where $\Xi$ is a sample from a parameter-free density $p$ such as a standard Gaussian and Laplace distribution, while $[\mu(h_{\mathcal{S}}), \log \sigma(h_{\mathcal{S}})] = f(h_{\mathcal{S}})$ for a PI function $f \colon \mathbb{R}^{|\mathcal{S}| \times D_E} \to \mathbb{R}^{2D}$. Likewise, for mixture distributions thereof, assume that for a PI function $f$,

$$[\mu_1(h_{\mathcal{S}}), \log \sigma_1(h_{\mathcal{S}}), \dots, \mu_K(h_{\mathcal{S}}), \log \sigma_K(h_{\mathcal{S}}), \log \omega(h_{\mathcal{S}})] = f(h_{\mathcal{S}}) \in \mathbb{R}^{2DK+K}$$

and $Z = \mu_L(h_{\mathcal{S}}) + \sigma_L(h_{\mathcal{S}}) \odot \Xi$ with $L \sim \text{Cat}(\omega(h_{\mathcal{S}}))$ denoting the sampled mixture component out of $K$ mixtures. For simplicity, we consider here only two examples of PI functions $f$ that have representations with parameter $\vartheta$ in the form

$$f_\vartheta(h_{\mathcal{S}}) = \rho_\vartheta \left( \sum_{s \in \mathcal{S}} g_\vartheta(h_{\mathcal{S}})_s \right)$$

for a function $\rho_\vartheta \colon \mathbb{R}^{D_P} \to \mathbb{R}^{D_O}$ and permutation-equivariant function $g_\vartheta \colon \mathbb{R}^{N \times D_E} \to \mathbb{R}^{N \times D_P}$.

**Example 1 (Sum Pooling Encoders)** The Deep Set (Zaheer et al., 2017) construction $f_\vartheta(h_{\mathcal{S}}) = \rho_\vartheta \left( \sum_{s \in \mathcal{S}} \chi_\vartheta(h_s) \right)$ applies the same neural network $\chi_\vartheta \colon \mathbb{R}^{D_E} \to \mathbb{R}^{D_P}$ to each encoded feature $h_s$. We assume that $\chi_\vartheta$ is a feed-forward neural network and remark that pre-activation ResNets (He et al., 2016) have been advocated for deeper $\chi_\vartheta$. For exponential family models, the optimal natural parameters of the posterior solve an optimization problem where the dependence on the generative parameters from the different modalities decomposes as a sum, see Appendix F.

**Example 2 (Set Transformer Encoders)** Let $\text{MTB}_\vartheta$ be a multi-head pre-layer-norm transformer block (Wang et al., 2019a; Xiong et al., 2020), see Appendix D for precise definitions. For some neural network $\chi_\vartheta \colon \mathbb{R}^{D_E} \to \mathbb{R}^{D_P}$, set $g_{\mathcal{S}}^0 = \chi_\vartheta(h_{\mathcal{S}})$ and for $k \in \{1, \dots, L\}$, set $g_{\mathcal{S}}^k = \text{MTB}_\vartheta(g_{\mathcal{S}}^{k-1})$. We then consider $f_\vartheta(h_{\mathcal{S}}) = \rho_\vartheta \left( \sum_{s \in \mathcal{S}} g_s^L \right)$. This can be seen as a Set Transformer (Lee et al., 2019; Zhang et al., 2022a) model

---

[4]The function $f^\star$ generally depends on the cardinality of $\mathcal{S}$. Finite-length exchangeable sequences imply a de Finetti latent variable representation only up to approximation errors (Diaconis and Freedman, 1980).

without any inducing points as for most applications, a computational complexity that scales quadratically in the number of modalities can be acceptable. In our experiments, we use layer normalization (Ba et al., 2016) within the transformer model, although, for example, set normalization (Zhang et al., 2022a) could be used alternatively.

Note that the PoE's aggregation mechanism involves taking inverses, which can only be approximated by the learned aggregation models. The considered permutation-invariant models can thus only recover a PoE scheme under universal approximation assumptions.

**Remark 11 (Mixture-of-Product-of-Experts or MoPoEs)** Sutter et al. (2021) introduced a MoPoE aggregation scheme that extends MoE or PoE schemes by considering a mixture distribution of all $2^M$ modality subsets, where each mixture component consists of a PoE model, i.e.,

$$q_\phi^{\text{MoPoE}}(z|x_\mathcal{M}) = \frac{1}{2^M} \sum_{x_\mathcal{S} \in \mathcal{P}(x_\mathcal{M})} q_\phi^{\text{PoE}}(z|x_\mathcal{S}).$$

This can also be seen as another PI model. While it does not require learning separate encoding models for all modality subsets, it, however, becomes computationally expensive to evaluate for large $M$. Our mixture models using components with a SumPooling or SelfAttention aggregation can be seen as an alternative that allows one to choose the number of mixture components $K$ to be smaller than $2^M$, with non-uniform weights, while the individual mixture components are not constrained to have a PoE form.

**Remark 12 (Pooling expert opinions)** Combining expert distributions has a long tradition in decision theory and Bayesian inference; see Genest and Zidek (1986) for early works, with popular schemes being linear pooling (i.e., MoE) or log-linear pooling (i.e., PoE with tempered densities). These are optimal schemes for minimizing different objectives, namely a weighted (forward or reverse) KL-divergence between the pooled distribution and the individual experts (Abbas, 2009). Log-linear pooling operators are externally Bayesian, allowing for consistent Bayesian belief updates when each expert updates her belief with the same likelihood function (Genest et al., 1986).

### 3.2 Permutation-equivariance and private latent variables

In principle, the general permutation invariant aggregation schemes that have been introduced could also be used for learning multi-modal models with private latent variables. For example, suppose that the generative model factorizes as

$$p_\theta(z, x) = p(z) \prod_{s \in \mathcal{M}} p_\theta(x_s|z', \tilde{z}_s) \tag{9}$$

for $z = (z', \tilde{z}_1, \dots, \tilde{z}_M) \in \mathsf{Z}$, for shared latent variables $Z'$ and private latent variable $\tilde{Z}^s$ for each $s \in \mathcal{M}$. Note that for $s \neq t \in [M]$,

$$X_s \perp\!\!\!\perp \tilde{Z}_t \mid Z', \tilde{Z}_s. \tag{10}$$

Consequently,

$$p_\theta(z', \tilde{z}_\mathcal{S}, \tilde{z}_{\backslash \mathcal{S}}|x_\mathcal{S}) = p_\theta(z', \tilde{z}_\mathcal{S}, |x_\mathcal{S})p_\theta(\tilde{z}_{\backslash \mathcal{S}}|z', \tilde{z}_\mathcal{S}, x_\mathcal{S}) = p_\theta(z', \tilde{z}_\mathcal{S}, |x_\mathcal{S})p_\theta(\tilde{z}_{\backslash \mathcal{S}}|z', \tilde{z}_\mathcal{S}). \tag{11}$$

An encoding distribution $q_\phi(z|x_\mathcal{S})$ that approximates $p_\theta(z|x_\mathcal{S})$ should thus be unaffected by the inputs $x_\mathcal{S}$ when encoding $\tilde{z}_s$ for $s \notin \mathcal{S}$, provided that, a priori, all private and shared latent variables are independent. Observe that for $f_\vartheta$ with the representation (3.1) where $\rho_\vartheta$ has aggregated inputs $y$, and that parameterizes the encoding distribution of $z = (z', \tilde{z}_\mathcal{S}, \tilde{z}_{\backslash \mathcal{S}})$, the gradients of its $i$-th dimension with respect to the modality values $x_s$ is

$$\frac{\partial}{\partial x_s} \left[ f_\vartheta(h_\mathcal{S}(x_\mathcal{S}))_i \right] = \frac{\partial \rho_{\vartheta,i}}{\partial y} \left( \sum_{t \in \mathcal{S}} g_\vartheta(h_\mathcal{S}(x_\mathcal{S})_t) \right) \frac{\partial}{\partial x_s} \left( \sum_{t \in \mathcal{S}} g_\vartheta(h_\mathcal{S}(x_\mathcal{S}))_t \right).$$

In the case of a SumPooling aggregation, the gradient simplifies to

$$\frac{\partial \rho_{\vartheta,i}}{\partial y} \left( \sum_{t \in \mathcal{S}} \chi_\vartheta(h_t(x_t)) \right) \frac{\partial \chi_\vartheta}{\partial h} (h_s(x_s)) \frac{\partial h_s(x_s)}{\partial x_s}.$$

Suppose that the $i$-th component of $\rho_\vartheta$ maps to the mean or log-standard deviation of some component of $\tilde{Z}_s$ for some $s \in \mathcal{M} \setminus \mathcal{S}$. Notice that only the first factor depends on $i$ so that for this gradient to be zero, $\rho_{\vartheta,i}$ has to be locally constant around $y = \sum_{s \in \mathcal{S}} \chi_\vartheta(h_s(x_s))$ if some other components have a non-zero gradient with respect to $X_s$. It it thus very likely that inputs $X_s$ for $s \in \mathcal{S}$ can impact the distribution of the private latent variables $\tilde{z}_{\setminus \mathcal{S}}$.

However, the specific generative model also lends itself to an alternative parameterization that guarantees that cross-modal likelihoods from $X_{\setminus \mathcal{S}}$ do not affect the encoding distribution of $\tilde{Z}_\mathcal{S}$ under our new variational objective. The assumption of private latent variables suggests an additional permutation-equivariance into the encoding distribution that approximates the posterior in (11), in the sense that for any permutation $\pi \in \mathbb{S}_\mathcal{S}$, it holds that

$$q'_\phi(\tilde{z}_\mathcal{S}|\pi \cdot h_\varphi(x_\mathcal{S}), z') = q'_\phi(\pi \cdot \tilde{z}_\mathcal{S}|h_\varphi(x_\mathcal{S}), z'),$$

assuming that all private latent variables are of the same dimension $D$.[5] Indeed, suppose we have modality-specific feature functions $h_{\varphi,s}$ such that $\{H_s = h_{\varphi,s}(X_s)\}_{s \in \mathcal{S}}$ is exchangeable. Clearly, (10) implies for any $s \neq t$ that

$$h_{\varphi,s}(X_s) \perp\!\!\!\perp \tilde{Z}_t \mid Z', \tilde{Z}_s.$$

The results from Bloem-Reddy and Teh (2020) then imply, for fixed $|\mathcal{S}|$, the existence of a function $f^\star$ such that for all $s \in \mathcal{S}$, almost surely,

$$(H_\mathcal{S}, \tilde{Z}_s) = (H_\mathcal{S}, f^\star(\Xi_s, Z', H_s, \mathbb{M}_{H_\mathcal{S}})), \text{ where } \Xi_s \sim \mathcal{U}[0,1] \text{ iid and } \Xi_s \perp\!\!\!\perp H_\mathcal{S}. \tag{12}$$

This fact suggests an alternative route to approximate the posterior distribution in (11): First, $p_\theta(\tilde{z}_{\setminus \mathcal{S}}|z', \tilde{z}_\mathcal{S})$ can often be computed analytically based on the learned or fixed prior distribution. Second, a permutation-invariant scheme can be used to approximate $p_\theta(z'|x_\mathcal{S})$. Finally, a permutation-equivariant scheme can be employed to approximate $p_\theta(\tilde{z}_\mathcal{S}|x_\mathcal{S}, z')$ with a reparameterization in the form of (12). The variational objective that explicitly uses private latent variables is detailed in Appendix E. Three examples of such permutation-equivariant schemes are given below with pseudocode for optimizing the variational objective given in Algorithm 2. Note that the assumption $q_\phi(\tilde{z}_\mathcal{S}|z', \tilde{z}_{\setminus \mathcal{S}}, x_\mathcal{S}) = q_\phi(\tilde{z}_\mathcal{S}|z', x_\mathcal{S})$ is an inductive bias that generally decreases the variational objective as it imposes a restriction on the encoding distribution that only approximates the posterior where this independence assumption holds. However, this independence assumption allows us to respect the modality-specific nature of the private latent variables during encoding. In particular, for some permutation-invariant encoder $q_\phi(z'|x_\mathcal{S})$ for the private latent variables and permutation-equivariant encoder $q_\phi(\tilde{z}_\mathcal{S}|z', x_\mathcal{S})$ for the private latent variables of the observed modalities, we can encode via

$$q_\phi(z', \tilde{z}_\mathcal{M}|x_\mathcal{S}) = q_\phi(z'|x_\mathcal{S})p_\theta(\tilde{z}_{\setminus \mathcal{S}}|z')q_\phi(\tilde{z}_\mathcal{S}|z', x_\mathcal{S})$$

so that the modality-specific information of $x_\mathcal{S}$ as encoded via $\tilde{z}_\mathcal{S}$ is not impacted by the realisation $\tilde{Z}_{\setminus \mathcal{S}}$ of modality-specific variation from the other modalities.

**Example 3 (Permutation-equivariant PoE)** Similar to previous work Wang et al. (2016); Lee and Pavlovic (2021); Sutter et al. (2020), we consider an encoding density of the form

$$q_\phi(z', \tilde{z}_\mathcal{M}|x_\mathcal{S}) = q_\varphi^{\text{PoE}}(z'|x_\mathcal{S}) \prod_{s \in \mathcal{S}} q_\mathcal{N}(\tilde{z}_s|\tilde{\mu}_{s,\varphi}(x_s), \tilde{\Sigma}_{s,\varphi}(x_s)) \prod_{s \in \mathcal{M} \setminus \mathcal{S}} p_\theta(\tilde{z}_s),$$

where

$$q_\varphi^{\text{PoE}}(z'|x_\mathcal{S}) = \frac{1}{\mathcal{Z}} p_\theta(z') \prod_{s \in \mathcal{S}} q_\mathcal{N}(z'|\mu'_{s,\varphi}(x_s), \Sigma'_{s,\varphi}(x_s))$$

is a (permutation-invariant) PoE aggregation, and we assumed that the prior density factorizes over the shared and different private variables. For each modality $s$, we encode different features $h'_{s,\varphi} = (\mu'_{s,\varphi}, \Sigma'_{s,\varphi})$ and $\tilde{h}_{s,\varphi} = (\tilde{\mu}_{s,\varphi}, \tilde{\Sigma}_{s,\varphi})$ for the shared, respectively, private, latent variables. We followed previous works (Tsai et al., 2019b; Lee and Pavlovic, 2021; Sutter et al., 2020) in that the encodings and prior distributions for the modality-specific latent variables are independent of the shared latent variables. However, this assumption can be relaxed, as long as the distributions remain Gaussian.

---

[5]The effective dimension can vary across modalities in practice if the decoders are set to mask redundant latent dimensions.

**Example 4 (Permutation-equivariant Sum-Pooling)** We consider an encoding density that is written as

$$q_\phi(z', \tilde{z}_\mathcal{M}|x_\mathcal{S}) = q_\phi^{\mathrm{SumP}}(z'|x_\mathcal{S})q_\phi^{\mathrm{Equiv\text{-}SumP}}(\tilde{z}_\mathcal{S}|z', x_\mathcal{S}) \prod_{s\in\mathcal{M}\setminus\mathcal{S}} p_\theta(\tilde{z}_s|z').$$

Here, we use a (permutation-invariant) Sum-Pooling aggregation scheme for constructing the shared latent variable $Z' = \mu'(h_\mathcal{S}) + \sigma'(h_\mathcal{S}) \odot \Xi' \sim q_\phi^{\mathrm{SumP}}(z'|x_\mathcal{S})$, where $\Xi' \sim p$ and $f_\vartheta \colon \mathbb{R}^{|\mathcal{S}|\times D_E} \to \mathbb{R}^D$ given as in Example (1) with $[\mu'(h), \log\sigma'(h)] = f_\vartheta(h)$. To sample $\tilde{Z}_\mathcal{S} \sim q_\phi^{\mathrm{Equiv\text{-}SumP}}(\tilde{z}_\mathcal{S}|z', x_\mathcal{S})$, consider functions $\chi_{j,\vartheta}\colon \mathbb{R}^{D_E} \to \mathbb{R}^{D_P}$, $j \in [3]$, and $\rho_\vartheta\colon \mathbb{R}^{D_P} \to \mathbb{R}^{D_O}$, e.g., fully-connected neural networks. We define $f_\vartheta^{\mathrm{Equiv\text{-}SumP}}\colon \mathsf{Z} \times \mathbb{R}^{|\mathcal{S}|\times D_E} \to \mathbb{R}^{|\mathcal{S}|\times D_O}$ via

$$f_\vartheta^{\mathrm{Equiv\text{-}SumP}}(z', h_\mathcal{S})_s = \rho_\vartheta\left(\left[\sum_{t\in\mathcal{S}} \chi_{0,\vartheta}(h_t)\right] + \chi_{1,\vartheta}(z') + \chi_{2,\vartheta}(h_s)\right).$$

With $\left[\tilde{\mu}(h_\mathcal{S})^\top, \log\tilde{\sigma}(h_\mathcal{S})^\top\right]^\top = f_\vartheta^{\mathrm{Equiv\text{-}SumP}}(z', h_\mathcal{S})$, we then set $\tilde{Z}_s = \tilde{\mu}(h_\mathcal{S})_s + \tilde{\sigma}(h_\mathcal{S})_s \odot \tilde{\Xi}_s$ for $\tilde{\Xi}_s \sim p$ iid, $h_s = h_{\varphi,s}(x_s)$ for modality-specific feature functions $h_{\varphi,s}\colon \mathsf{X}_s \to \mathbb{R}^{D_E}$.

**Example 5 (Permutation-equivariant Self-Attention)** Similar to a Sum-Pooling approach, we consider an encoding density that is written as

$$q_\phi(z', \tilde{z}_\mathcal{M}|x_\mathcal{S}) = q_\phi^{\mathrm{SA}}(z'|x_\mathcal{S})q_\phi^{\mathrm{Equiv\text{-}SA}}(\tilde{z}_\mathcal{S}|z', x_\mathcal{S}) \prod_{s\in\mathcal{M}\setminus\mathcal{S}} p_\theta(\tilde{z}_s|z').$$

Here, the shared latent variable $Z'$ is sampled via the permutation-invariant aggregation above by summing the elements of a permutation-equivariant transformer model of depth $L'$. For encoding the private latent variables, we follow the example above but set

$$\left[\tilde{\mu}(h_\mathcal{S})^\top, \log\tilde{\sigma}(h_\mathcal{S})^\top\right]^\top = f_\vartheta^{\mathrm{Equiv\text{-}SA}}(z', h_\mathcal{S})_s = g_\mathcal{S}^L,$$

with $g_\mathcal{S}^k = \mathrm{MTB}_\vartheta(g_\mathcal{S}^{k-1})$ an $g^0 = (\chi_{1,\vartheta}(h_s) + \chi_{2,\vartheta}(z'))_{s\in\mathcal{S}}$.

## 4 Identifiability and model extensions

### 4.1 Identifiability

Identifiability of parameters and latent variables in latent structure models is a classic problem (Koopmans and Reiersol, 1950; Kruskal, 1976; Allman et al., 2009), that has been studied increasingly for non-linear latent variable models, e.g., for ICA (Hyvarinen and Morioka, 2016; Hälvä and Hyvarinen, 2020; Hälvä et al., 2021), VAEs (Khemakhem et al., 2020a; Zhou and Wei, 2020; Wang et al., 2021; Moran et al., 2021; Lu et al., 2022; Kim et al., 2023), EBMs (Khemakhem et al., 2020b), flow-based (Sorrenson et al., 2020) or mixture models (Kivva et al., 2022).

Non-linear generative models are generally unidentifiable without imposing some structure (Hyvärinen and Pajunen, 1999; Xi and Bloem-Reddy, 2022). Yet, identifiability up to some ambiguity can be achieved in some conditional models based on observed auxiliary variables and injective decoder functions wherein the prior density is conditional on auxiliary variables. Observations from different modalities can act as auxiliary variables to obtain identifiability of conditional distributions given some modality subset under analogous assumptions.

**Example 6 (Auxiliary variable as a modality)** In the iVAE model (Khemakhem et al., 2020a), the latent variable distribution $p_\theta(z|x_1)$ is independently modulated via an auxiliary variable $X_1 = U$. Instead of interpreting this distribution as a (conditional) prior density, we view it as a posterior density given the first modality $X_1$. Khemakhem et al. (2020a) estimate a model for another modality $X_2$ by lower bounding $\log p_\theta(x_2|x_1)$ via $\mathcal{L}_{\setminus\{1\}}$ under the assumption that $q_\phi(z|x_1)$ is given by the prior density $p_\theta(z|x_1)$. Similarly, Mita et al. (2021) optimize $\log p_\theta(x_1, x_2)$ by a double VAE bound that reduces to $\mathcal{L}$ for a masking

distribution $\rho(s_1, s_2) = (\delta_1 \otimes \delta_0)(s_1, s_2)$ that always masks the modality $X_2$ and choosing to parameterize separate encoding functions for different conditioning sets. Our bound thus generalizes these procedures to multiple modalities in a scalable way.

We are interested in identifiability, conditional on having observed some non-empty modality subset $\mathcal{S} \subset \mathcal{M}$. For illustration, we translate an identifiability result from the uni-modal iVAE setting in Lu et al. (2022), which does not require the conditional independence assumption from Khemakhem et al. (2020a). We assume that the encoding distribution $q_\phi(z|x_\mathcal{S})$ approximates the true posterior $p_\theta(z|x_\mathcal{S})$ and belongs to a strongly exponential family, i.e.,

$$p_\theta(z|x_\mathcal{S}) = q_\phi(z|x_\mathcal{S}) = p_{V_{\phi,\mathcal{S}}, \lambda_{\phi,\mathcal{S}}}^{\mathrm{EF}}(z|x_\mathcal{S}), \tag{13}$$

with

$$p_{V_\mathcal{S}, \lambda_\mathcal{S}}^{\mathrm{EF}}(z|x_\mathcal{S}) = \mu(z) \exp\left[\langle V_\mathcal{S}(z), \lambda(x_\mathcal{S})\rangle - \log \Gamma_\mathcal{S}(\lambda_\mathcal{S}(x_\mathcal{S}))\right],$$

where $\mu$ is a base measure, $V_\mathcal{S} \colon \mathsf{Z} \to \mathbb{R}^k$ is the sufficient statistics, $\lambda_\mathcal{S}(x_\mathcal{S}) \in \mathbb{R}^k$ the natural parameters and $\Gamma_\mathcal{S}$ a normalizing term. Furthermore, one can only reduce the exponential component to the base measure on sets having measure zero. In this section, we assume that

$$p_\theta(x_s|z) = p_{s,\epsilon}(x_s - f_{\theta,s}(z)) \tag{14}$$

for some fixed noise distribution $p_{s,\epsilon}$ with a Lebesgue density, which excludes observation models for discrete modalities. Let $\Theta_\mathcal{S}$ be the domain of the parameters $\theta_\mathcal{S} = (f_{\backslash\mathcal{S}}, V_\mathcal{S}, \lambda_\mathcal{S})$ with $f_{\backslash\mathcal{S}} \colon \mathsf{Z} \ni z \mapsto (f_s(z))_{s \in \mathcal{M} \backslash \mathcal{S}} \in \times_{s \in \mathcal{M} \backslash \mathcal{S}} \mathsf{X}_s = \mathsf{X}_{\backslash\mathcal{S}}$. Assuming (13), note that

$$p_{\theta_\mathcal{S}}(x_{\backslash\mathcal{S}}|x_\mathcal{S}) = \int p_{V_\mathcal{S}, \lambda_\mathcal{S}}(z|x_\mathcal{S}) p_{\backslash\mathcal{S},\epsilon}(x_{\backslash\mathcal{S}} - f_{\backslash\mathcal{S}}(z)) \mathrm{d}z,$$

with $p_{\backslash\mathcal{S},\epsilon} = \otimes_{s \in \mathcal{M} \backslash \mathcal{S}} p_{s,\epsilon}$. We define an equivalence relation on $\Theta_\mathcal{S}$ by $(f_{\backslash\mathcal{S}}, V_\mathcal{S}, \lambda_\mathcal{S}) \sim_{A_\mathcal{S}} (\tilde{f}_{\backslash\mathcal{S}}, \tilde{V}_\mathcal{S}, \tilde{\lambda}_\mathcal{S})$ iff there exist invertible $A_\mathcal{S} \in \mathbb{R}^{k \times k}$ and $c_\mathcal{S} \in \mathbb{R}^k$ such that

$$V_\mathcal{S}(f_{\backslash\mathcal{S}}^{-1}(x_{\backslash\mathcal{S}})) = A_\mathcal{S} \tilde{V}_\mathcal{S}(\tilde{f}_{\backslash\mathcal{S}}^{-1}(x_{\backslash\mathcal{S}})) + c_\mathcal{S}$$

for all $x_{\backslash\mathcal{S}} \in \mathsf{X}_{\backslash\mathcal{S}}$.

**Proposition 13 (Weak identifiability)** *Consider the data generation mechanism $p_\theta(z, x) = p_\theta(z) \prod_{s \in \mathcal{M}} p_\theta(x_s|z)$ where the observation model satisfies (14) for an injective $f_{\backslash\mathcal{S}}$. Suppose further that $p_\theta(z|x_\mathcal{S})$ is strongly exponential and (13) holds. Assume that the set $\{x_{\backslash\mathcal{S}} \in \mathsf{X}_{\backslash\mathcal{S}} | \varphi_{\backslash\mathcal{S},\epsilon}(x_{\backslash\mathcal{S}}) = 0\}$ has measure zero, where $\varphi_{\backslash\mathcal{S},\epsilon}$ is the characteristic function of the density $p_{\backslash\mathcal{S},\epsilon}$. Furthermore, suppose that there exist $k + 1$ points $x_\mathcal{S}^0, \ldots, x_\mathcal{S}^k \in \mathsf{X}_\mathcal{S}$ such that*

$$L = \left[\lambda_\mathcal{S}(x_\mathcal{S}^1) - \lambda_\mathcal{S}(x_\mathcal{S}^0), \ldots, \lambda_\mathcal{S}(x_\mathcal{S}^k) - \lambda_\mathcal{S}(x_\mathcal{S}^0)\right] \in \mathbb{R}^{k \times k}$$

*is invertible. Then $p_{\theta_\mathcal{S}}(x_{\backslash\mathcal{S}}|x_\mathcal{S}) = p_{\tilde{\theta}_\mathcal{S}}(x_{\backslash\mathcal{S}}|x_\mathcal{S})$ for all $x \in \mathsf{X}$ implies $\theta \sim_{A_\mathcal{S}} \tilde{\theta}$.*

This result follows from Theorem 4 in Lu et al. (2022). Note that $p_{\theta_\mathcal{S}}(x_{\backslash\mathcal{S}}|x_\mathcal{S}) = p_{\tilde{\theta}_\mathcal{S}}(x_{\backslash\mathcal{S}}|x_\mathcal{S})$ for all $x \in \mathsf{X}$ implies with the regularity assumption on $\varphi_{\backslash\mathcal{S},\epsilon}$ that the transformed variables $Z = f_{\backslash\mathcal{S}}^{-1}(X_{\backslash\mathcal{S}})$ and $\tilde{Z} = \tilde{f}_{\backslash\mathcal{S}}^{-1}(X_{\backslash\mathcal{S}})$ have the same density function conditional on $X_\mathcal{S}$.

**Remark 14 (Conditional identifiability)** The identifiability result above is about conditional models and does not contradict the un-identifiability of VAEs: When $\mathcal{S} = \emptyset$ and we view $x = x_\mathcal{M}$ as one modality, then the parameters of $p_{\theta_\emptyset}(x)$ characterized by the parameters $V_\emptyset$ and $\lambda_\emptyset$ of the prior $p_{\theta_\emptyset}(z|x_\emptyset)$ and the encoders $f_\mathcal{M}$ will not be identifiable as the invertibility condition will not be satisfied.

**Remark 15 (Private latent variables)** For models with private latent variables, we might not expect that conditioning on $X_\mathcal{S}$ helps to identify $\tilde{Z}_{\backslash\mathcal{S}}$ as

$$p_\theta(z', \tilde{z}_\mathcal{S}, \tilde{z}_{\backslash\mathcal{S}}|x_\mathcal{S}) = p_\theta(z', \tilde{z}_\mathcal{S}|x_\mathcal{S}) p_\theta(\tilde{z}_{\backslash\mathcal{S}}|z', \tilde{z}_\mathcal{S}).$$

Indeed, Proposition 13 will not apply in such models as $f_{\backslash\mathcal{S}}$ will not be injective.

**Remark 16 (Data supported on low-dimensional manifolds)** Note that (14) and (13) imply that each modality has a Lebesgue density under the generative model. This assumption may not hold for some modalities, such as imaging data that can be supported (closely) on a lower-dimensional manifold (Roweis and Saul, 2000), causing issues in likelihood-based methods such as VAEs (Dai and Wipf, 2018; Loaiza-Ganem et al., 2022). Moreover, different conditioning sets or modalities may result in different dimensions of the underlying manifold for conditional data (Zheng et al., 2022). Some two-step approaches (Dai and Wipf, 2018; Zheng et al., 2022) first estimate the dimension $r$ of the ground-truth manifold as a function of the encoder variance relative to the variance under the (conditional) prior for each latent dimension $i$, $i \in [D]$, with $r \leq D$. It would, therefore, be interesting to analyze in future work if more flexible aggregation schemes that do not impose strong biases on the variance components of the encoder can better learn the manifold dimensions in conditional or multi-modal models following an analogous two-step approach.

Recall that the identifiability considered here concerns parameters of the multi-modal posterior distribution and the conditional generative distribution. It is thus preliminary to estimation and only concerns the generative model and not the inference approach. However, both the multi-modal posterior distribution and the conditional generative distribution are intractable. In practice, we thus replace them with approximations. We believe that our inference approach is beneficial for this type of identifiability when making these variational approximations because (a) unlike some other variational bounds, the posterior is the optimal variational distribution with $\mathcal{L}_{\backslash \mathcal{S}}(x)$ being an approximation of a lower bound on $\log p_\theta(x_{\backslash \mathcal{S}}|x_{\mathcal{S}})$, see Remark 10, and (b) the trainable aggregation schemes can be more flexible for approximating the optimal encoding distribution.

## 4.2   Mixture models

An alternative to the choice of uni-modal prior densities $p_\theta$ has been to use Gaussian mixture priors (Johnson et al., 2016; Jiang et al., 2017; Dilokthanakul et al., 2016) or more flexible mixture models (Falck et al., 2021). Following previous work, we include a latent cluster indicator variable $c \in [K]$ that indicates the mixture component out of $K$ possible mixtures with augmented prior $p_\theta(c, z) = p_\theta(c)p_\theta(z|c)$. The classic example is $p_\theta(c)$ being a categorical distribution and $p_\theta(z|c)$ a Gaussian with mean $\mu_c$ and covariance matrix $\Sigma_c$. Similar to Falck et al. (2021) that use an optimal variational factor in a mean-field model, we use an optimal factor of the cluster indicator in a structured variational density $q_\phi(c, z|x_{\mathcal{S}}) = q_\phi(z|x_{\mathcal{S}})q_\phi(c|z, x_{\mathcal{S}})$ with $q_\phi(c|z, x_{\mathcal{S}}) = p_\theta(c|z)$. Appendix G details how one can optimize an augmented multi-modal bound. Concurrent work (Palumbo et al., 2024) considered a similar optimal variational factor for a discrete mixture model under a MoE aggregation.

## 4.3   Missing modalities

In practical applications, modalities can be missing for different data points. We describe this missingness pattern by missingness mask variables $m_s \in \{0, 1\}$ where $m_s = 1$ indicates that observe modality $s$, while $m_s = 0$ means it is missing. The joint generative model with missing modalities will be of the form $p_\theta(z, x, m) = p_\theta(z) \prod_{s \in \mathcal{M}} p_\theta(x_s|z)p_\theta(m|x)$ for some distribution $p_\theta(m|x)$ over the mask variables $m = (m_s)_{s \in \mathcal{M}}$. For $\mathcal{S} \subset \mathcal{M}$, we denote by $x_{\mathcal{S}}^o = \{x_s : m_s = 1, s \in \mathcal{S}\}$ and $x_{\mathcal{S}}^m = \{x_s : m_s = 0, s \in \mathcal{S}\}$ the set of observed, respectively missing, modalities. The full likelihood of the observed and missingness masks becomes then $p_\theta(x_{\mathcal{S}}^o, m) = \int p_\theta(z) \prod_{s \in \mathcal{S}} p_\theta(x_s|z)p_\theta(m|x)\mathrm{d}x_s^m\mathrm{d}z$. If $p_\theta(m|x)$ does not depend on the observations, that is, observations are missing completely at random (Rubin, 1976), then the missingness mechanisms $p_\theta(m|x)$ for inference approaches maximizing $p_\theta(x^o, m)$ can be ignored. Consequently, one can instead concentrate on maximizing $\log p_\theta(x^o)$ only, based on the joint generative model $p_\theta(z, x^o) = p_\theta(z) \prod_{\{s \in \mathcal{M} : m_s = 1\}} p_\theta(x_s|z)$. In particular, one can employ the variational objectives above by considering only the observed modalities. Since masking operations are readily supported for the considered permutation-invariant models, appropriate imputation strategies (Nazabal et al., 2020; Ma et al., 2019) for the encoded features of the missing modalities are not necessarily required. Settings allowing for not (completely) at random missingness have been considered in the uni-modal case, for instance, in Ipsen et al. (2021); Ghalebikesabi et al. (2021); Gong et al. (2021), and we leave multi-modal extensions thereof for future work for a given aggregation approach.

## 5    Experiments

We conduct a series of numerical experiments to illustrate the effects of different variational objectives and aggregation schemes. Recall that the full reconstruction log-likelihood is the negative full distortion $-D_{\mathcal{M}}$ based on all modalities, while the full rate $R_{\mathcal{M}}$ is the averaged KL between the encoding distribution of all modalities and the prior. Note that mixture-based bounds maximize directly the cross-modal log-likelihood $-D^c_{\backslash\mathcal{S}}$, see (4), and do not contain a cross-rate term $R_{\backslash\mathcal{S}}$, i.e. the KL between the encoding distribution for all modalities relative to a modality subset, as a regulariser, in contrast to our objective (Lemma 7 and Corollary 8). The log-likelihood should be higher if a generative model is able to capture modality-specific information for models trained with $\beta = 1$. For arbitrary $\beta$, we can take a rate-distortion perspective and look at how different generative models self-reconstruct all modalities, i.e., the full reconstruction term $-D_{\mathcal{M}}$, relative to the KL-divergence between the multi-modal encoding distribution and the prior, i.e. $R_{\mathcal{M}}$. This corresponds to a rate-distortion analysis of a VAE that merges all modalities into a single modality. A high full-reconstruction term is thus indicative of the encoder and decoder being able to reconstruct all modalities precisely so that they do not produce an average prediction. Note that neither our objective nor the mixture-based bound optimize for the full-reconstruction term directly.

### 5.1    Linear multi-modal VAEs

The relationship between uni-modal VAEs and probabilistic PCA (Tipping and Bishop, 1999) has been studied in previous work (Dai et al., 2018; Lucas et al., 2019; Rolinek et al., 2019; Huang et al., 2020; Mathieu et al., 2019). We analyze how different multi-modal fusion schemes and multi-modal variational objectives affect (a) the learned generative model in terms of its true marginal log-likelihood (LLH) and (b) the latent representations in terms of information-theoretic quantities and identifiability. To evaluate the (weak) identifiability of the method, we follow Khemakhem et al. (2020a;b) to compute the mean correlation coefficient (MCC) between the true latent variables $Z$ and samples from the variational distribution $q_\phi(\cdot|x_{\mathcal{M}})$ after an affine transformation using CCA.

**Generative model.**    Suppose that a latent variable $Z$ taking values in $\mathbb{R}^D$ is sampled from a standard Gaussian prior $p_\theta(z) = \mathcal{N}(0, \mathrm{I})$ generates $M$ data modalities $X_s \in \mathbb{R}^{D_s}$, $D \leq D_s$, based on a linear decoding model $p_\theta(x_s|z) = \mathcal{N}(W_s z + b_s, \sigma^2 \mathrm{I})$ for a factor loading matrix $W_s \in \mathbb{R}^{D_s \times D}$, bias $b_s \in \mathbb{R}^{D_s}$ and observation scale $\sigma > 0$. Note that the annealed likelihood function $\tilde{p}_{\beta,\theta}(x_s|z) = \mathcal{N}(W_s z + b_s, \beta \sigma^2 \mathrm{I})$ corresponds to a scaling of the observation noise, so that we consider only the choice $\sigma = 1$, set $\sigma_\beta = \sigma \beta^{1/2}$ and vary $\beta > 0$. It is obvious that for any $\mathcal{S} \subset \mathcal{M}$, it holds that $\tilde{p}_{\beta,\theta}(x_{\mathcal{S}}|z) = \mathcal{N}(W_{\mathcal{S}} z + b_{\mathcal{S}}, \sigma_\beta^2 \mathrm{I}_{\mathcal{S}})$, where $W_{\mathcal{S}}$ and $b_{\mathcal{S}}$ are given by concatenating row-wise the emission or bias matrices for modalities in $\mathcal{S}$, while $\sigma_\beta^2 \mathrm{I}_{\mathcal{S}}$ is the diagonal matrix of the variances of the corresponding observations. By standard properties of Gaussian distributions, it follows that $\tilde{p}_{\beta,\theta}(x_{\mathcal{S}}) = \mathcal{N}(b_{\mathcal{S}}, C_{\mathcal{S}})$ where $C_{\mathcal{S}} = W_{\mathcal{S}} W_{\mathcal{S}}^\top + \sigma_\beta^2 \mathrm{I}_{\mathcal{S}}$ is the data covariance matrix. Furthermore, with $K_{\mathcal{S}} = W_{\mathcal{S}}^\top W_{\mathcal{S}} + \sigma_\beta^2 \mathrm{I}_d$, the adjusted posterior is $\tilde{p}_{\beta,\theta}(z|x_{\mathcal{S}}) = \mathcal{N}(K_{\mathcal{S}}^{-1} W_{\mathcal{S}}^\top (x_{\mathcal{S}} - b_{\mathcal{S}}), \sigma_\beta^2 \mathrm{I}_d K_{\mathcal{S}}^{-1})$. If we sample orthogonal rows of $W$, the posterior covariance becomes diagonal so that it can - in principle - be well approximated by an encoding distribution with a diagonal covariance matrix. Indeed, the inverse of the posterior covariance matrix is only a function of the generative parameters of the modalities within $\mathcal{S}$ and can be written as the sum $\sigma_\beta^2 \mathrm{I} + W_{\mathcal{S}}^\top W_{\mathcal{S}} = \sigma_\beta^2 \mathrm{I} + \sum_{s \in \mathcal{S}} W_s^\top W_s$, while the posterior mean function is $x_{\mathcal{S}} \mapsto (\sigma_\beta^2 \mathrm{I} + \sum_{s \in \mathcal{S}} W_s^\top W_s)^{-1} \sum_{s \in \mathcal{S}} W_s(x_s - b_s)$.

**Illustrative example.**    We consider a bi-modal setup comprising a less noisy and more noisy modality. Concretely, for a latent variable $Z = (Z_1, Z_2, Z_3) \in \mathbb{R}^3$, assume that the observed modalities can be represented as

$$X_1 = Z_0 + Z_1 + U_1$$
$$X_2 = Z_0 + 10 Z_2 + U_2,$$

for a standard Gaussian prior $Z \sim \mathcal{N}(0, \mathrm{I})$ and independent noise variables $U_1, U_2 \sim \mathcal{N}(0, 1)$. Note that the second modality is more noisy compared to the first one. The results in Table 1 for the obtained log-likelihood

Table 1: Gaussian model with a noisy and less noisy modality. Relative difference of the true MLE vs the (analytical) LLH from the learned model in the first two columns, followed by multi-modal information theoretic quantities.

| Aggregation | Relative LLH gap | | Full Reconstruction | | Full Rates | | Cross Prediction | | Cross Rates | |
|---|---|---|---|---|---|---|---|---|---|---|
| | our obj. | mixture bound | our obj. | mixture bound | our obj. | mixture bound | our ob. | mixture bound | our obj. | mixture bound |
| PoE | 1.29 | 7.11 | $-2.30 \cdot 10^{35}$ | $-2.2 \cdot 10^{35}$ | $2.1 \cdot 10^{35}$ | $2.0 \cdot 10^{35}$ | $-2.4 \cdot 10^{34}$ | $-1.9 \cdot 10^{35}$ | $1.4 \cdot 10^{35}$ | $1.7 \cdot 10^{35}$ |
| MoE | 0.11 | 0.6 | -32.07 | -30.09 | **1.02** | 2.84 | -33.27 | -28.52 | 2.37 | 19.33 |
| SumPooling | $\mathbf{3.6 \cdot 10^{-5}}$ | 0.06 | **-2.84** | -3.23 | 2.88 | 2.82 | -52.58 | **-27.26** | 1.42 | 27.35 |
| SelfAttention | $\mathbf{3.4 \cdot 10^{-5}}$ | 0.06 | **-2.85** | -3.23 | 2.87 | 2.82 | -52.59 | **-27.25** | 1.42 | 27.41 |

Table 2: Gaussian model with five modalities: Relative difference of true LLH to the learned LLH. MCC to true latent. The generative model for the invariant aggregation schemes uses dense decoders, whereas the ground truth model for the permutation-equivariant encoders uses sparse decoders to account for private latent variables. We report mean values with standard deviations in parentheses over five independent runs.

| Aggregation | Invariant aggregation | | | | Equivariant aggregation | | | |
|---|---|---|---|---|---|---|---|---|
| | Proposed objective | | Mixture bound | | Proposed objective | | Mixture bound | |
| | LLH Gap | MCC | LLH Gap | MCC | LLH Gap | MCC | LLH Gap | MCC |
| PoE | 0.03 (0.058) | 0.75 (0.20) | 0.04 (0.074) | 0.77 (0.21) | **0.00 (0.000)** | **0.91 (0.016)** | 0.01 (0.001) | 0.88 (0.011) |
| MoE | 0.01 (0.005) | 0.82 (0.04) | 0.02 (0.006) | 0.67 (0.03) | | | | |
| SumPooling | **0.00 (0.000)** | **0.84 (0.00)** | 0.00 (0.002) | **0.84 (0.02)** | **0.00 (0.000)** | 0.85 (0.004) | **0.00 (0.000)** | 0.82 (0.003) |
| SelfAttention | **0.00 (0.003)** | **0.84 (0.00)** | 0.02 (0.007) | 0.83 (0.00) | **0.00 (0.000)** | 0.83 (0.006) | **0.00 (0.000)** | 0.83 (0.003) |

values show first that learnable aggregation models yield higher log-likelihoods[6], and second that our bound yields higher log-likelihood values compared to mixture-based bounds for any given fixed aggregation model. We also compute various information theoretic quantities, confirming that our bound leads to higher full reconstructions at higher full rates and lower cross predictions at lower cross rates compared to mixture-based bounds. More flexible aggregation schemes increase the full and cross predictions for any given bound while not necessarily increasing the full or cross rates, i.e., they can result in an improved point within a rate-distortion curve for some configurations.

**Simulation study.** We consider $M = 5$ modalities following multi-variate Gaussian laws. We consider generative models, where all latent variables are shared across all modalities, as well as generative models, where only parts of the latent variables are shared across all modalities, while the remaining latent variables are modality-specific. The setting of private latent variables can be incorporated by imposing sparsity structures on the decoding matrices and allows us to analyze scenarios with considerable modality-specific variation described through private latent variables. We provide more details about the data generation mechanisms in Appendix J. For illustration, we use multi-modal encoders with shared latent variables using invariant aggregations in the first case and multi-modal encoders that utilize additional equivariant aggregations for the private latent variables in the second case. Results in Table 2 suggest that more flexible aggregation schemes improve the LLH and the identifiability for both variational objectives. Furthermore, our new bound yields higher LLH for a given aggregation scheme.

## 5.2 Non-linear identifiable models

**Auxiliary labels as modalities.** We construct artificial data following Khemakhem et al. (2020a), with the latent variables $Z \in \mathbb{R}^D$ being conditionally Gaussian having means and variances that depend on an observed index value $X_2 \in [K]$. More precisely, $p_\theta(z|x_2) = \mathcal{N}(\mu_{x_2}, \Sigma_{x_2})$, where $\mu_c \sim \otimes \mathcal{U}(-5, 5)$ and $\Sigma_c = \text{diag}(\Lambda_c)$, $\Lambda_c \sim \otimes \mathcal{U}(0.5, 3)$ iid for $c \in [K]$. The marginal distribution over the labels is uniform $\mathcal{U}([K])$ so that the prior density $p_\theta(z) = \int_{[K]} p_\theta(z|x_2) p_\theta(x_2) \mathrm{d}x_2$ becomes a Gaussian mixture. We choose an injective decoding function $f_1 \colon \mathbb{R}^D \to \mathbb{R}^{D_1}$, $D \leq D_1$, as a composition of MLPs with LeakyReLUs and full rank weight matrices having monotonically increasing row dimensions (Khemakhem et al., 2020b), with iid randomly sampled entries. We assume $X_1|Z \sim \mathcal{N}(f_1(Z), \sigma^2 \mathrm{I})$ and set $\sigma = 0.1$, $D = D_1 = 2$. $f_1$ has a

---

[6]We found that a PoE model can have numerical issues here.

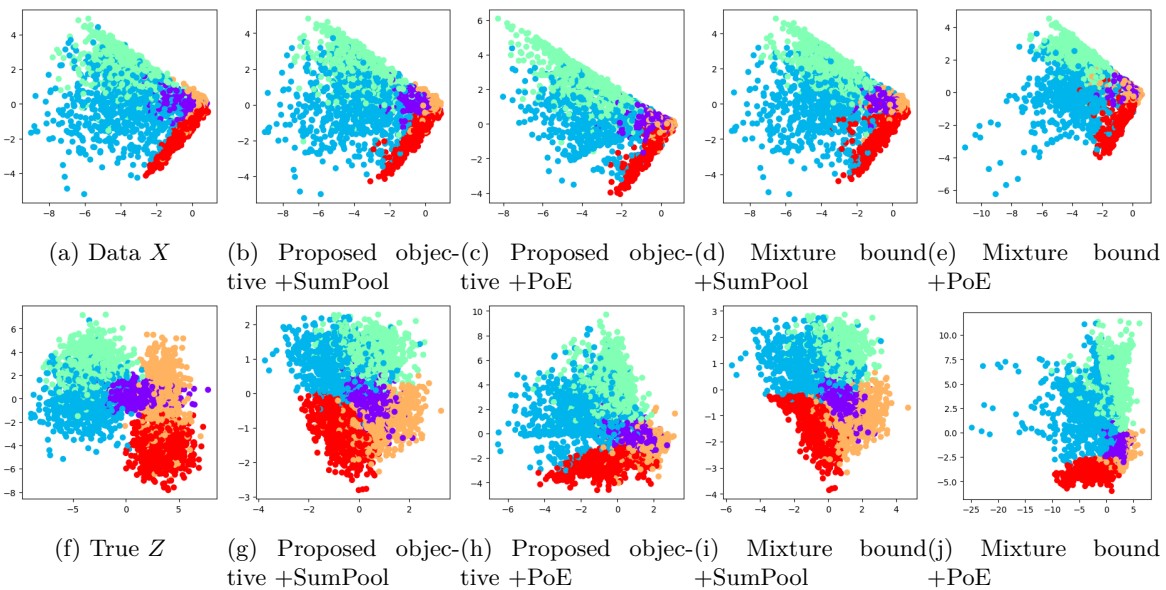

Figure 3: Continuous data modality in (a) and reconstructions using different bounds and fusion models in (b)-(e). The true latent variables are shown in (f), with the inferred latent variables in (g)-(j) with a linear transformation indeterminacy. Labels are color-coded.

single hidden layer of size $D_1 = 2$. One realization of bi-modal data $X$, the true latent variable $Z$, as well as inferred latent variables and reconstructed data for a selection of different bounds and aggregation schemes, are shown in Figure 3, with more examples given in Figures 6 and 7. We find that learning the aggregation model through a SumPooling model improves the data reconstruction and better recovers the ground truth latents, up to rotations, in contrast to a PoE model. Simulating five different such datasets, the results in Table 3 indicate first that our bound obtains better log-likelihood estimates for different fusion schemes. Second, it demonstrates the advantages of our new fusion schemes that achieve better log-likelihoods for both bounds. Third, it shows the benefit of using aggregation schemes that have the capacity to accommodate prior distributions different from a single Gaussian. Also, MoE schemes lead to low MCC values, while PoE schemes have high MCC values.

Table 3: Non-linear identifiable model with one real-valued modality and an auxiliary label acting as a second modality: The first four rows use a fixed standard Gaussian prior, while the last four rows use a Gaussian mixture prior with 5 components. Mean and standard deviation over 4 repetitions. Log-likelihoods are estimated using importance sampling with 64 particles.

| Aggregation | Proposed objective | | | Mixture bound | | |
| --- | --- | --- | --- | --- | --- | --- |
| | LLH ($\beta = 1$) | MCC ($\beta = 1$) | MCC ($\beta = 0.1$) | LLH ($\beta = 1$) | MCC ($\beta = 1$) | MCC ($\beta = 0.1$) |
| PoE | -43.4 (10.74) | 0.98 (0.006) | 0.99 (0.003) | -318 (361.2) | 0.97 (0.012) | 0.98 (0.007) |
| MoE | -20.5 (6.18) | 0.94 (0.013) | 0.93 (0.022) | -57.9 (6.23) | 0.93 (0.017) | 0.93 (0.025) |
| SumPooling | -17.9 (3.92) | 0.99 (0.004) | 0.99 (0.002) | -18.9 (4.09) | 0.99 (0.005) | 0.99 (0.008) |
| SelfAttention | -18.2 (4.17) | 0.99 (0.004) | 0.99 (0.003) | -18.6 (3.73) | 0.99 (0.004) | 0.99 (0.007) |
| SumPooling | **-15.4 (2.12)** | **1.00 (0.001)** | 0.99 (0.004) | -18.6 (2.36) | 0.98 (0.008) | 0.99 (0.006) |
| SelfAttention | **-15.2 (2.05)** | **1.00 (0.001)** | **1.00 (0.004)** | -18.6 (2.27) | 0.98 (0.014) | 0.98 (0.006) |
| SumPoolingMixture | **-15.1 (2.15)** | **1.00 (0.001)** | 0.99 (0.012) | -18.2 (2.80) | 0.98 (0.010) | 0.99 (0.005) |
| SelfAttentionMixture | **-15.3 (2.35)** | 0.99 (0.005) | 0.99 (0.004) | -18.4 (2.63) | 0.99 (0.007) | 0.99 (0.007) |

**Multiple modalities.** Considering the same generative model for $Z$ with a Gaussian mixture prior, suppose now that instead of observing the auxiliary label, we observe multiple modalities $X_s \in \mathbb{R}^{D_s}$, $X_s | Z \sim \mathcal{N}(f_s(Z), \sigma^2 \mathrm{I})$, for injective MLPs $f_s$ constructed as above, with $D = 10$, $D_s = 25$, $\sigma = 0.5$

Table 4: Partially observed ($\eta = 0.5$) and fully observed ($\eta = 0$) non-linear identifiable model with 5 modalities: The first four rows use a fixed standard Gaussian prior, while the last four rows use a Gaussian mixture prior.

| | Partially observed | | | | Fully observed | | | |
| | Proposed objective | | Mixture bound | | Proposed objective | | Mixture bound | |
| Aggregation | LLH | MCC | LLH | MCC | LLH | MCC | LLH | MCC |
|---|---|---|---|---|---|---|---|---|
| PoE | -250.9 (5.19) | 0.94 (0.015) | -288.4 (8.53) | 0.93 (0.018) | -473.6 (9.04) | 0.98 (0.005) | -497.7 (11.26) | 0.97 (0.008) |
| MoE | -250.1 (4.77) | 0.92 (0.022) | -286.2 (7.63) | 0.90 (0.019) | -477.9 (8.50) | 0.91 (0.014) | -494.6 (9.20) | 0.92 (0.004) |
| SumPooling | -249.6 (4.85) | 0.95 (0.016) | -275.6 (7.35) | 0.92 (0.031) | -471.4 (8.29) | **0.99 (0.004)** | -480.5 (8.84) | 0.98 (0.005) |
| SelfAttention | -249.7 (4.83) | 0.95 (0.014) | -275.5 (7.45) | 0.93 (0.022) | -471.4 (8.97) | **0.99 (0.002)** | -482.8 (10.51) | 0.98 (0.004) |
| SumPooling | -247.3 (4.23) | 0.95 (0.009) | -269.6 (7.42) | 0.94 (0.018) | **-465.4 (8.16)** | 0.98 (0.002) | -475.1 (7.54) | 0.98 (0.003) |
| SelfAttention | -247.5 (4.22) | 0.95 (0.013) | -269.9 (6.06) | 0.93 (0.022) | -469.3 (4.76) | 0.98 (0.003) | -474.7 (8.20) | 0.98 (0.002) |
| SumPoolingMixture | **-244.8 (4.44)** | 0.95 (0.011) | -271.9 (6.54) | 0.93 (0.021) | **-464.5 (8.16)** | **0.99 (0.003)** | -474.2 (7.61) | 0.98 (0.004) |
| SelfAttentionMixture | -245.4 (4.55) | **0.96 (0.010)** | -270.3 (5.96) | 0.94 (0.016) | **-464.4 (8.50)** | **0.99 (0.003)** | -473.6 (8.24) | 0.98 (0.002) |

Table 5: Test LLH estimates for the joint data (M+S+T) and marginal data (importance sampling with 512 particles). The first part of the table is based on the same generative model with shared latent variable $Z \in \mathbb{R}^{40}$, while the second part of the table is based on a restrictive generative model with a shared latent variable $Z' \in \mathbb{R}^{10}$ and modality-specific latent variables $\tilde{Z}_s \in \mathbb{R}^{10}$.

| | Proposed objective | | | | Mixture bound | | | |
| Aggregation | M+S+T | M | S | T | M+S+T | M | S | T |
|---|---|---|---|---|---|---|---|---|
| PoE+ | 6872 (9.62) | **2599 (5.6)** | 4317 (1.1) | -9 (0.2) | 5900 (10) | 2449 (10.4) | 3443 (11.7) | -19 (0.4) |
| PoE | 6775 (54.9) | 2585 (18.7) | 4250 (8.1) | -10 (2.2) | 5813 (1.2) | 2432 (11.6) | 3390 (17.5) | -19 (0.1) |
| MoE+ | 5428 (73.5) | 2391 (104) | 3378 (92.9) | -74 (88.7) | 5420 (60.1) | 2364 (33.5) | 3350 (58.1) | -112 (133.4) |
| MoE | 5597 (26.7) | 2449 (7.6) | 3557 (26.4) | -11 (0.1) | 5485 (4.6) | 2343 (1.8) | 3415 (5.0) | -17 (0.4) |
| SumPooling | **7056 (124)** | 2478 (9.3) | **4640 (114)** | **-6 (0.0)** | 6130 (4.4) | 2470 (10.3) | 3660 (1.5) | -16 (1.6) |
| SelfAttention | **7011 (57.9)** | 2508 (18.2) | **4555 (38.1)** | -7 (0.5) | 6127 (26.1) | 2510 (12.7) | 3621 (8.5) | -13 (0.2) |
| PoE+ | 6549 (33.2) | 2509 (7.8) | 4095 (37.2) | -7 (0.2) | 5869 (29.6) | 2465 (4.3) | 3431 (8.3) | -19 (1.7) |
| SumPooling | 6337 (24.0) | 2483 (9.8) | 3965 (16.9) | **-6 (0.2)** | 5930 (23.8) | 2468 (16.8) | 3491 (18.3) | -7 (0.1) |
| SelfAttention | 6662 (20.0) | 2516 (8.8) | 4247 (31.2) | **-6 (0.4)** | 6716 (21.8) | 2430 (26.9) | 4282 (49.7) | -27 (1.1) |

and $K = M = 5$. We consider a semi-supervised setting where modalities are missing completely at random, as in Zhang et al. (2019), with a missing rate $\eta$ as the sample average of $\frac{1}{|\mathcal{M}|}\sum_{s \in \mathcal{M}}(1 - M_s)$. Table 4 shows that using the new variational objective improves the LLH and the identifiability of the latent representation. Furthermore, using learnable aggregation schemes benefits both variational objectives.

### 5.3 MNIST-SVHN-Text

Following previous work (Sutter et al., 2020; 2021; Javaloy et al., 2022), we consider a tri-modal dataset based on augmenting the MNIST-SVHN dataset (Shi et al., 2019) with a text-based modality. Herein, SVHN consists of relatively noisy images, whilst MNIST and text are clearer modalities. Multi-modal VAEs have been shown to exhibit differing performances relative to their multi-modal coherence, latent classification accuracy or test LLH, see Appendix I for definitions. Previous works often differ in their hyperparameters, from neural network architectures, latent space dimensions, priors and likelihood families, likelihood weightings, decoder variances, etc. We have chosen the same hyperparameters for all models, thereby providing a clearer disentanglement of how either the variational objective or the aggregation scheme affects different multi-modal evaluation measures. In particular, we consider multi-modal generative models with (i) shared latent variables and (ii) private and shared latent variables. We also consider PoE or MoE schemes (denoted PoE+, resp., MoE+) with additional neural network layers in their modality-specific encoding functions so that the number of parameters matches or exceeds those of the introduced PI models, see Appendix M.5 for details. For models without private latent variables, estimates of the test LLHs in Table 5 suggest that our bound improves the LLH across different aggregation schemes for all modalities and different $\beta$s (Table 7), with similar results for PE schemes, except for a Self-Attention model. More flexible fusion schemes yield higher LLHs for both bounds. Qualitative results for the reconstructed modalities are given in Figures

4 with shared latent variables, in Figure 10 for different $\beta$-hyperparameters and in Figure 11 for models with private latent variables. Cross-generation of the SVHN modality is challenging for the mixture-based bound with all aggregation schemes. In contrast, our bound, particularly when combined with learnable aggregation schemes, leads to more realistic samples of the cross-generated SVHN modality. No variational objective or aggregation scheme performs best across all modalities by the generative coherence measures (see Table 6 for uni-modal inputs, Table 8 for bi-modal ones and Tables 9- 12 for models with private latent variables and different $\beta$s), along with reported results from external baselines (MVAE, MMVAE, MoPoE, MMJSD, MVTCAE). Overall, our objective is slightly more coherent for cross-generating SVHN or Text, but less coherent for MNIST. The mixture-based bound tends to improve the unsupervised latent classification accuracy across different fusion approaches and modalities, see Table 13. To provide complementary insights into the trade-offs for the different objectives and fusion schemes, we consider a multi-modal rate-distortion evaluation in Figure 5. Ignoring MoE where reconstructions are similar, our bound improves the full reconstruction with higher full rates and across various fusion schemes. The mixture-based bound yields improved cross-predictions for all aggregation models, with increased cross-rate terms. Flexible PI architectures for our bound improve the full reconstruction, even at lower full rates.

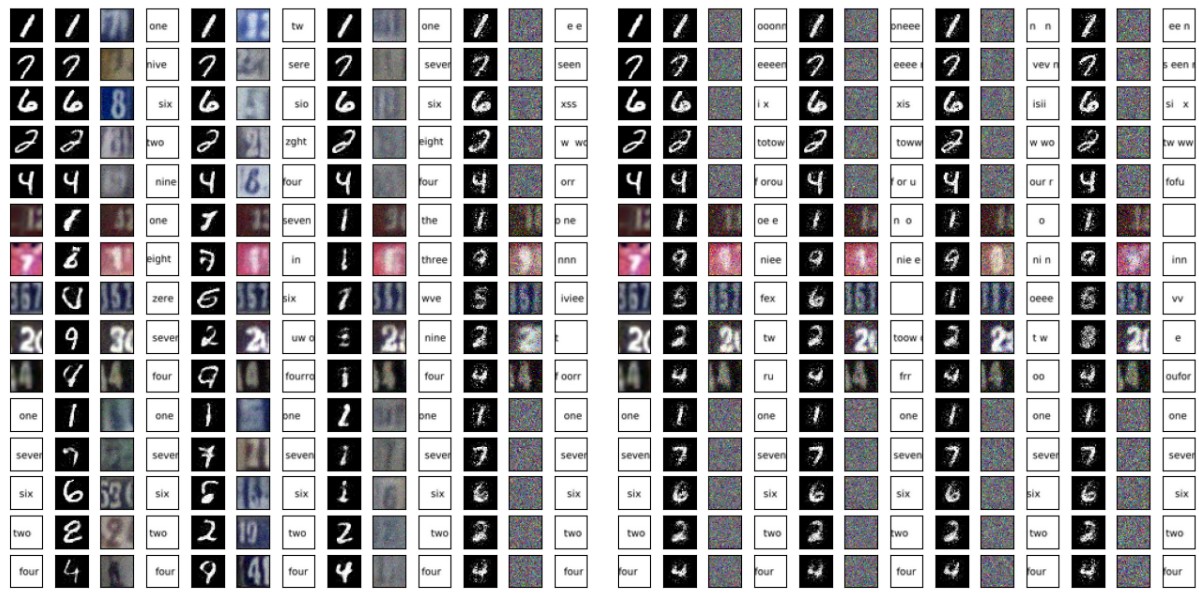

(a) Proposed objective          (b) Mixture-based bound

Figure 4: Conditional generation for different aggregation schemes and bounds and shared latent variables. The first column is the conditioned modality. The next three columns are the generated modalities using a SumPooling aggregation, followed by the three columns for a SelfAttention aggregation, followed by PoE+, and lastly MoE+.

## 5.4 Summary of experimental results

We presented a series of numerical experiments that illustrate the benefits of learning more flexible aggregation models and that optimizing our variational objective leads to higher log-likelihood values. Overall, we find that for a given choice of aggregation scheme, our objective achieves a higher log-likelihood across the different experiments. Likewise, fixing the variational objective, we observe that Sum-Pooling or Self-Attention encoders achieve higher multi-modal log-likelihoods compared to MoE or PoE schemes. Moreover, we demonstrate that our variational objective results in models that differ in their information theoretic quantities compared to those models trained with a mixture-based bound. In particular, our variational objective achieves higher full-reconstruction terms with higher full rates across different data sets, aggregation schemes, and beta values. Conversely, the mixture-based bound improves the cross-prediction while having higher cross-rate terms.

Table 6: Conditional coherence with shared latent variables and uni-modal inputs. The letters on the second line represent the generated modality based on the input modalities on the line below it.

| | Proposed objective | | | | | | | | | Mixture bound | | | | | | | | |
| | M | | | S | | | T | | | M | | | S | | | T | | |
| Aggregation | M | S | T | M | S | T | M | S | T | M | S | T | M | S | T | M | S | T |
| PoE | **0.97** | 0.22 | 0.56 | **0.29** | 0.60 | 0.36 | 0.78 | 0.43 | **1.00** | 0.96 | 0.83 | **0.99** | 0.11 | 0.57 | 0.10 | 0.44 | 0.39 | **1.00** |
| PoE+ | **0.97** | 0.15 | 0.63 | 0.24 | 0.63 | **0.42** | **0.79** | 0.35 | **1.00** | 0.96 | 0.83 | **0.99** | 0.11 | 0.59 | 0.11 | 0.45 | 0.39 | **1.00** |
| MoE | 0.96 | 0.80 | **0.99** | 0.11 | 0.59 | 0.11 | 0.44 | 0.37 | **1.00** | 0.94 | 0.81 | 0.97 | 0.10 | 0.54 | 0.10 | 0.45 | 0.39 | **1.00** |
| MoE+ | 0.93 | 0.77 | 0.95 | 0.11 | 0.54 | 0.10 | 0.44 | 0.37 | 0.98 | 0.94 | 0.80 | 0.98 | 0.10 | 0.53 | 0.10 | 0.45 | 0.39 | **1.00** |
| SumPooling | **0.97** | 0.48 | 0.87 | 0.25 | **0.72** | 0.36 | 0.73 | **0.48** | **1.00** | **0.97** | **0.86** | **0.99** | 0.10 | 0.63 | 0.10 | 0.45 | 0.40 | **1.00** |
| SelfAttention | **0.97** | 0.44 | 0.79 | 0.20 | 0.71 | 0.36 | 0.61 | 0.43 | **1.00** | **0.97** | **0.86** | **0.99** | 0.10 | 0.63 | 0.11 | 0.45 | 0.40 | **1.00** |
| | Results from Sutter et al. (2021), Sutter et al. (2020) and Hwang et al. (2021) | | | | | | | | | | | | | | | | | |
| MVAE | NA | 0.24 | 0.20 | 0.43 | NA | 0.30 | 0.28 | 0.17 | NA | | | | | | | | | |
| MMVAE | NA | 0.75 | **0.99** | 0.31 | NA | 0.30 | 0.96 | 0.76 | NA | | | | | | | | | |
| MoPoE | NA | 0.74 | **0.99** | 0.36 | NA | 0.34 | 0.96 | 0.76 | NA | | | | | | | | | |
| MMJSD | NA | **0.82** | **0.99** | 0.37 | NA | **0.36** | 0.97 | **0.83** | NA | | | | | | | | | |
| MVTCAE (w/o T) | NA | 0.60 | NA | **0.82** | NA | NA | NA | NA | NA | | | | | | | | | |

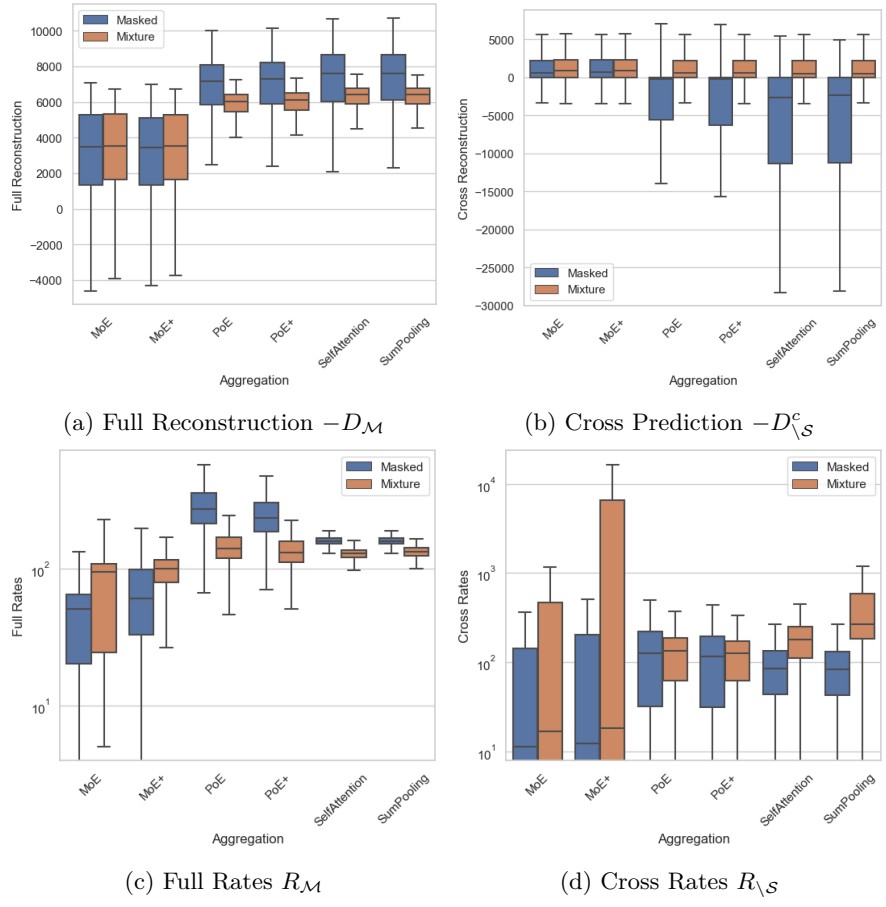

(a) Full Reconstruction $-D_{\mathcal{M}}$

(b) Cross Prediction $-D^c_{\backslash \mathcal{S}}$

(c) Full Rates $R_{\mathcal{M}}$

(d) Cross Rates $R_{\backslash \mathcal{S}}$

Figure 5: Rate and distortion terms for MNIST-SVHN-Text with shared latent variables ($\beta = 1$) for our proposed objective ('Masked') and the 'Mixture' based bound.

## 6 Conclusion

**Limitations.** A drawback of our bound is that computing a gradient step is more expensive as it requires drawing samples from two encoding distributions. Similarly, learning aggregation functions are more compu-

tationally expensive compared to fixed schemes. Mixture-based bounds might be preferred if one is interested primarily in cross-modal reconstructions.

**Outlook.** Using modality-specific encoders to learn features and aggregating them with a PI function is clearly not the only choice for building multi-modal encoding distributions. However, it allows us to utilize modality-specific architectures for the encoding functions. Alternatively, our bounds could also be used, e.g., when multi-modal transformer architectures (Xu et al., 2022) encode multiple modalities with modality-specific tokenization and embeddings onto a shared latent space. Our approach applies to general prior densities if we can compute its cross-entropy relative to the multi-modal encoding distributions. An example would be to apply it with more flexible prior distributions, e.g., as specified via score-based diffusion models (Vahdat et al., 2021). Likewise, diffusion models could be utilized to specify PI conditional prior distribution in the conditional bound by utilizing permutation-equivariant score models (Dutordoir et al., 2023; Yim et al., 2023; Mathieu et al., 2023).

### Acknowledgments

This work is supported by funding from the Wellcome Leap 1kD Program and by the RIE2025 Human Potential Programme Prenatal/Early Childhood Grant (H22P0M0002), administered by A*STAR. The computational work for this article was partially performed on resources of the National Supercomputing Centre, Singapore (`https://www.nscc.sg`).

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

# A   Multi-modal distribution matching

**Proof** [Proof of Proposition 1] The equations for $\mathcal{L}_{\mathcal{S}}(x_{\mathcal{S}})$ are well known for uni-modal VAEs, see for example Zhao et al. (2019). To derive similar representations for the conditional bound, note that the first equation ($\mathsf{ZX}_{\text{conditional}}$) for matching the joint distribution of the latent and the missing modalities conditional on a modality subset follows from the definition of $\mathcal{L}_{\backslash \mathcal{S}}$,

$$
\int p_d(x_{\backslash \mathcal{S}}|x_{\mathcal{S}})\mathcal{L}_{\backslash \mathcal{S}}(x,\theta,\phi)\mathrm{d}x_{\backslash \mathcal{S}}
$$
$$
= \int p_d(x_{\backslash \mathcal{S}}|x_{\mathcal{S}})\int q_\phi(z|x)\left[\log p_\theta(x_{\backslash \mathcal{S}}|z) - \log q_\phi(z|x) + \log q_\phi(z|x_{\mathcal{S}}))\right]\mathrm{d}z\mathrm{d}x_{\backslash \mathcal{S}}
$$
$$
= \int p_d(x_{\backslash \mathcal{S}}|x_{\mathcal{S}})\log p_d(x_{\backslash \mathcal{S}}|x_{\mathcal{S}})\mathrm{d}x_{\backslash \mathcal{S}} + \int p_d(x_{\backslash \mathcal{S}}|x_{\mathcal{S}})\int q_\phi(z|x)\left[\log \frac{p_\theta(x_{\backslash \mathcal{S}}|z)q_\phi(z|x_{\mathcal{S}}))}{q_\phi(z|x)p_d(x_{\backslash \mathcal{S}}|x_{\mathcal{S}})}\right]\mathrm{d}z\mathrm{d}x_{\backslash \mathcal{S}}
$$
$$
= -\mathcal{H}(p_d(x_{\backslash \mathcal{S}}|x_{\mathcal{S}})) - \mathsf{KL}\left(q_\phi(z|x)p_d(x_{\backslash \mathcal{S}}|x_{\mathcal{S}})\big|p_\theta(x_{\backslash \mathcal{S}}|z)q_\phi(z|x_{\mathcal{S}})\right).
$$

To obtain the second representation ($\mathsf{X}_{\text{conditional}}$) for matching the conditional distributions in the data space, observe that $p_\theta(x_{\backslash \mathcal{S}}|x_{\mathcal{S}},z) = p_\theta(x_{\backslash \mathcal{S}}|z)$ and consequently,

$$
-\int p_d(x_{\backslash \mathcal{S}}|x_{\mathcal{S}})\mathcal{L}_{\backslash \mathcal{S}}(x,\theta,\phi)\mathrm{d}x_{\backslash \mathcal{S}} - \mathcal{H}(p_d(x_{\backslash \mathcal{S}}|x_{\mathcal{S}}))
$$
$$
= \int p_d(x_{\backslash \mathcal{S}}|x_{\mathcal{S}})q_\phi(z|x)\log\frac{p_d(x_{\backslash \mathcal{S}}|x_{\mathcal{S}})q_\phi(z|x)}{p_\theta(x_{\backslash \mathcal{S}}|z)q_\phi(z|x_{\mathcal{S}})}\mathrm{d}z\mathrm{d}x_{\backslash \mathcal{S}}
$$
$$
= \int p_d(x_{\backslash \mathcal{S}}|x_{\mathcal{S}})q_\phi(z|x)\log\frac{p_d(x_{\backslash \mathcal{S}}|x_{\mathcal{S}})q_\phi(z|x)p_\theta(z|x_{\mathcal{S}})}{p_\theta(x_{\backslash \mathcal{S}}|z)p_\theta(z|x_{\mathcal{S}})q_\phi(z|x_{\mathcal{S}})}\mathrm{d}z\mathrm{d}x_{\backslash \mathcal{S}}
$$
$$
= \int p_d(x_{\backslash \mathcal{S}}|x_{\mathcal{S}})q_\phi(z|x)\log\frac{p_d(x_{\backslash \mathcal{S}}|x_{\mathcal{S}})q_\phi(z|x)p_\theta(z|x_{\mathcal{S}})}{p_\theta(x_{\backslash \mathcal{S}}|z,x_{\mathcal{S}})p_\theta(z|x_{\mathcal{S}})q_\phi(z|x_{\mathcal{S}})}\mathrm{d}z\mathrm{d}x_{\backslash \mathcal{S}}
$$
$$
= \int p_d(x_{\backslash \mathcal{S}}|x_{\mathcal{S}})q_\phi(z|x)\log\frac{p_d(x_{\backslash \mathcal{S}}|x_{\mathcal{S}})q_\phi(z|x)p_\theta(z|x_{\mathcal{S}})}{p_\theta(x_{\backslash \mathcal{S}}|x_{\mathcal{S}})p_\theta(z|x_{\mathcal{S}},x_{\backslash \mathcal{S}})q_\phi(z|x_{\mathcal{S}})}\mathrm{d}z\mathrm{d}x_{\backslash \mathcal{S}}
$$
$$
= \mathsf{KL}(p_d(x_{\backslash \mathcal{S}}|x_{\mathcal{S}})|p_\theta(x_{\backslash \mathcal{S}}|x_{\mathcal{S}})) + \int p_d(x_{\backslash \mathcal{S}}|x_{\mathcal{S}})\int q_\phi(z|x)\left[\log\frac{q_\phi(z|x)}{p_\theta(z|x)} + \log\frac{p_\theta(z|x_{\mathcal{S}})}{q_\phi(z|x_{\mathcal{S}})}\right]\mathrm{d}z\mathrm{d}x_{\backslash \mathcal{S}}.
$$

Lastly, the representation ($\mathsf{Z}_{\text{conditional}}$) for matching the distributions in the latent space given a modality subset follows by recalling that

$$
p_d(x_{\backslash \mathcal{S}}|x_{\mathcal{S}})q_\phi(z|x) = q_{\phi,\backslash \mathcal{S}}^{\text{agg}}(z|x_{\mathcal{S}})q^\star(x_{\backslash \mathcal{S}}|z,x_{\mathcal{S}})
$$

and consequently,

$$
-\int p_d(x_{\backslash \mathcal{S}}|x_{\mathcal{S}})\mathcal{L}_{\backslash \mathcal{S}}(x,\theta,\phi)\mathrm{d}x_{\backslash \mathcal{S}} - \mathcal{H}(p_d(x_{\backslash \mathcal{S}}|x_{\mathcal{S}}))
$$
$$
= \int p_d(x_{\backslash \mathcal{S}}|x_{\mathcal{S}})q_\phi(z|x)\log\frac{p_d(x_{\backslash \mathcal{S}}|x_{\mathcal{S}})q_\phi(z|x)}{p_\theta(x_{\backslash \mathcal{S}}|z)q_\phi(z|x_{\mathcal{S}})}\mathrm{d}z\mathrm{d}x_{\backslash \mathcal{S}}
$$
$$
= \int q_{\phi,\backslash \mathcal{S}}^{\text{agg}}(z|x_{\mathcal{S}})q^\star(x_{\backslash \mathcal{S}}|z,x_{\mathcal{S}})\log\frac{q_{\phi,\backslash \mathcal{S}}^{\text{agg}}(z|x_{\mathcal{S}})q^\star(x_{\backslash \mathcal{S}}|z,x_{\mathcal{S}})}{p_\theta(x_{\backslash \mathcal{S}}|z)q_\phi(z|x_{\mathcal{S}})}\mathrm{d}z\mathrm{d}x_{\backslash \mathcal{S}}
$$
$$
= \mathsf{KL}(q_{\phi,\backslash \mathcal{S}}^{\text{agg}}(z|x_{\mathcal{S}})|q_\phi(z|x_{\mathcal{S}})) - \int q_{\phi,\backslash \mathcal{S}}^{\text{agg}}(z|x_{\mathcal{S}})\left(\mathsf{KL}(q^\star(x_{\backslash \mathcal{S}}|z,x_{\mathcal{S}})|p_\theta(x_{\backslash \mathcal{S}}|z))\right)\mathrm{d}z.
$$

$\blacksquare$

# B  Meta-learning and Neural processes

**Meta-learning.**  We consider a standard meta-learning setup but use slightly non-standard notations to remain consistent with notations used in other parts of this work. We consider a compact input or covariate space $\mathcal{A}$ and output space $\mathcal{X}$. Let $\mathcal{D} = \cup_{M=1}^{\infty} (\mathcal{A} \times \mathcal{X})^M$ be the collection of all input-output pairs. In meta-learning, we are given a meta-dataset, i.e., a collection of elements from $\mathcal{D}$. Each individual data set $D = (a, x) = D_c \cup D_t \in \mathcal{D}$ is called a task and split into a context set $D_c = (a_c, x_c)$, and target set $D_t = (a_t, x_t)$. We aim to predict the target set from the context set. Consider, therefore, the prediction map

$$\pi \colon D_c = (a_c, x_c) \mapsto p(x_t | a_t, D_c) = p(x_t, x_c | a_t, a_c)/p(x_c | a_c),$$

mapping each context data set to the predictive stochastic process conditioned on $D_c$.

**Variational lower bounds for Neural processes.**  Latent Neural processes (Garnelo et al., 2018b; Foong et al., 2020) approximate this prediction map by using a latent variable model with parameters $\theta$ in the form of

$$z \sim p_\theta, \ p_\theta(x_t | a_t, z) = \prod_{(a,x) \in D_t} p_\epsilon(x - f_\theta(a, z))$$

for a prior $p_\theta$, decoder $f_\theta$ and a parameter free density $p_\epsilon$. The model is then trained by (approximately) maximizing a lower bound on $\log p_\theta(x_t | a_t, a_c, x_c)$. Note that for an encoding density $q_\phi$, we have that

$$\log p_\theta(x_t | a_t, a_c, x_x) = \int q_\phi(z | x, a) \log p_\theta(x_t | a_t, z) \mathrm{d}z - \mathsf{KL}(q_\phi(z | a, x) | p_\theta(z | a_c, x_c)).$$

Since the posterior distribution $p_\theta(z | a_c, x_c)$ is generally intractable, one instead replaces it with a variational approximation or learned conditional prior $q_\phi(z | a_c, x_c)$, and optimizes the following objective

$$\mathcal{L}_{\backslash \mathcal{C}}^{\mathrm{LNP}}(x, a) = \int q_\phi(z | x, a) \log p_\theta(x_t | a_t, z) \mathrm{d}z - \mathsf{KL}(q_\phi(z | a, x) | q_\phi(z | a_c, x_c)).$$

Note that this objective coincides with $\mathcal{L}_{\backslash \mathcal{C}}$ conditioned on the covariate values $a$ and where $\mathcal{C}$ comprises the indices of the data points that are part of the context set. Using the variational lower bound $\mathcal{L}_{\backslash \mathcal{C}}^{\mathrm{LNP}}$ can yield subpar performance compared to another biased log-likelihood objective (Kim et al., 2018; Foong et al., 2020),

$$\log \hat{p}_\theta(x_t | a_t, a_c, x_c) = \log \left[ \frac{1}{L} \sum_{l=1}^{L} \exp \left( \sum_{(x_t, a_t) \in D_t} \log p_\theta(x_t | a_t, z_c^l) \right) \right]$$

for $L$ importance samples $z_c^l \sim q_\phi(z_c | x_c, a_c)$ drawn from the conditional prior as the proposal distribution. The required number of importance samples $L$ for accurate estimation scales exponentially in the forward $\mathsf{KL}(q_\phi(z | x, a) | q_\phi(z | x_c, a_c))$, see Chatterjee et al. (2018). Unlike a variational approach, such an estimator does not enforce a Bayes-consistency term for the encoders and may be beneficial in the setting of finite data and model capacity. Note that the Bayes consistency term for including the target set $(x_t, a_t)$ into the context set $(x_c, a_c)$ writes as

$$\mathsf{KL}(q_{\phi, \backslash \mathcal{C}}^{\mathrm{agg}}(z | x_c, a_c) | q_\phi(z | x_c, a_c)) = \mathsf{KL} \left( \int p_d(x_t | a_t, x_c, a_c) q_\phi(z | x, a) \mathrm{d}x_t \middle| q_\phi(z | x_c, a_c) \right).$$

Moreover, if one wants to optimize not only the conditional but also the marginal distributions, one may additionally optimize the variational objective corresponding to $\mathcal{L}_\mathcal{C}$, i.e.,

$$\mathcal{L}_\mathcal{C}^{\mathrm{LNP}}(x_c, a_c) = \int q_\phi(z | x_c, a_c) \log p_\theta(x_c | a_c, z) \mathrm{d}z - \mathsf{KL}(q_\phi(z | a_c, x_c) | p_\theta(z)),$$

as we do in this work for multi-modal generative models. Note that the objective $\mathcal{L}_\mathcal{C}^{\mathrm{LNP}}$ alone can be seen as a form of a Neural Statistician model (Edwards and Storkey, 2016) where $\mathcal{C}$ coincides with the indices of

the target set, while a form of the mixture-based bound corresponds to a Neural process bound similar to variational Homoencoders (Hewitt et al., 2018), see also the discussion in Le et al. (2018). The multi-view variational information bottleneck approach developed in Lee and van der Schaar (2021) for predicting $X_{\setminus S}$ given $X_S$ involves the joint variational objective

$$\mathcal{L}_S^{\text{IB}}(x, \theta, \phi, \beta) = \int q_\phi(z|x_S) \log p_\theta(x_{\setminus S}|z) \mathrm{d}z - \beta \mathsf{KL}(q_\phi(z|x_S)|p_\theta(z))$$

which can be interpreted as maximizing $\hat{\mathrm{I}}_{q_\phi}^{\text{lb}}(X_{\setminus S}, Z_S) - \beta \hat{\mathrm{I}}_{q_\phi}^{\text{ub}}(X_S, Z_S)$ and corresponds to the variational information bottleneck for meta-learning in Titsias et al. (2021).

## C Information-theoretic perspective

We recall first that the mutual information is given by

$$\mathrm{I}_{q_\phi}(X_S, Z_S) = \int q_\phi(x_S, z_S) \log \frac{q_\phi(x_S, z_S)}{p_d(x_S) q_{\phi,S}^{\text{agg}}(z_S)} \mathrm{d}z_S \mathrm{d}x_S,$$

where $q_{\phi,S}^{\text{agg}}(z) = \int p_d(x_S) q_\phi(z|x_S) \mathrm{d}x_S$ is the aggregated prior (Makhzani et al., 2016). It can be bounded by standard (Barber and Agakov, 2004; Alemi et al., 2016; 2018) lower and upper bounds using the rate and distortion:

$$\mathcal{H}_S - D_S \leq \mathcal{H}_S - D_S + \Delta_1 = \mathrm{I}_{q_\phi}(X_S, Z_S) = R_S - \Delta_2 \leq R_S, \tag{15}$$

with $\Delta_1 = \int q_\phi^{\text{agg}}(z) \mathsf{KL}(q^\star(x_S|z)|p_\theta(x_S|z)) \mathrm{d}z > 0$, $\Delta_2 = \mathsf{KL}(q_{\phi,S}^{\text{agg}}(z)|p_\theta(z)) > 0$ and $q^\star(x_S|z) = q_\phi(x_S, z)/q_\phi^{\text{agg}}(z)$.

Moreover, if the bounds in (7) become tight with $\Delta_1 = \Delta_2 = 0$ in the hypothetical scenario of infinite-capacity decoders and encoders, one obtains $\int p_d \mathcal{L}_S = (1 - \beta) \mathrm{I}_{q_\phi}(X_S, Z_S) + \mathcal{H}_S$. For $\beta > 1$, maximizing $\mathcal{L}_S$ yields an auto-decoding limit that minimizes $\mathrm{I}_{q_\phi}(x_S, z)$ for which the latent representations do not encode any information about the data, whilst $\beta < 1$ yields an auto-encoding limit that maximizes $\mathrm{I}_{q_\phi}(X_S, Z_S)$ and for which the data is perfectly encoded and decoded.

The mixture-based bound can be interpreted as the maximization of a variational lower bound of $I_{q_\phi}(X_\mathcal{M}, Z_S)$ and the minimization of a variational upper bound of $I_{q_\phi}(X_S, Z_S)$. Indeed, see also Daunhawer et al. (2022),

$$\mathcal{H}_\mathcal{M} - D_S - D_S^c \leq \mathcal{H}_\mathcal{M} - D_S - D_S^c + \Delta_1' = \mathrm{I}_{q_\phi}(X_\mathcal{M}, Z_S),$$

where $\Delta_1' = \int q_\phi^{\text{agg}}(z) \mathsf{KL}(q^\star(x|z)|p_\theta(x|z)) \mathrm{d}z > 0$, due to

$$I_{q_\phi}(X_\mathcal{M}, Z_S) = \mathcal{H}_\mathcal{M} - \mathcal{H}_{q_\phi}(X|Z_S) = \mathcal{H}_\mathcal{M} + \int p_d(x) q_\phi(z|x_S) \left[\log q^\star(x|z)\right] \mathrm{d}z \mathrm{d}x$$

$$= \mathcal{H}_\mathcal{M} + \int p_d(x) q_\phi(z|x_S) \left[\log p_\theta(x_S|z) + \log p_\theta(x_{\setminus S}|z) + \log \frac{q^\star(x|z)}{p_\theta(x|z)}\right] \mathrm{d}z \mathrm{d}x.$$

Recalling that

$$\int p_d(\mathrm{d}x) \mathcal{L}_S^{\text{Mix}}(x) = -D_S - D_{\setminus S}^c - \beta R_S,$$

one can see that maximizing the first part of the mixture-based variational bound corresponds to maximizing $-D_S - D_{\setminus S}^c$ as a variational lower bound of $I_{q_\phi}(X_\mathcal{M}, Z_S)$, when ignoring the fixed entropy of the multi-modal data. Maximizing the second part of the mixture-based variational bound corresponds to minimizing $R_S$ as a variational upper bound of $\mathrm{I}_{q_\phi}(X_S, Z_S)$, see (15).

**Proof** [Proof of Lemma 7] The proof follows by adapting the arguments in Alemi et al. (2018). The law of $X_{\setminus S}$ and $Z$ conditional on $X_S$ on the encoder path can be written as

$$q_\phi(z, x_{\setminus S}|x_S) = p_d(x_{\setminus S}|x_S) q_\phi(z|x) = q_{\phi,\setminus S}^{\text{agg}}(z|x_S) q^\star(x_{\setminus S}|z, x_S)$$

with $q^\star(x_{\backslash\mathcal{S}}|z, x_\mathcal{S}) = q_\phi(z, x_{\backslash\mathcal{S}}|x_\mathcal{S})/q_{\phi,\backslash\mathcal{S}}^{\mathrm{agg}}(z|x_\mathcal{S})$. To prove a lower bound on the conditional mutual information, note that

$$
\begin{aligned}
&\mathrm{I}_{q_\phi}(X_{\backslash\mathcal{S}}, Z_\mathcal{M}|X_\mathcal{S}) \\
&= \int p_d(x_\mathcal{S}) \int q_{\phi,\backslash\mathcal{S}}^{\mathrm{agg}}(z|x_\mathcal{S}) \int q^\star(x_{\backslash\mathcal{S}}|z, x_\mathcal{S}) \log \frac{q_{\phi,\backslash\mathcal{S}}^{\mathrm{agg}}(z|x_\mathcal{S}) q^\star(x_{\backslash\mathcal{S}}|z, x_\mathcal{S})}{q_{\phi,\backslash\mathcal{S}}^{\mathrm{agg}}(z|x_\mathcal{S}) p_d(x_{\backslash\mathcal{S}}|x_{\backslash\mathcal{S}})} \mathrm{d}z \mathrm{d}x_{\backslash\mathcal{S}} \mathrm{d}x_\mathcal{S} \\
&= \int p_d(x_\mathcal{S}) \int q_{\phi,\backslash\mathcal{S}}^{\mathrm{agg}}(z|x_\mathcal{S}) \left[ \int q^\star(x_{\backslash\mathcal{S}}|z, x_\mathcal{S}) \log p_\theta(x_{\backslash\mathcal{S}}|z)) \mathrm{d}x_{\backslash\mathcal{S}} + \mathsf{KL}(q^\star(x_{\backslash\mathcal{S}}|z, x_\mathcal{S})|p_\theta(x_{\backslash\mathcal{S}}|z)) \right] \mathrm{d}z \mathrm{d}x_\mathcal{S} \\
&\quad - \int p_d(x_\mathcal{S}) \int p_d(x_{\backslash\mathcal{S}}|x_\mathcal{S}) \log p_d(x_{\backslash\mathcal{S}}|x_\mathcal{S}) \mathrm{d}x \\
&= \int p_d(x) \int q_\phi(z|x) \log p_\theta(x_{\backslash\mathcal{S}}|z) \mathrm{d}z \mathrm{d}x \underbrace{- \int p_d(x_\mathcal{S}) \int p_d(x_{\backslash\mathcal{S}}|x_\mathcal{S}) \log p_d(x_{\backslash\mathcal{S}}|x_\mathcal{S}) \mathrm{d}x}_{=-\mathcal{H}_{\backslash\mathcal{S}} = -\mathcal{H}(X_{\backslash\mathcal{S}}|X_\mathcal{S})} \\
&\quad + \underbrace{\int p_d(x_\mathcal{S}) \int q_{\phi,\backslash\mathcal{S}}^{\mathrm{agg}}(z|x_\mathcal{S}) \mathsf{KL}(q^\star(x_{\backslash\mathcal{S}}|z, x_\mathcal{S})|p_\theta(x_{\backslash\mathcal{S}}|z)) \mathrm{d}x_\mathcal{S}}_{=\Delta_{\backslash\mathcal{S},1} \geq 0} \\
&= \Delta_{\backslash\mathcal{S},1} + D_{\backslash\mathcal{S}} + \mathcal{H}_{\backslash\mathcal{S}}.
\end{aligned}
$$

The upper bound follows by observing that

$$
\begin{aligned}
&\mathrm{I}_{q_\phi}(X_{\backslash\mathcal{S}}, Z_\mathcal{M}|X_\mathcal{S}) \\
&= \int p_d(x_\mathcal{S}) \int p_d(x_{\backslash\mathcal{S}}|x_\mathcal{S}) q_\phi(z|x) \log \frac{q_\phi(z|x) p_d(x_{\backslash\mathcal{S}}|x_\mathcal{S})}{q_{\phi,\backslash\mathcal{S}}^{\mathrm{agg}}(z|x_\mathcal{S}) p_d(x_{\backslash\mathcal{S}}|x_\mathcal{S})} \mathrm{d}z \mathrm{d}x \\
&= \int p_d(x) \mathsf{KL}(q_\phi(z|x)|q_\phi(z|x_\mathcal{S})) \mathrm{d}x \underbrace{- \int p_d(x_\mathcal{S}) \mathsf{KL}(q_{\phi,\backslash\mathcal{S}}^{\mathrm{agg}}(z|x_\mathcal{S})|q_\phi(z|x_\mathcal{S})) \mathrm{d}x_\mathcal{S}}_{=\Delta_{\backslash\mathcal{S},2} \geq 0} \\
&= R_{\backslash\mathcal{S}} - \Delta_{\backslash\mathcal{S},2}.
\end{aligned}
$$

∎

**Remark 17 (Total correlation based objectives)** The objective suggested in Hwang et al. (2021) is motivated by a conditional variational bottleneck perspective that aims to maximize the reduction of total correlation of $X$ when conditioned on $Z$, as measured by the conditional total correlation, see Watanabe (1960); Ver Steeg and Galstyan (2015); Gao et al. (2019), i.e.,

$$
\text{minimizing} \left\{ \mathrm{TC}(X|Z) = \mathrm{TC}(X) - \mathrm{TC}(X, Z) = \mathrm{TC}(X) + \mathrm{I}_{q_\phi}(X, Z) - \sum_{s=1}^M \mathrm{I}_{q_\phi}(X_s, Z) \right\}, \tag{16}
$$

where $\mathrm{TC}(X) = \mathsf{KL}(p(x)| \prod_{i=1}^d p(x_i))$ for $d$-dimensional $X$. Resorting to variational lower bounds and using a constant $\beta > 0$ that weights the contributions of the mutual information terms, approximations of (16) can be optimized by maximizing

$$
\mathcal{L}^{\mathrm{TC}}(\theta, \phi, \beta) = \int \rho(\mathcal{S}) \int \left\{ q_\phi(z|x) \left[ \log p_\theta(x|z) \right] \mathrm{d}z - \beta \mathsf{KL}(q_\phi(z|x)|q_\phi(z|x_\mathcal{S})) \right\} \mathrm{d}\mathcal{S},
$$

where $\rho$ is concentrated on the uni-modal subsets of $\mathcal{M}$.

**Remark 18 (Entropy regularised optimization)** Let $q$ be a density over $\mathsf{C}$, $\exp(g)$ be integrable with respect to $q$ and $\tau > 0$. The maximum of

$$
f(q) = \int_\mathsf{C} q(c) \left[ g(c) - \tau \log q(c) \right] \mathrm{d}c
$$

that is attained at $q^\star(c) = \frac{1}{\mathcal{Z}}\, e^{g(c)/\tau}$ with normalizing constant $\mathcal{Z} = \int_{\mathsf{C}} e^{g(c)/\tau}\, dc$ is

$$f^\star = f(q^\star) = \tau \log \int_{\mathsf{C}} e^{g(c)/\tau}\, dc.$$

**Remark 19 (Optimal variational distribution)** The optimal variational density for the mixture-based (1) multi-modal objective,

$$\int p_d(dx)\mathcal{L}_{\mathcal{S}}^{\mathrm{Mix}}(x) = \int p_d(x_{\mathcal{S}}) \int q_\phi(z|x_{\mathcal{S}}) \int p_d(x_{\backslash\mathcal{S}}|x_{\mathcal{S}})$$
$$\left[\log p_\theta(x_{\mathcal{S}}|z) + \log p_\theta(x_{\backslash\mathcal{S}}|z) + \beta \log p_\theta(z) - \beta \log q_\phi(z|x_{\mathcal{S}})\right] dx_{\backslash\mathcal{S}} dz dx_{\mathcal{S}},$$

using Remark 18, is attained at

$$q^\star(z|x_{\mathcal{S}}) \propto \exp\left(\frac{1}{\beta} \int p_d(x_{\backslash\mathcal{S}}|x_{\mathcal{S}}) \left[\log p_\theta(x_{\mathcal{S}}|z) + \log p_\theta(x_{\backslash\mathcal{S}}|z) + \beta \log p_\theta(z)\right] dx_{\backslash\mathcal{S}}\right)$$

$$\propto \tilde{p}_{\beta,\theta}(z|x_{\mathcal{S}}) \exp\left(\int p_d(x_{\backslash\mathcal{S}}|x_{\mathcal{S}}) \log \tilde{p}_{\beta,\theta}(x_{\backslash\mathcal{S}}|z) dx_{\backslash\mathcal{S}}\right).$$

## D   Permutation-invariant architectures

**Multi-head attention and masking.**   We introduce here a standard multi-head attention (Bahdanau et al., 2014; Vaswani et al., 2017) mapping $\mathrm{MHA}_\vartheta \colon \mathbb{R}^{I \times D_X} \times \mathbb{R}^{S \times D_Y} \to \mathbb{R}^{I \times D_Y}$ given by

$$\mathrm{MHA}_\vartheta(X, Y) = W^O \left[\mathrm{Head}^1(X, Y, Y), \ldots, \mathrm{Head}^H(X, Y, Y)\right], \quad \vartheta = (W_Q, W_K, W_V, W_O),$$

with output matrix $W_O \in \mathbb{R}^{D_A \times D_Y}$, projection matrices $W_Q \in \mathbb{R}^{D_X \times D_A}$ $W_K, W_V \in \mathbb{R}^{D_Y \times D_A}$ and

$$\mathrm{Head}^h(Q, K, V) = \mathrm{Att}(QW_Q^h, KW_K^h, VW_V^h) \in \mathbb{R}^{I \times D} \tag{17}$$

where we assume that $D = D_A/H \in \mathbb{N}$ is the head size. Here, the dot-product attention function is

$$\mathrm{Att}(Q, K, V) = \sigma(QK^\top)V,$$

where $\sigma$ is the softmax function applied to each column of $QK^\top$.

**Masked multi-head attention.**   In practice, it is convenient to consider masked multi-head attention models $\mathrm{MMHA}_{\vartheta,M} \colon \mathbb{R}^{I \times D_X} \times \mathbb{R}^{T \times D_Y} \to \mathbb{R}^{I \times D_Y}$ for mask matrix $M \in \{0,1\}^{I \times T}$ that operate on key or value sequences of fixed length $T$ where the $h$-th head (17) is given by

$$\mathrm{Head}^h(Q, K, V) = \left[M \odot \sigma(QW_Q^h(KW_K^h)^\top)\right] V_{t'} W_V^h \in \mathbb{R}^{T \times D}.$$

Using the softmax kernel function $\mathrm{SM}_D(q, k) = \exp(q^\top k/\sqrt{D})$, we set

$$\mathrm{MMHA}_{\vartheta,M}(X, Y)_i = \sum_{t=1}^{T} \sum_{h=1}^{H} \frac{M_{it}\mathrm{SM}_D(W_h^Q X_i, W_h^K Y_t)}{\sum_{t'=1}^{T} M_{it'}\mathrm{SM}_D(X_i W_h^Q, Y_{t'} W_h^K)} Y_t W_h^V W_h^O \tag{18}$$

which does not depend on $Y_t$ if $M_{\cdot t} = 0$.

**Masked self-attention.**   For mask matrix $M = mm^\top$ with $m = (1_{\{s \in \mathcal{S}\}})_{s \in \mathcal{M}}$, we write

$$\mathrm{MHA}_\vartheta(Y_{\mathcal{S}}, Y_{\mathcal{S}}) = \mathrm{MMHA}_{\vartheta,M}(\mathfrak{i}(Y_{\mathcal{S}}), \mathfrak{i}(Y_{\mathcal{S}}))_{\mathcal{S}}.$$

where $\mathrm{MMHA}_{\vartheta,M}$ operates on sequences with fixed length and $\mathfrak{i}(Y_{\mathcal{S}}))_t = Y_t$ if $t \in \mathcal{S}$ and 0 otherwise.

**LayerNorm and SetNorm.** Let $h \in \mathbb{R}^{T \times D}$ and consider the normalization

$$\mathrm{N}(h) = \frac{h - \mu(h)}{\sigma(h)} \odot \gamma + \beta$$

where $\mu$ and $\sigma$ standardize the input $h$ by computing the mean, and the variance, respectively, over some axis of $h$, whilst $\gamma$ and $\beta$ define a transformation. LayerNorm (Ba et al., 2016) standardises inputs over the last axis, e.g., $\mu(h) = \frac{1}{D} \sum_{d=1}^{D} \mu_{\cdot,d}$, i.e., separately for each element. In contrast, SetNorm (Zhang et al., 2022b) standardises inputs over both axes, e.g., $\mu(h) = \frac{1}{TD} \sum_{t=1}^{T} \sum_{d=1}^{D} \mu_{t,d}$, thereby losing the global mean and variance only. In both cases, $\gamma$ and $\beta$ share their values across the first axis. Both normalizations are permutation-equivariant.

**Transformer.** We consider a masked pre-layer-norm (Wang et al., 2019a; Xiong et al., 2020) multi-head transformer block

$$(\mathrm{MMTB}_{\vartheta,M}(\mathfrak{i}_{\mathcal{S}}(Y_{\mathcal{S}})))_{\mathcal{S}} = (Z + \sigma_{\mathrm{ReLU}}(\mathrm{LN}(Z)))_{\mathcal{S}}$$

with $\sigma_{\mathrm{ReLU}}$ being a ReLU non-linearity and

$$Z = \mathfrak{i}_{\mathcal{S}}(Y_{\mathcal{S}}) + \mathrm{MMHA}_{\vartheta,M}(\mathrm{LN}(\mathfrak{i}_{\mathcal{S}}(Y_{\mathcal{S}})), \mathrm{LN}(\mathfrak{i}_{\mathcal{S}}(Y_{\mathcal{S}})))$$

where $M = mm^\top$ for $m = (1_{\{s \in \mathcal{S}\}})_{s \in \mathcal{M}}$.

**Set-Attention Encoders.** Set $g^0 = \mathfrak{i}_{\mathcal{S}}(\chi_\vartheta(h_{\mathcal{S}}))$ and for $k \in \{1, \ldots, L\}$, let

$$g^k = \mathrm{MMTB}_{\vartheta,M}(g_{\mathcal{S}}^{k-1}).$$

Then, we can express the self-attention multi-modal aggregation mapping via $f_\vartheta(h_{\mathcal{S}}) = \rho_\vartheta \left( \sum_{s \in \mathcal{S}} g_s^L \right)$.

**Remark 20 (Context-aware pooling)** Assuming a single head for the transformer encoder in Example 2 with head size $D$ and projection matrices $W_Q, W_K, W_V \in \mathbb{R}^{D_P \times D}$, the attention scores for the initial input sequence $g_{\mathcal{S}} = g_{\mathcal{S}}^0 = \chi_\vartheta(h_{\mathcal{S}}) \in \mathbb{R}^{|\mathcal{S}| \times D_P}$ are $a(g_s, g_t) = \langle W_Q^\top g_s, W_K^\top g_t \rangle / \sqrt{D}$. The attention outputs $o_s \in \mathbb{R}^D$ for $s \in \mathcal{S}$ can then be written as

$$o_s = \frac{1}{Z} \sum_{t \in \mathcal{S}} \kappa(g_s, g_t) v(g_t),$$

where $Z = \sum_{t \in \mathcal{S}} \kappa(g_s, g_t) > 0$, $v(g_t) = W_V^\top g_t$ and

$$\kappa(g_s, g_t) = \exp(a(g_s, g_t)) = \exp\left( \langle W_Q^\top g_s, W_K^\top g_t \rangle / \sqrt{D} \right)$$

can be seen as a learnable non-symmetric kernel (Wright and Gonzalez, 2021; Cao, 2021). Conceptually, the attention encoder pools a learnable $D$-dimensional function $v$ using a learnable context-dependent weighting function. While such attention models directly account for the interaction between the different encodings, a DeepSet aggregation approach may require a sufficiently high-dimensional latent space $D_P$ to achieve universal approximation properties (Wagstaff et al., 2022).

**Remark 21 (Multi-modal time series models)** We have introduced a multi-modal generative model in a general form that also applies to the time-series setup, such as when a latent Markov process drives multiple time series. For example, consider a latent Markov process $Z = (Z_t)_{t \in \mathbb{N}}$ with prior dynamics $p_\theta(z_1, \ldots, z_T) = p_\theta(z_1) \prod_{t=2}^{T} p_\theta(z_t|z_{t-1})$ for an initial density $p_\theta(z_1)$ and homogeneous Markov kernels $p_\theta(z_t|z_{t-1})$. Conditional on $Z$, suppose that the time-series $(X_{s,t})_{t \in \mathbb{N}}$ follows the dynamics $p_\theta(x_{s,1}, \ldots, x_{s,T}|z_1, \ldots, z_T) = \prod_{t=2}^{T} p_\theta(x_{s,t}|z_t)$ for decoding densities $p_\theta(x_{s,t}|z_t)$. A common choice (Chung et al., 2015) for modeling the encoding distribution for such sequential (uni-modal) VAEs is to assume the factorization $q_\phi(z_1, \ldots z_T | x_1, \ldots x_T) = q_\phi(z_1|x_1) \prod_{t=2}^{T} q_\phi(z_t|z_{t-1}, x_t)$ for $x_t = (x_{s,t})_{s \in \mathcal{M}}$, with initial encoding densities $q_\phi(z_1|x_1)$ and encoding Markov kernels $q_\phi(z_t|z_{t-1}, x_t)$. One can again consider modality-specific encodings $h_s = (h_{s,1}, \ldots, h_{s,T})$, $h_{s,t} = h_{s,\varphi}(x_{s,t})$, now applied separately at each time step that are then used to construct Markov kernels that are permutation-invariant in the form of $q_\phi'(z_t|z_{t-1}, \pi h_\varphi(x_{t,\mathcal{S}})) =$

$q'_\phi(z_t|z_{t-1}, h_\varphi(x_{t,\mathcal{S}}))$ for permutations $\pi \in \mathbb{S}_\mathcal{S}$. Alternatively, in the absence of the auto-regressive encoding structure with Markov kernels, one could also use transformer models that use absolute or relative positional embeddings across the last temporal axis but no positional embeddings across the first modality axis, followed by a sum-pooling operation across the modality axis. Note that previous works using multi-modal time series such as Kramer et al. (2022) use a non-amortized encoding distribution for the full multi-modal posterior only. A numerical evaluation of permutation-invariant schemes for time series models is, however, outside the scope of this work.

**Remark 22 (Alternative multi-modal encoding models)** Learning different encoders for each modality subset can be an alternative when the number of modalities is small. For example, one could learn different MLP heads for each modality subset $\mathcal{S}$ that aggregates the encoded features $h_\mathcal{S}$ from modality-specific encoders that are shared for any modality mask. Our initial experiments with such encoders for the simulated linear modalities in Section 5.1 for $M = 5$ modalities did not improve on the permutation-invariant models, while also being more computationally demanding.

## E  Permutation-equivariance and private latent variables

**Remark 23 (Variational bounds with private latent variables)** To compute the multi-modal variational bounds, notice that the required KL divergences can be written as follows:

$$\mathsf{KL}(q_\phi(z', \tilde{z}|x_\mathcal{S})|p_\theta(z', \tilde{z})) = \mathsf{KL}(q_\phi(z'|x_\mathcal{S})|p_\theta(z')) + \int q_\phi(z'|x_\mathcal{S})\mathsf{KL}(q_\phi(\tilde{z}_\mathcal{S}|z', x_\mathcal{S})|p_\theta(\tilde{z}_\mathcal{S}|z'))\mathrm{d}z'$$

and

$$
\begin{aligned}
&\mathsf{KL}(q_\phi(z', \tilde{z}|x_\mathcal{M})|q_\phi(z', \tilde{z}|x_\mathcal{S})) \\
=&\mathsf{KL}(q_\phi(z'|x_\mathcal{M})|(q_\phi(z'|x_\mathcal{S})) + \int q_\phi(z'|x_\mathcal{M})\mathsf{KL}(q_\phi(\mathsf{P}_\mathcal{S}\tilde{z}|z', x_\mathcal{M})|q_\phi(\mathsf{P}_\mathcal{S}\tilde{z}|z', x_\mathcal{S}))\mathrm{d}z' \\
&+ \int q_\phi(z'|x_\mathcal{M})\mathsf{KL}(q_\phi(\mathsf{P}_{\backslash\mathcal{S}}\tilde{z}|z', x_\mathcal{S})|p_\theta(\mathsf{P}_{\backslash\mathcal{S}}\tilde{z}|z'))\mathrm{d}z'
\end{aligned}
$$

where $\mathsf{P}_\mathcal{S}\colon (\tilde{z}_1, \ldots \tilde{z}_M) \mapsto (\tilde{z}_s)_{s\in\mathcal{S}}$ projects all private latent variables to those contained in $\mathcal{S}$.

These expressions can be used to compute our overall variational bound $\mathcal{L}_\mathcal{S} + \mathcal{L}_{\backslash\mathcal{S}}$ via

$$
\begin{aligned}
&\int q_\phi(z'|x_\mathcal{S})q_\phi(\tilde{z}_\mathcal{S}|z', x_\mathcal{S})]\log p_\theta(x_\mathcal{S}|z', \tilde{z}_\mathcal{S})\mathrm{d}z'\mathrm{d}\tilde{z}_\mathcal{S} \\
&- \mathsf{KL}\Big(q_\phi(z'|x_\mathcal{S})q_\phi(\tilde{z}_\mathcal{S}|z', x_\mathcal{S})\Big|p_\theta(z')p_\theta(\tilde{z}_\mathcal{S}|z')\Big) \\
&+ \int q_\phi(z'|x_\mathcal{M})q_\phi(\tilde{z}_{\backslash\mathcal{S}}|z', x_\mathcal{M})]\log p_\theta(x_\mathcal{S}|z', \tilde{z}_{\backslash\mathcal{S}})\mathrm{d}z'\mathrm{d}\tilde{z}_\mathcal{S} \\
&- \mathsf{KL}\Big(q_\phi(z', \tilde{z}_\mathcal{S}, \tilde{z}_{\backslash\mathcal{S}}|x_\mathcal{M})\Big|q_\phi(z', \tilde{z}_\mathcal{S}, \tilde{z}_{\backslash\mathcal{S}}|x_\mathcal{S})\Big).
\end{aligned}
$$

**Remark 24 (Comparison with MMVAE+ variational bound)** It is instructive to compare our bound with the MMVAE+ approach suggested in Palumbo et al. (2023). Assuming a uniform masking distribution restricted to uni-modal sets so that $\mathcal{S} = \{s\}$ for some $s \in \mathcal{M}$, we can write the bound from Palumbo et al. (2023) as $\frac{1}{M}\sum_{s=1}^{M}\mathcal{L}_{\{s\}}^{\mathrm{MMVAE+}}(x)$ with

$$
\begin{aligned}
\mathcal{L}_{\{s\}}^{\mathrm{MMVAE+}}(x) =& \int q_\phi(z'|x_{\{s\}})q_\phi(\tilde{z}_{\{s\}}|x_{\{s\}})\Big[\log p_\theta(x_{\{s\}}|z', \tilde{z}_{\{s\}})\Big]\mathrm{d}z'\mathrm{d}\tilde{z}_{\{s\}} \\
&+ \int q_\phi(z'|x_{\{s\}})r_\phi(\tilde{z}_{\backslash\{s\}})\Big[\log p_\theta(x_{\backslash\{s\}}|z', \tilde{z}_{\backslash\{s\}})\Big]\mathrm{d}z'\mathrm{d}\tilde{z}_{\backslash\{s\}} \\
&- \mathsf{KL}\Big(q_\phi^{\mathrm{MoE}}(z', \tilde{z}_\mathcal{M}|x_\mathcal{M})\Big|p_\theta(z')p_\theta(\tilde{z}_\mathcal{M})\Big).
\end{aligned}
$$

Here, it is assumed that the multi-modal encoding distribution for computing the KL-divergence is of the form

$$q_\phi^{\text{MoE}}(z', \tilde{z}_\mathcal{M}|x_\mathcal{M}) = \frac{1}{M} \sum_{s \in \mathcal{M}} (q_\phi(z'|x_s) q_\phi(\tilde{z}_s|x_s))$$

and $r_\phi(\tilde{z}_\mathcal{A}) = \prod_{s \in \mathcal{A}} r_\phi(\tilde{z}_s)$ are additional trainable *prior* distributions.

**Remark 25 (Cross-modal context variables and permutation-equivariant models)** In contrast to the PoE model, where the private encodings are independent, the private encodings are dependent in the Sum-Pooling model by conditioning on a sample from the shared latent space. The shared latent variable $Z'$ can be seen as a shared cross-modal context variable, and similar probabilistic constructions to encode such context variables via permutation-invariant models have been suggested in few-shot learning algorithms (Edwards and Storkey, 2016; Giannone and Winther, 2022) or, particularly, for neural process models (Garnelo et al., 2018b;a; Kim et al., 2018). Permutation-equivariant models have been studied for stochastic processes where invariant priors correspond to equivariant posteriors (Holderrieth et al., 2021), such as Gaussian processes or Neural processes with private latent variables, wherein dependencies in the private latent variables can be constructed hierarchically (Wang and Van Hoof, 2020; Xu et al., 2023).

## F  Multi-modal posterior in exponential family models

Consider the setting where the decoding and encoding distributions are of the exponential family form, that is

$$p_\theta(x_s|z) = \mu_s(x_s) \exp\left[\langle T_s(x_s), f_{s,\theta}(z)\rangle - \log Z_s(f_{s,\theta}(z))\right]$$

for all $s \in \mathcal{M}$, while for all $\mathcal{S} \subset \mathcal{M}$,

$$q_\phi(z|x_\mathcal{S}) = \mu(z) \exp\left[\langle V(z), \lambda_{\phi,\mathcal{S}}(x_\mathcal{S})\rangle - \log \Gamma_\mathcal{S}(\lambda_{\phi,\mathcal{S}}(x_\mathcal{S}))\right]$$

where $\mu_s$ and $\mu$ are base measures, $T_s(x_s)$ and $V(z)$ are sufficient statistics, while the natural parameters $\lambda_{\phi,\mathcal{S}}(x_\mathcal{S})$ and $f_{s,\theta}(z)$ are parameterized by the decoder or encoder networks, respectively, with $Z_s$ and $\Gamma_\mathcal{S}$ being normalizing functions. Note that we made a standard assumption that the multi-modal encoding distribution has a fixed base measure and sufficient statistics for any modality subset. For fixed generative parameters $\theta$, we want to learn a multi-modal encoding distribution that minimizes over $x_\mathcal{S} \sim p_d$,

$$\begin{aligned}
&\mathsf{KL}(q_\phi(z|x_\mathcal{S})|p_\theta(z|x_\mathcal{S})) \\
&= \int q_\phi(z|x_\mathcal{S})\Big[\log q_\phi(z|x_\mathcal{S}) - \log p_\theta(z) - \sum_{s \in \mathcal{S}} \log p_\theta(x_s|z)\Big] \mathrm{d}z - \log p_\theta(x_\mathcal{S}) \\
&= \int q_\phi(z|x_\mathcal{S})\Big[\langle V(z), \lambda_{\phi,\mathcal{S}}(x_\mathcal{S})\rangle - \log \Gamma_\mathcal{S}(\lambda_{\phi,\mathcal{S}}(x_\mathcal{S})) - \sum_{s \in \mathcal{S}} \log \mu_s(x_s) \\
&\quad - \Big\{\sum_{s \in \mathcal{S}} \langle T_{s,\theta}(x_s), f_{s,\theta}(z)\rangle + \log p_\theta(z) - \sum_{s \in \mathcal{S}} Z_s(f_{s,\theta}(z))\Big\}\Big] \mathrm{d}z - \log p_\theta(x_\mathcal{S}) \\
&= \int q_{\phi,\vartheta}(z|x_\mathcal{S})\Big[\Big\langle \begin{bmatrix} V(z) \\ 1 \end{bmatrix}, \begin{bmatrix} \lambda_{\phi,\vartheta,\mathcal{S}}(x_\mathcal{S}) \\ -\log \Gamma_\mathcal{S}(\lambda_{\phi,\vartheta,\mathcal{S}}(x_\mathcal{S})) \end{bmatrix}\Big\rangle - \sum_{s \in \mathcal{S}} \Big\langle \begin{bmatrix} T_s(x_s) \\ 1 \end{bmatrix}, \begin{bmatrix} f_{\theta,s}(z) \\ b_{\theta,s}(z) \end{bmatrix}\Big\rangle\Big] \mathrm{d}z,
\end{aligned}$$

with $b_{\theta,s}(z) = \frac{1}{|\mathcal{S}|} p_\theta(z) - \log Z_s(f_{s,\theta}(z))$.

## G  Mixture model extensions for different variational bounds

We consider the optimization of an augmented variational bound

$$\begin{aligned}
\mathcal{L}(x, \theta, \phi) = \int \rho(\mathcal{S})\Big[&\int q_\phi(c, z|x_\mathcal{S})\left[\log p_\theta(c, x_\mathcal{S}|z)\right] \mathrm{d}z\mathrm{d}c - \mathsf{KL}(q_\phi(c, z|x_\mathcal{S})|p_\theta(c, z)) \\
&+ \int q_\phi(c, z|x_\mathcal{S})\left[\log p_\theta(x_{\backslash\mathcal{S}}|z)\right] \mathrm{d}z\mathrm{d}c - \mathsf{KL}(q_\phi(c, z|x)|q_\phi(c, z|x_\mathcal{S}))\Big] \mathrm{d}\mathcal{S}.
\end{aligned}$$

We will pursue here an encoding approach that does not require modeling the encoding distribution over the discrete latent variables explicitly, thus avoiding large variances in score-based Monte Carlo estimators (Ranganath et al., 2014) or resorting to advanced variance reduction techniques (Kool et al., 2019) or alternatives such as continuous relaxation approaches (Jang et al., 2016; Maddison et al., 2016).

Assuming a structured variational density of the form

$$q_\phi(c, z|x_\mathcal{S}) = q_\phi(z|x_\mathcal{S})q_\phi(c|z, x_\mathcal{S}),$$

we can express the augmented version of (3) via

$$\mathcal{L}_\mathcal{S}(x_\mathcal{S}, \theta, \phi) = \int q_\phi(c, z|x_\mathcal{S}) \left[\log p_\theta(c, x_\mathcal{S}|z)\right] \mathrm{d}z - \beta \mathsf{KL}(q_\phi(c, z|x_\mathcal{S})|p_\theta(c, z))$$

$$= \int q_\phi(z|x_\mathcal{S}) \left[f_x(z, x_\mathcal{S}) + f_c(z, x_\mathcal{S})\right] \mathrm{d}z,$$

where $f_x(z, x_\mathcal{S}) = \log p_\theta(x_\mathcal{S}|z) - \beta \log q_\phi(z|x_\mathcal{S}))$ and

$$f_c(z, x_\mathcal{S}) = \int q_\phi(c|z, x_\mathcal{S}) \left[-\beta \log q_\phi(c|z, x_\mathcal{S}) + \beta \log p_\theta(c, z)\right] \mathrm{d}c. \tag{19}$$

We can also write the augmented version of (5) in the form of

$$\mathcal{L}_{\backslash \mathcal{S}}(x, \theta, \phi) = \int q_\phi(c, z|x_\mathcal{S}) \left[\log p_\theta(x_{\backslash \mathcal{S}}|z)\right] \mathrm{d}z - \beta \mathsf{KL}(q_\phi(c, z|x)|q_\phi(c, z|x_\mathcal{S}))$$

$$= \int q_\phi(z|x) g_x(z, x) \mathrm{d}z$$

where

$$g_x(z, x) = \log p_\theta(x_{\backslash \mathcal{S}}|z) - \beta \log q_\phi(z|x) + \beta \log q_\phi(z|x_\mathcal{S})$$

which does not depend on the encoding density of the cluster variable. To optimize the variational bound with respect to the cluster density, we can thus optimize (19), which attains its maximum value of

$$f_c^\star(z, x_\mathcal{S}) = \beta \log \int p_\theta(c) p_\theta(z|c) \mathrm{d}c = \beta \log p_\theta(z)$$

at $q_\phi(c|z, x_\mathcal{S}) = p_\theta(c|z)$ due to Remark 18 below with $g(c) = \beta \log p_\theta(c, z)$.

We can derive an analogous optimal structured variational density for the mixture-based and total-correlation-based variational bounds. First, we can write the mixture-based bound (1) as

$$\mathcal{L}_\mathcal{S}^{\mathrm{Mix}}(x, \theta, \phi) = \int q_\phi(z|x_\mathcal{S}) \left[\log p_\theta(c, x|z)\right] \mathrm{d}z - \beta \mathsf{KL}(q_\phi(c, z|x_\mathcal{S})|p_\theta(c, z))$$

$$= \int q_\phi(z|x_\mathcal{S}) \left[f_x^{\mathrm{Mix}}(z, x) + f_c(z, x)\right] \mathrm{d}z,$$

where $f_x^{\mathrm{Mix}}(z, x) = \log p_\theta(x|z) - \beta \log q_\phi(z|x_\mathcal{S})$ and $f_c(z, x)$ has a maximum value of $f_c^\star(z, x) = \beta \log p_\theta(z)$. Second, we can express the corresponding terms from the total-correlation-based bound as

$$\mathcal{L}_\mathcal{S}^{\mathrm{TC}}(\theta, \phi) = \int q_\phi(z|x) \left[\log p_\theta(x|z)\right] \mathrm{d}z - \beta \mathsf{KL}(q_\phi(c, z|x)|q_\phi(c, z|x_\mathcal{S}))$$

$$= \int q_\phi(z|x) \left[f_x^{\mathrm{TC}}(z, x)\right] \mathrm{d}z,$$

where $f_x^{\mathrm{TC}}(z, x) = \log p_\theta(x|z) - \beta \log q_\phi(z|x) + \beta \log q_\phi(z|x_\mathcal{S})$.

# H  Algorithm and STL-gradient estimators

We consider a multi-modal extension of the sticking-the-landing (STL) gradient estimator (Roeder et al., 2017) that has also been used in previous multi-modal bounds (Shi et al., 2019). The gradient estimator ignores the score function terms when sampling $q_\phi(z|x_\mathcal{S})$ for variance reduction purposes because it has a zero expectation. For the bounds (2) that involves sampling from $q_\phi(z|x_\mathcal{S})$ and $q_\phi(z|x_\mathcal{M})$, we thus ignore the score terms for both integrals. Consider the reparameterization with noise variables $\epsilon_\mathcal{S}$, $\epsilon_\mathcal{M} \sim p$ and transformations $z_\mathcal{S} = t_\mathcal{S}(\phi, \epsilon_\mathcal{S}, x_\mathcal{S}) = f_{\text{invariant-agg}}(\vartheta, \epsilon_\mathcal{S}, \mathcal{S}, h_\mathcal{S})$, for $h_\mathcal{S} = h_{\varphi,s}(x_s)_{s \in \mathcal{S}}$ and $z_\mathcal{M} = t_\mathcal{M}(\phi, \epsilon_\mathcal{M}, x_\mathcal{M}) = f_{\text{invariant-agg}}(\vartheta, \epsilon_\mathcal{M}, \mathcal{M}, h_\mathcal{M})$, for $h_\mathcal{M} = h_{\varphi,s}(x_s)_{s \in \mathcal{M}}$. We need to learn only a single aggregation function that applies and masks the modalities appropriately. Pseudo-code for computing the gradients are given in Algorithm 1. If the encoding distribution is a mixture distribution, we apply the stop-gradient operation also to the mixture weights. Notice that in the case of a mixture prior and an encoding distribution that includes the mixture component, the optimal encoding density over the mixture variable has no variational parameters and is given as the posterior density of the mixture component under the generative parameters of the prior.

---

**Algorithm 1** Single training step for computing unbiased gradients of $\mathcal{L}(x)$.

---

**Input:** Multi-modal data point $x$, generative parameter $\theta$, variational parameters $\phi = (\varphi, \vartheta)$.
Sample $\mathcal{S} \sim \rho$.
Sample $\epsilon_\mathcal{S}, \epsilon_\mathcal{M} \sim p$.
Set $z_\mathcal{S} = t_\mathcal{S}(\phi, \epsilon_\mathcal{S}, x_\mathcal{M})$ and $z_\mathcal{M} = t_\mathcal{M}(\phi, \epsilon_\mathcal{M}, x_\mathcal{M})$.
Stop gradients of variational parameters $\phi' = \texttt{stop\_grad}(\phi)$.
Set $\widehat{\mathcal{L}}_\mathcal{S}(\theta, \phi) = \log p_\theta(x_\mathcal{S}|z_\mathcal{S}) + \beta \log p_\theta(z_\mathcal{S}) - \beta \log q_{\phi'}(z_\mathcal{S}|x_\mathcal{S})$.
Set $\widehat{\mathcal{L}}_{\backslash \mathcal{S}}(\theta, \phi) = \log p_\theta(x_{\backslash \mathcal{S}}|z_\mathcal{M}) + \beta \log q_\phi(z_\mathcal{M}|x_\mathcal{S}) - \beta \log q_{\phi'}(z_\mathcal{M}|x_\mathcal{M})$.
**Output:** $\nabla_{\theta, \phi} \left[ \widehat{\mathcal{L}}_\mathcal{S}(\theta, \phi) + \widehat{\mathcal{L}}_{\backslash \mathcal{S}}(\theta, \phi) \right]$

---

In the case of private latent variables, we proceed analogously and rely on reparameterizations $z'_\mathcal{S} = t'_\mathcal{S}(\phi, \epsilon'_\mathcal{S}, x_\mathcal{S})$ for the shared latent variable $z'_\mathcal{S} \sim q_\phi(z'|x_\mathcal{S})$ as above and $\tilde{z}_\mathcal{S} = \tilde{t}_\mathcal{S}(\phi, z', \epsilon_\mathcal{S}, x_\mathcal{S}) = f_{\text{equivariant-agg}}(\vartheta, \tilde{\epsilon}_\mathcal{S}, z', \mathcal{S}, h_\mathcal{S})$ for the private latent variables $\tilde{z}_\mathcal{S} \sim q_\phi(\tilde{z}_\mathcal{S}|z', x_\mathcal{S})$. Moreover, we write $\mathsf{P}_\mathcal{S}$ for a projection on the $\mathcal{S}$-coordinates. Pseudo-code for computing unbiased gradient estimates for our bound is given in Algorithm 2.

---

**Algorithm 2** Single training step for computing unbiased gradients of $\mathcal{L}(x)$ with private latent variables.

---

**Input:** Multi-modal data point $x$, generative parameter $\theta$, variational parameters $\phi = (\varphi, \vartheta)$.
Sample $\mathcal{S} \sim \rho$.
Sample $\epsilon'_\mathcal{S}, \epsilon_\mathcal{S}, \epsilon_{\backslash \mathcal{S}}, \epsilon'_\mathcal{M}, \epsilon_\mathcal{M}, \epsilon_{\backslash \mathcal{M}} \sim p$.
Set $z'_\mathcal{S} = t'_\mathcal{S}(\phi, \epsilon'_\mathcal{S}, x_\mathcal{S})$, $\tilde{z}_\mathcal{S} = \tilde{t}_\mathcal{S}(\phi, z'_\mathcal{S}, \epsilon_\mathcal{S}, x_\mathcal{S})$.
Set $z'_\mathcal{M} = t'_\mathcal{M}(\phi, \epsilon'_\mathcal{M}, x_\mathcal{M})$, $\tilde{z}_\mathcal{M} = \tilde{t}_\mathcal{M}(\phi, z'_\mathcal{M}, \epsilon_\mathcal{M}, x_\mathcal{M})$.
Stop gradients of variational parameters $\phi' = \texttt{stop\_grad}(\phi)$.
Set $\widehat{\mathcal{L}}_\mathcal{S}(\theta, \phi) = \log p_\theta(x_\mathcal{S}|z'_\mathcal{S}, \tilde{z}_\mathcal{S}) + \beta \log p_\theta(z'_\mathcal{S}) - \beta \log q_{\phi'}(z'_\mathcal{S}|x_\mathcal{S}) + \beta \log p_\theta(\tilde{z}_\mathcal{S}|z'_\mathcal{S}) - \beta \log q_{\phi'}(\tilde{z}_\mathcal{S}|z'_\mathcal{S}, x_\mathcal{S})$.
Set $\widehat{\mathcal{L}}_{\backslash \mathcal{S}}(\theta, \phi) = \log p_\theta(x_{\backslash \mathcal{S}}|z'_\mathcal{M}) + \beta \log q_\phi(z'_\mathcal{M}|x_\mathcal{S}) - \beta \log q_{\phi'}(\tilde{z}_\mathcal{M}|z'_\mathcal{M}, x_\mathcal{M}) + \beta \log q_\phi(\mathsf{P}_\mathcal{S}(\tilde{z}_\mathcal{M})|z'_\mathcal{M}, x_\mathcal{S}) + \beta \log p_\theta(\mathsf{P}_{\backslash \mathcal{S}}(\tilde{z}_\mathcal{M})|z'_\mathcal{M}, \tilde{z}_\mathcal{M}) - \beta \log q_{\phi'}(\tilde{z}_\mathcal{M}|z'_\mathcal{M}, x_\mathcal{M})$.
**Output:** $\nabla_{\theta, \phi} \left[ \widehat{\mathcal{L}}_\mathcal{S}(\theta, \phi) + \widehat{\mathcal{L}}_{\backslash \mathcal{S}}(\theta, \phi) \right]$

---

# I  Evaluation of multi-modal generative models

We evaluate models using different metrics suggested previously for multi-modal learning, see for example Shi et al. (2019); Wu and Goodman (2019); Sutter et al. (2021).

**Marginal, conditional and joint log-likelihoods.** We can estimate the marginal log-likelihood using classic importance sampling

$$\log p_\theta(x_\mathcal{S}) \approx \log \frac{1}{K} \sum_{k=1}^{K} \frac{p_\theta(z^k, x_\mathcal{S})}{q_\phi(z^k|x_\mathcal{S})}$$

for $z^k \sim q_\phi(\cdot|x_\mathcal{S})$. This also allows to approximate the joint log-likelihood $\log p_\theta(x)$, and consequently also the conditional $\log p_\theta(x_{\backslash \mathcal{S}}|x_\mathcal{S}) = \log p_\theta(x) - \log p_\theta(x_\mathcal{S})$.

**Generative coherence with joint auxiliary labels.** Following previous work (Shi et al., 2019; Sutter et al., 2021; Daunhawer et al., 2022; Javaloy et al., 2022), we assess whether the generated data share the same information in the form of the class labels across different modalities. To do so, we use pre-trained classifiers $\mathrm{clf}_s \colon \mathsf{X}_s \to [K]$ that classify values from modality $s$ to $K$ possible classes. More precisely, for $\mathcal{S} \subset \mathcal{M}$ and $m \in \mathcal{M}$, we compute the self- ($m \in \mathcal{S}$) or cross- ($m \notin \mathcal{S}$) coherence $\mathrm{C}_{\mathcal{S} \to m}$ as the empirical average of

$$\mathbb{1}_{\{\mathrm{clf}_m(\hat{x}_m) = y\}},$$

over test samples $x$ with label $y$ where $\hat{z}_\mathcal{S} \sim q_\phi(z|x_\mathcal{S})$ and $\hat{x}_m \sim p_\theta(x_m|\hat{z}_\mathcal{S})$. The case $\mathcal{S} = \mathcal{M} \setminus \{m\}$ corresponds to a leave-one-out conditional coherence.

**Linear classification accuracy of latent representations.** To evaluate how the latent representation can be used to predict the shared information contained in the modality subset $\mathcal{S}$ based on a linear model, we consider the accuracy $\mathrm{Acc}_\mathcal{S}$ of a linear classifier $\mathrm{clf}_z \colon \mathsf{Z} \to [K]$ that is trained to predict the label based on latent samples $z_\mathcal{S} \sim q_\phi(z_\mathcal{S}|x_\mathcal{S}^{\mathrm{train}})$ from the training values $x_\mathcal{S}^{\mathrm{train}}$ and evaluated on latent samples $z_\mathcal{S} \sim q_\phi(z|x_\mathcal{S}^{\mathrm{test}})$ from the test values $x_\mathcal{S}^{\mathrm{test}}$.

## J   Linear models

**Data generation.** We generate 5 data sets of $N = 5000$ samples, each with $M = 5$ modalities. We set the latent dimension to $D = 30$, while the dimension $D_s$ of modality $s$ is drawn from $\mathcal{U}(30, 60)$. We set the observation noise to $\sigma = 1$, shared across all modalities, as is standard for a PCA model. We sample the components of $b_s$ independently from $\mathcal{N}(0, 1)$. For the setting without modality-specific latent variables, $W_s$ is the orthonormal matrix from a QR algorithm applied to a matrix with elements sampled iid from $\mathcal{U}(-1, 1)$. The bias coefficients $W_b$ are sampled independently from $\mathcal{N}(0, 1/d)$. Conversely, the setting with private latent variables in the ground truth model allows us to describe modality-specific variation by considering the sparse loading matrix

$$W_\mathcal{M} = \begin{bmatrix} W_1' & \tilde{W}_1 & 0 & \dots & 0 \\ W_2' & 0 & \tilde{W}_2 & \dots & 0 \\ \vdots & \vdots & \ddots & \ddots & \vdots \\ W_M' & 0 & \dots & 0 & \tilde{W}_M \end{bmatrix}.$$

Here, $W_s', \tilde{W}_s \in \mathbb{R}^{D_s \times D'}$ with $D' = D/(M+1) = 5$, Furthermore, the latent variable $Z$ can be written as $Z = (Z', \tilde{Z}_1, \dots, \tilde{Z}_M)$ for private and shared latent variables $\tilde{Z}_s$, resp. $Z'$. We similarly generate orthonormal $[W_s', \tilde{W}_s]$ from a QR decomposition. Observe that the general generative model with latent variable $Z$ corresponds to the generative model (9) with shared $Z'$ and private latent variables $\tilde{Z}$ with straightforward adjustments for the decoding functions. Similar models have been considered previously, particularly from a Bayesian standpoint with different sparsity assumptions on the generative parameters (Archambeau and Bach, 2008; Virtanen et al., 2012; Zhao et al., 2016).

**Maximum likelihood estimation.** Assume now that we observe $N$ data points $\{x_n\}_{n \in [N]}$, consisting of stacking the views $x_n = (x_{s,n})_{s \in \mathcal{S}}$ for each modality in $\mathcal{S}$ and let $S = \frac{1}{N} \sum_{n=1}^{N} (x_n - b)(x_n - b)^\top \in \mathbb{R}^{D_x \times D_x}$, $D_x = \sum_{s=1}^{M} D_s$, be the sample covariance matrix across all modalities. Let $U_d \in \mathbb{R}^{D_x \times D}$ be the matrix of the first $D$ eigenvectors of $S$ with corresponding eigenvalues $\lambda_1, \dots \lambda_D$ stored in the diagonal matrix $\Lambda_D \in \mathbb{R}^{D \times D}$.

The maximum likelihood estimates are then given by $b_{\mathrm{ML}} = \frac{1}{N}\sum_{n=1}^{N} x_n$, $\sigma^2_{\mathrm{ML}} = \frac{1}{N-D}\sum_{j=D+1}^{N}\lambda_j$ and $W_{\mathrm{ML}} = U_D(\Lambda_D - \sigma^2_{\mathrm{ML}}\mathrm{I})^{1/2}$ with the loading matrix identifiable up to rotations.

**Model architectures.** We estimate the observation noise scale $\sigma$ based on the maximum likelihood estimate $\sigma_{\mathrm{ML}}$. We assume linear decoder functions $p_\theta(x_s|z) = \mathcal{N}(W_s^\theta z + b^\theta, \sigma^2_{\mathrm{ML}})$, fixed standard Gaussian prior $p(z) = \mathcal{N}(0, \mathrm{I})$ and generative parameters $\theta = (W_1^\theta, b_1^\theta, \ldots, W_M^\theta, b_M^\theta)$. Details about the various encoding architectures are given in Table 15. The modality-specific encoding functions for the PoE and MoE schemes have a hidden size of 512, whilst they are of size 256 for the learnable aggregation schemes having additional aggregation parameters $\varphi$.

## K  Non-linear identifiable models

We also show in Figure 6 the reconstructed modality values and inferred latent variables for one realization with our bound, with the corresponding results for a mixture-based bound in Figure 7.

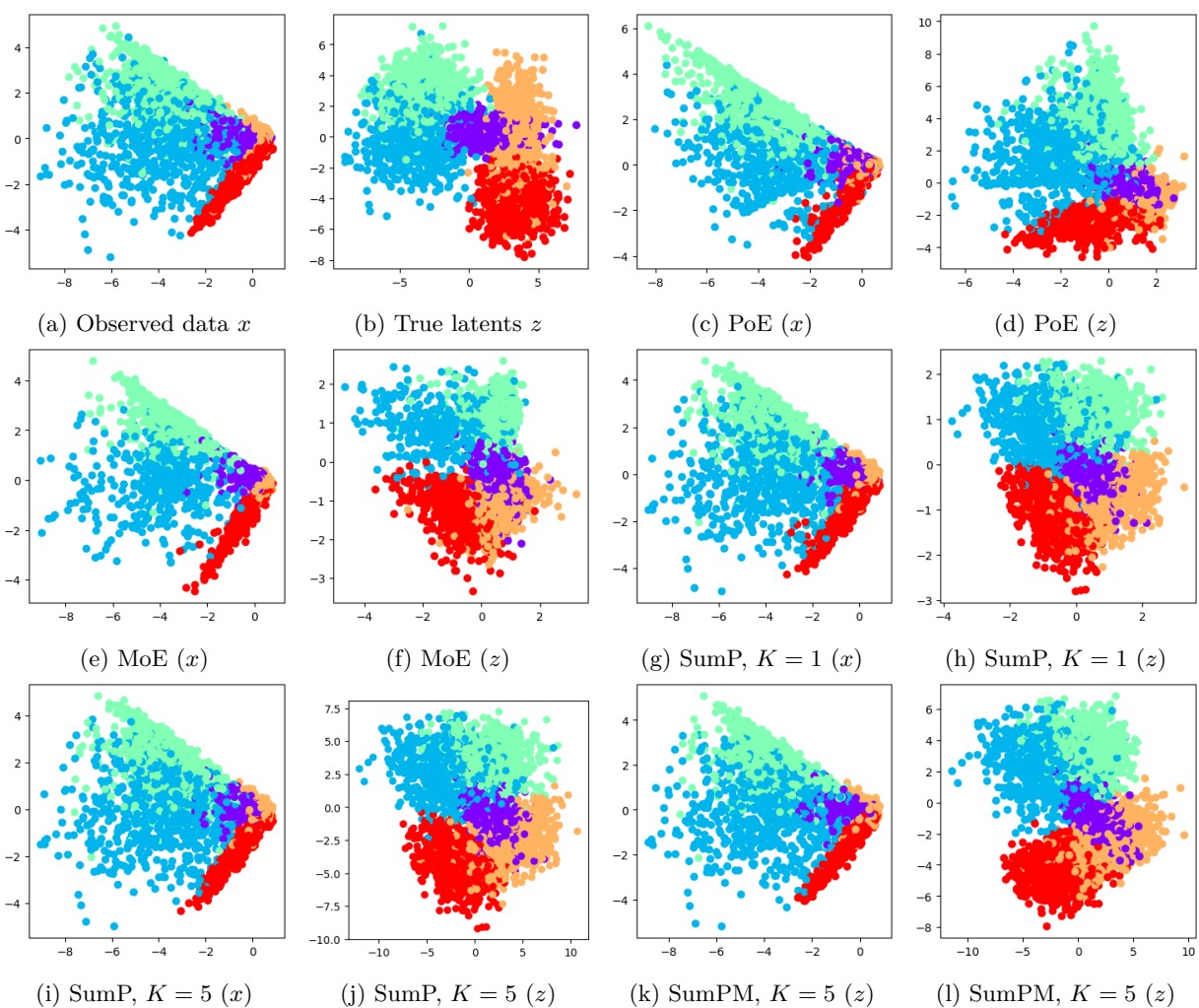

Figure 6: Bi-modal non-linear model with label and continuous modality based on our proposed objective. SumP: SumPooling, SumPM: SumPoolingMixture.

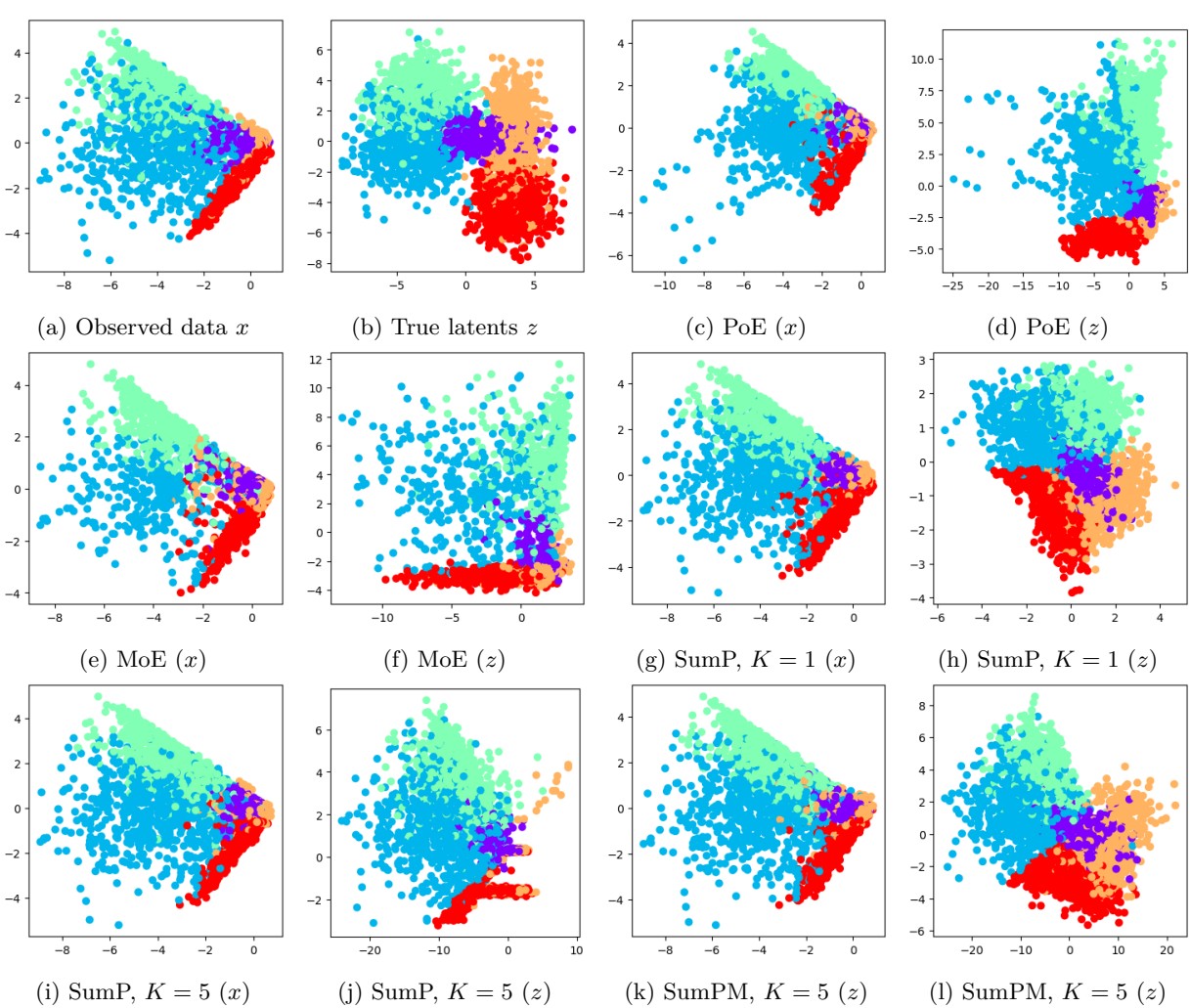

Figure 7: Bi-modal non-linear model with label and continuous modality based on mixture bound. SumP: SumPooling, SumPM: SumPoolingMixture.

# L MNIST-SVHN-Text

## L.1 Training hyperparamters

The MNIST-SVHN-Text data set is taken from the code accompanying Sutter et al. (2021) with around 1.1 million train and 200k test samples. All models are trained for 100 epochs with a batch size of 250 using Adam (Kingma and Ba, 2014) and a cosine decay schedule from 0.0005 to 0.0001.

## L.2 Multi-modal rates and distortions

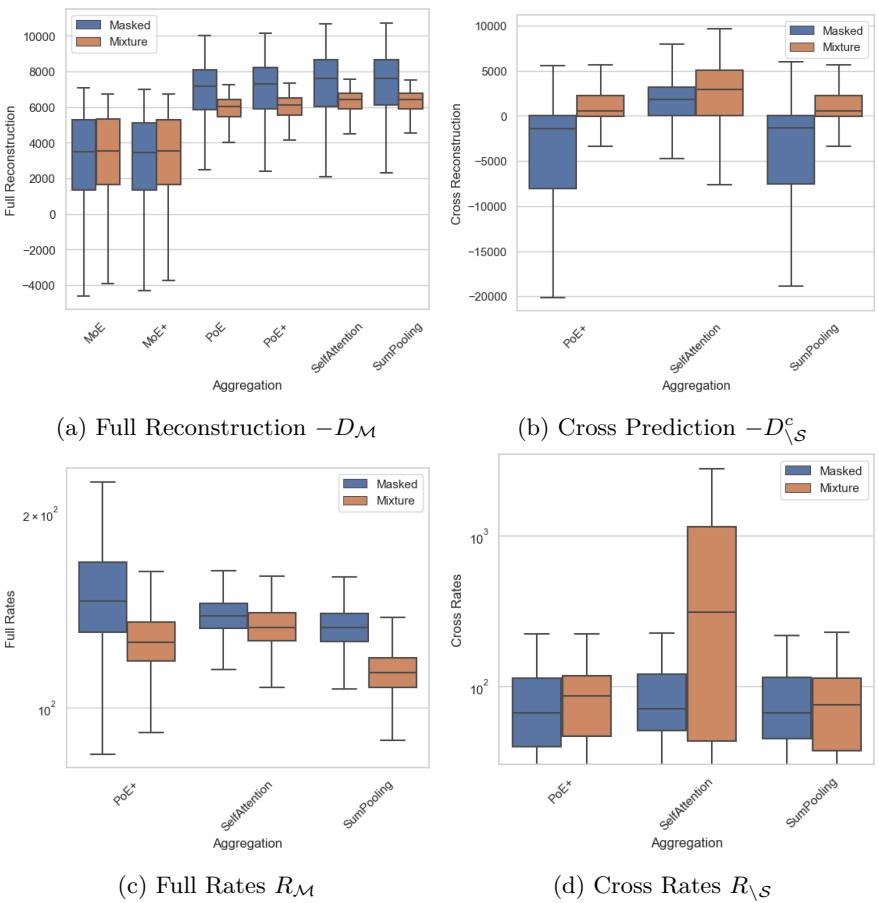

(a) Full Reconstruction $-D_{\mathcal{M}}$

(b) Cross Prediction $-D^{c}_{\backslash \mathcal{S}}$

(c) Full Rates $R_{\mathcal{M}}$

(d) Cross Rates $R_{\backslash \mathcal{S}}$

Figure 8: Rate and distortion terms for MNIST-SVHN-Text with shared and private latent variables.

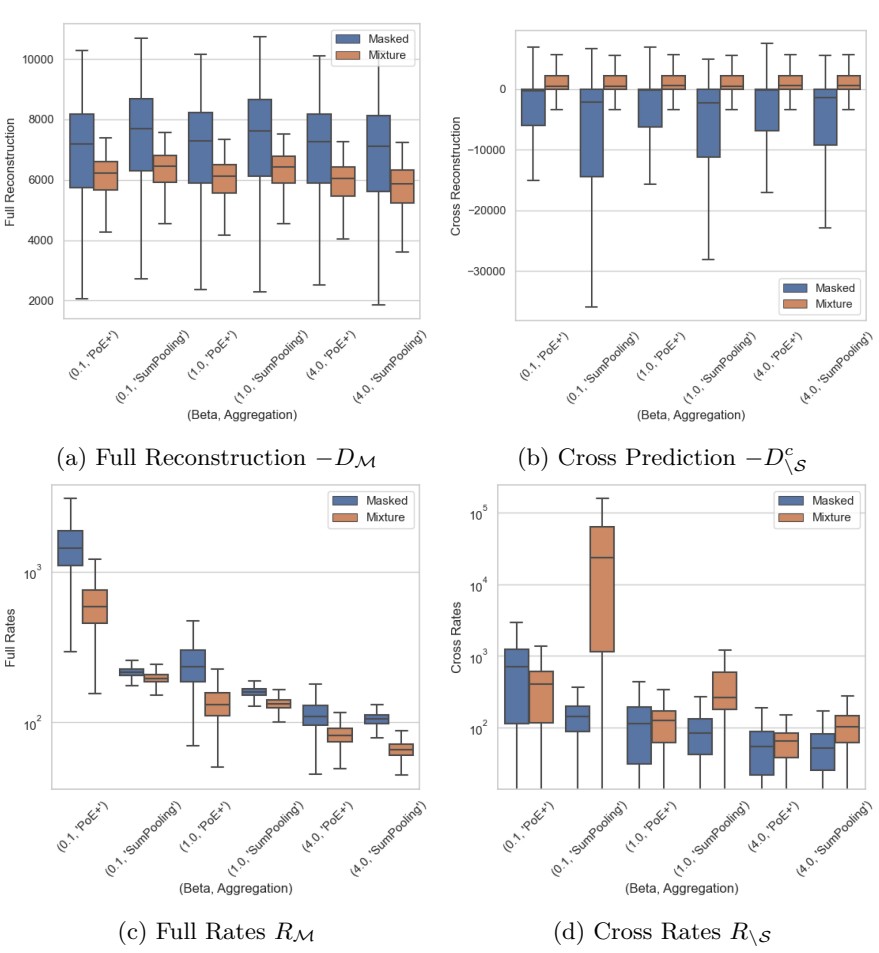

(a) Full Reconstruction $-D_{\mathcal{M}}$

(b) Cross Prediction $-D_{\backslash \mathcal{S}}^{c}$

(c) Full Rates $R_{\mathcal{M}}$

(d) Cross Rates $R_{\backslash \mathcal{S}}$

Figure 9: Rate and distortion terms for MNIST-SVHN-Text with shared latent variables and different $\beta$.

### L.3 Log-likelihood estimates

Table 7: Test log-likelihood estimates for varying $\beta$ choices for the joint data (M+S+T) as well as for the marginal data of each modality based on importance sampling (512 particles). Multi-modal generative model with a 40-dimensional shared latent variable. The second part of the Table contains reported log-likelihood values from baseline methods that, however, impose more restrictive assumptions on the decoder variances, which likely contributes to much lower log-likelihood values reported in previous works, irrespective of variational objectives and aggregation schemes.

| ($\beta$, Aggregation) | Proposed objective | | | | Mixture bound | | | |
|---|---|---|---|---|---|---|---|---|
| | M+S+T | M | S | T | M+S+T | M | S | T |
| (0.1, PoE+) | 5433 (24.5) | 1786 (41.6) | 3578 (63.5) | -29 (2.4) | 5481 (18.4) | 2207 (19.8) | 3180 (33.7) | -39 (1.0) |
| (0.1, SumPooling) | **7067 (78.0)** | 2455 (3.3) | **4701 (83.5)** | -9 (0.4) | 6061 (15.7) | 2398 (9.3) | 3552 (7.4) | -50 (1.9) |
| (1.0, PoE+) | 6872 (9.6) | 2599 (5.6) | 4317 (1.1) | -9 (0.2) | 5900 (10.0) | 2449 (10.4) | 3443 (11.7) | -19 (0.4) |
| (1.0, SumPooling) | 7056 (124.4) | 2478 (9.3) | 4640 (113.9) | -6 (0.0) | 6130 (4.4) | 2470 (10.3) | 3660 (1.5) | -16 (1.6) |
| (4.0, PoE+) | 7021 (13.3) | **2673 (13.2)** | 4413 (30.5) | **-5 (0.1)** | 5895 (6.2) | 2484 (5.5) | 3434 (2.2) | -13 (0.4) |
| (4.0, SumPooling) | 6690 (113.4) | 2483 (9.9) | 4259 (117.2) | **-5 (0.0)** | 5659 (48.3) | 2448 (10.5) | 3233 (27.7) | -10 (0.2) |
| Results from Sutter et al. (2021) and Sutter et al. (2020) | | | | | | | | |
| MVAE | -1790 (3.3) | NA | NA | NA | | | | |
| MMVAE | -1941 (5.7) | NA | NA | NA | | | | |
| MoPoE | -1819 (5.7) | NA | NA | NA | | | | |
| MMJSD | -1961 (NA) | NA | NA | NA | | | | |

### L.4 Generated modalities

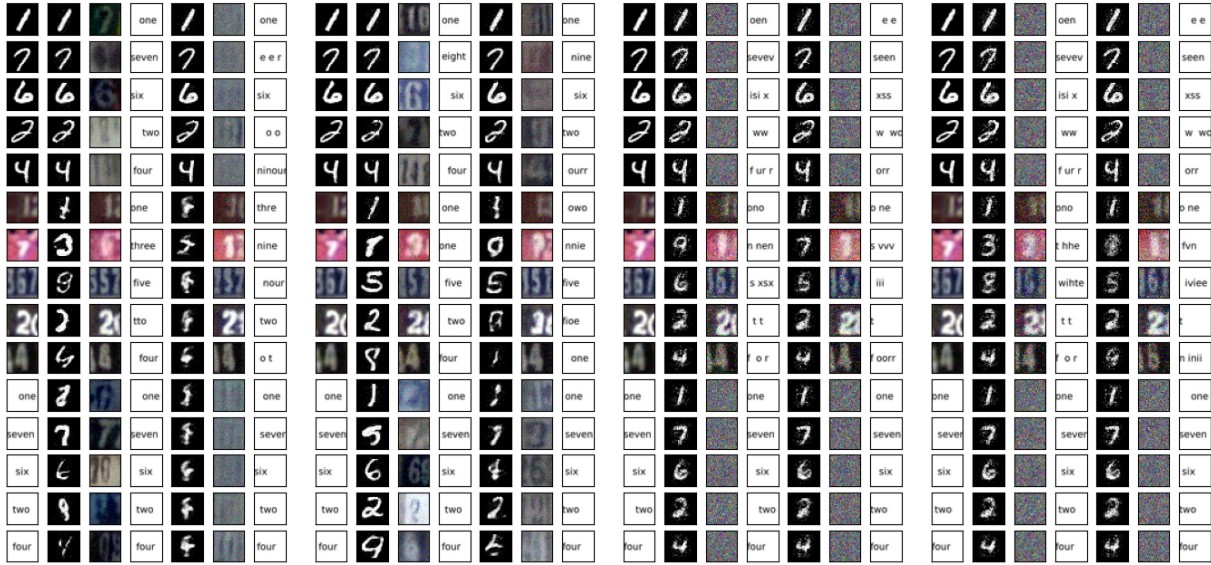

(a) Proposed objective, $\beta =$ 0.1    (b) Proposed objective, $\beta =$ 4    (c) Mixture-based bound, $\beta = 0.1$    (d) Mixture-based bound, $\beta = 4$

Figure 10: Conditional generation for different $\beta$ parameters. The first column is the conditioned modality. The next three columns are the generated modalities using a SumPooling aggregation, followed by the three columns for a PoE+ scheme.

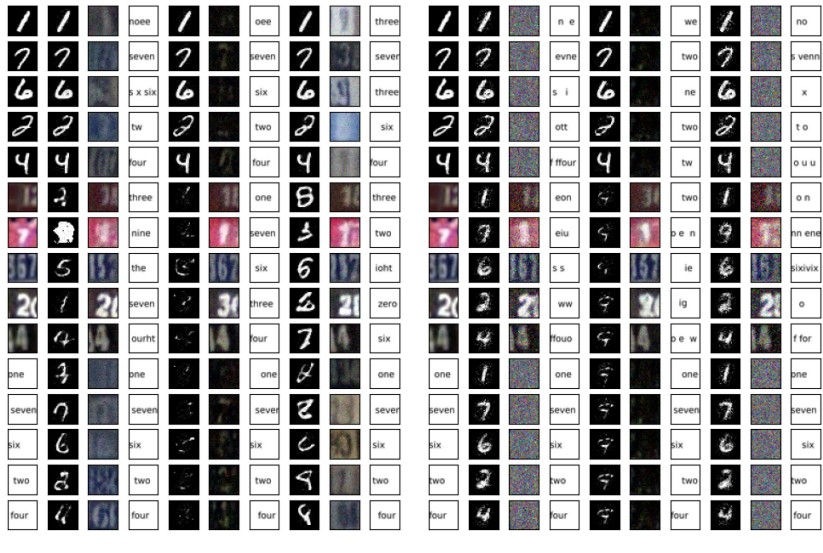

| (a) Our bound | (b) Mixture-based bound |
|---|---|

Figure 11: Conditional generation for permutation-equivariant schemes and private latent variable constraints. The first column is the conditioned modality. The next three columns are the generated modalities using a SumPooling aggregation, followed by the three columns for a SelfAttention scheme and a PoE model.

## L.5 Conditional coherence

Table 8: Conditional coherence for models with shared latent variables and bi-modal conditionals. The letters on the second line represent the modality which is generated based on the sets of modalities on the line below it.

| | Proposed objective | | | | | | | | | Mixture bound | | | | | | | | |
|---|---|---|---|---|---|---|---|---|---|---|---|---|---|---|---|---|---|---|
| | M | | | S | | | T | | | M | | | S | | | T | | |
| Aggregation | M+S | M+T | S+T | M+S | M+T | S+T | M+S | M+T | S+T | M+S | M+T | S+T | M+S | M+T | S+T | M+S | M+T | S+T |
| PoE | **0.98** | **0.98** | 0.60 | 0.75 | **0.58** | 0.77 | 0.82 | **1.00** | **1.00** | 0.96 | 0.97 | 0.95 | 0.61 | 0.11 | 0.61 | 0.45 | 0.99 | 0.98 |
| PoE+ | 0.97 | **0.98** | 0.55 | 0.73 | 0.52 | 0.75 | **0.83** | **1.00** | 0.99 | 0.97 | 0.97 | **0.96** | 0.64 | 0.11 | 0.63 | 0.45 | 0.99 | 0.97 |
| MoE | 0.88 | 0.97 | 0.90 | 0.35 | 0.11 | 0.35 | 0.41 | 0.72 | 0.69 | 0.88 | 0.96 | 0.89 | 0.32 | 0.10 | 0.33 | 0.42 | 0.72 | 0.69 |
| MoE+ | 0.85 | 0.94 | 0.86 | 0.32 | 0.10 | 0.32 | 0.40 | 0.71 | 0.67 | 0.87 | 0.96 | 0.89 | 0.32 | 0.10 | 0.32 | 0.42 | 0.72 | 0.69 |
| SumPooling | 0.97 | 0.97 | 0.86 | 0.78 | 0.30 | **0.80** | 0.76 | 0.99 | **1.00** | 0.97 | 0.97 | 0.95 | 0.65 | 0.10 | 0.65 | 0.45 | 0.99 | 0.97 |
| SelfAttention | 0.97 | 0.97 | 0.82 | **0.76** | 0.30 | 0.78 | 0.69 | **1.00** | **1.00** | 0.97 | 0.97 | 0.99 | 0.66 | 0.10 | 0.65 | 0.45 | 0.99 | **1.00** |
| | Results from Sutter et al. (2021), Sutter et al. (2020) and Hwang et al. (2021) | | | | | | | | | | | | | | | | | |
| MVAE | NA | NA | 0.32 | NA | 0.43 | NA | 0.29 | NA | NA | | | | | | | | | |
| MMVAE | NA | NA | 0.87 | NA | 0.31 | NA | 0.84 | NA | NA | | | | | | | | | |
| MoPoE | NA | NA | 0.94 | NA | 0.36 | NA | 0.93 | NA | NA | | | | | | | | | |
| MMJSD | NA | NA | 0.95 | NA | 0.48 | NA | 0.92 | NA | NA | | | | | | | | | |
| MVTCAE (w/o T) | NA | NA | NA | NA | NA | NA | NA | NA | NA | | | | | | | | | |

Table 9: Conditional coherence for models with private latent variables and uni-modal conditionals. The letters on the second line represent the modality which is generated based on the sets of modalities on the line below it.

| | Proposed objective | | | | | | | | | Mixture bound | | | | | | | | |
|---|---|---|---|---|---|---|---|---|---|---|---|---|---|---|---|---|---|---|
| | M | | | S | | | T | | | M | | | S | | | T | | |
| Aggregation | M | S | T | M | S | T | M | S | T | M | S | T | M | S | T | M | S | T |
| PoE+ | 0.97 | 0.12 | 0.13 | 0.20 | 0.62 | 0.24 | 0.16 | 0.15 | **1.00** | 0.96 | 0.83 | **0.99** | 0.11 | 0.58 | 0.11 | 0.44 | 0.39 | 1.00 |
| SumPooling | 0.97 | 0.42 | 0.59 | **0.44** | 0.67 | **0.40** | **0.65** | **0.45** | **1.00** | 0.97 | **0.86** | **0.99** | 0.11 | 0.62 | 0.11 | 0.45 | 0.40 | 1.00 |
| SelfAttention | 0.97 | 0.12 | 0.12 | 0.27 | **0.71** | 0.28 | 0.46 | 0.40 | 1.00 | 0.96 | 0.09 | 0.08 | 0.12 | 0.67 | 0.12 | 0.15 | 0.17 | 1.00 |

Table 10: Conditional coherence for models with private latent variables and bi-modal conditionals. The letters on the second line represent the modality, which is generated based on the sets of modalities on the line below it.

| | Proposed objective | | | | | | | | | Mixture bound | | | | | | | | |
| --- | --- | --- | --- | --- | --- | --- | --- | --- | --- | --- | --- | --- | --- | --- | --- | --- | --- | --- |
| | M | | | S | | | T | | | M | | | S | | | T | | |
| Aggregation | M+S | M+T | S+T | M+S | M+T | S+T | M+S | M+T | S+T | M+S | M+T | S+T | M+S | M+T | S+T | M+S | M+T | S+T |
| PoE+ | **0.97** | **0.97** | 0.14 | 0.66 | 0.33 | 0.67 | 0.18 | **1.00** | **1.00** | **0.97** | **0.97** | **0.94** | 0.63 | 0.11 | 0.63 | 0.45 | 0.99 | 0.96 |
| SumPooling | **0.97** | **0.97** | 0.54 | 0.79 | **0.43** | 0.80 | **0.57** | **1.00** | **1.00** | **0.97** | **0.97** | 0.93 | 0.64 | 0.11 | 0.63 | 0.45 | 0.99 | 0.97 |
| SelfAttention | **0.97** | **0.97** | 0.12 | **0.80** | 0.29 | **0.81** | 0.49 | **1.00** | **1.00** | 0.96 | 0.96 | 0.08 | 0.70 | 0.12 | 0.70 | 0.15 | **1.00** | **1.00** |

Table 11: Conditional coherence for models with shared latent variables for different $\beta$s and uni-modal conditionals. The letters on the second line represent the modality which is generated based on the sets of modalities on the line below it.

| | Proposed objective | | | | | | | | | Mixture bound | | | | | | | | |
| --- | --- | --- | --- | --- | --- | --- | --- | --- | --- | --- | --- | --- | --- | --- | --- | --- | --- | --- |
| | M | | | S | | | T | | | M | | | S | | | T | | |
| ($\beta$, Aggregation) | M | S | T | M | S | T | M | S | T | M | S | T | M | S | T | M | S | T |
| (0.1, PoE+) | **0.98** | 0.11 | 0.12 | 0.12 | 0.62 | 0.14 | 0.61 | 0.25 | **1.00** | 0.96 | 0.83 | **0.99** | 0.11 | 0.58 | 0.11 | 0.45 | 0.39 | **1.00** |
| (0.1, SumPooling) | 0.97 | 0.48 | 0.81 | 0.30 | **0.72** | 0.33 | **0.86** | **0.55** | **1.00** | 0.97 | 0.86 | **0.99** | 0.11 | 0.64 | 0.11 | 0.45 | 0.40 | **1.00** |
| (1.0, PoE+) | 0.97 | 0.15 | 0.63 | 0.24 | 0.63 | 0.42 | 0.79 | 0.35 | **1.00** | 0.96 | 0.83 | **0.99** | 0.11 | 0.59 | 0.11 | 0.45 | 0.39 | **1.00** |
| (1.0, SumPooling) | 0.97 | 0.48 | 0.87 | 0.25 | **0.72** | 0.36 | 0.73 | 0.48 | **1.00** | 0.97 | **0.86** | **0.99** | 0.10 | 0.63 | 0.10 | 0.45 | 0.40 | **1.00** |
| (4.0, PoE+) | 0.97 | 0.29 | 0.83 | **0.41** | 0.60 | **0.58** | 0.76 | 0.38 | **1.00** | 0.96 | 0.82 | **0.99** | 0.10 | 0.57 | 0.10 | 0.44 | 0.38 | **1.00** |
| (4.0, SumPooling) | 0.97 | 0.48 | 0.88 | 0.35 | 0.66 | 0.44 | 0.83 | 0.53 | **1.00** | 0.96 | 0.85 | **0.99** | 0.11 | 0.57 | 0.10 | 0.45 | 0.39 | **1.00** |

Table 12: Conditional coherence for models with shared latent variables for different $\beta$s and bi-modal conditionals. The letters on the second line represent the modality, which is generated based on the sets of modalities on the line below it.

| | Proposed objective | | | | | | | | | Mixture bound | | | | | | | | |
| --- | --- | --- | --- | --- | --- | --- | --- | --- | --- | --- | --- | --- | --- | --- | --- | --- | --- | --- |
| | M | | | S | | | T | | | M | | | S | | | T | | |
| ($\beta$, Aggregation) | M+S | M+T | S+T | M+S | M+T | S+T | M+S | M+T | S+T | M+S | M+T | S+T | M+S | M+T | S+T | M+S | M+T | S+T |
| (0.1, PoE+) | **0.98** | **0.98** | 0.15 | 0.70 | 0.14 | 0.72 | 0.66 | **1.00** | **1.00** | 0.96 | 0.96 | 0.93 | 0.62 | 0.11 | 0.62 | 0.45 | 0.99 | 0.95 |
| (0.1, SumPooling) | 0.97 | 0.97 | 0.86 | **0.83** | 0.31 | **0.84** | 0.85 | 0.99 | **1.00** | 0.97 | 0.97 | 0.94 | 0.66 | 0.11 | 0.65 | 0.45 | 0.99 | 0.96 |
| (1.0, PoE+) | 0.97 | **0.98** | 0.55 | 0.73 | 0.52 | 0.75 | 0.83 | **1.00** | 0.99 | 0.97 | 0.97 | **0.96** | 0.64 | 0.11 | 0.63 | 0.45 | 0.99 | 0.97 |
| (1.0, SumPooling) | 0.97 | 0.97 | 0.86 | 0.78 | 0.30 | 0.80 | 0.76 | 0.99 | **1.00** | 0.97 | 0.97 | 0.95 | 0.65 | 0.10 | 0.65 | 0.45 | 0.99 | 0.97 |
| (4.0, PoE+) | 0.97 | **0.98** | 0.84 | 0.76 | **0.66** | 0.78 | 0.82 | **1.00** | **1.00** | 0.97 | 0.97 | **0.96** | 0.62 | 0.10 | 0.62 | 0.45 | 0.99 | 0.98 |
| (4.0, SumPooling) | 0.97 | 0.97 | 0.89 | 0.77 | 0.40 | 0.78 | **0.86** | 0.99 | **1.00** | 0.97 | 0.97 | **0.96** | 0.61 | 0.10 | 0.60 | 0.45 | 0.99 | 0.97 |

## L.6  Latent classification accuracy

Table 13: Unsupervised latent classification for $\beta = 1$ and models with shared latent variables only (top half) and shared plus private latent variables (bottom half). Accuracy is computed with a linear classifier (logistic regression) trained on multi-modal inputs (M+S+T) or uni-modal inputs (M, S or T).

| Aggregation | Proposed objective | | | | Mixture bound | | | |
|---|---|---|---|---|---|---|---|---|
| | M+S+T | M | S | T | M+S+T | M | S | T |
| PoE | 0.988 (0.000) | 0.940 (0.009) | 0.649 (0.039) | 0.998 (0.001) | 0.991 (0.004) | 0.977 (0.002) | 0.845 (0.000) | **1.000 (0.000)** |
| PoE+ | 0.978 (0.002) | 0.934 (0.001) | 0.624 (0.040) | 0.999 (0.001) | **0.998 (0.000)** | 0.981 (0.000) | 0.851 (0.000) | **1.000 (0.000)** |
| MoE | 0.841 (0.008) | 0.974 (0.000) | 0.609 (0.032) | **1.000 (0.000)** | 0.940 (0.001) | 0.980 (0.001) | 0.843 (0.001) | **1.000 (0.000)** |
| MoE+ | 0.850 (0.039) | 0.967 (0.014) | 0.708 (0.167) | 0.983 (0.023) | 0.928 (0.017) | 0.983 (0.002) | 0.846 (0.001) | **1.000 (0.000)** |
| SelfAttention | 0.985 (0.001) | 0.954 (0.002) | 0.693 (0.037) | 0.986 (0.006) | 0.991 (0.000) | 0.981 (0.001) | 0.864 (0.003) | **1.000 (0.000)** |
| SumPooling | 0.981 (0.000) | 0.962 (0.000) | 0.704 (0.014) | 0.992 (0.008) | 0.994 (0.000) | **0.983 (0.000)** | 0.866 (0.002) | **1.000 (0.000)** |
| PoE+ | 0.979 (0.009) | 0.944 (0.000) | 0.538 (0.032) | 0.887 (0.07) | 0.995 (0.002) | 0.980 (0.002) | 0.848 (0.006) | **1.000 (0.000)** |
| SumPooling | 0.987 (0.004) | 0.966 (0.004) | 0.370 (0.348) | 0.992 (0.002) | 0.994 (0.001) | 0.982 (0.000) | **0.870 (0.001)** | **1.000 (0.000)** |
| SelfAttention | 0.990 (0.003) | 0.968 (0.002) | 0.744 (0.008) | 0.985 (0.000) | 0.997 (0.001) | 0.974 (0.000) | 0.681 (0.031) | **1.000 (0.000)** |
| Results from Sutter et al. (2021), Sutter et al. (2020) and Hwang et al. (2021) | | | | | | | | |
| MVAE | 0.96 (0.02) | 0.90 (0.01) | 0.44 (0.01) | 0.85 (0.10) | | | | |
| MMVAE | 0.86 (0.03) | 0.95 (0.01) | 0.79 (0.05) | 0.99 (0.01) | | | | |
| MoPoE | 0.98 (0.01) | 0.95 (0.01) | 0.80 (0.03) | 0.99 (0.01) | | | | |
| MMJSD | 0.98 (NA) | 0.97 (NA) | 0.82 (NA) | 0.99 (NA) | | | | |
| MVTCAE (w/o T) | NA | 0.93 (NA) | 0.78 (NA) | NA | | | | |

Table 14: Unsupervised latent classification for different $\beta$s and models with shared latent variables only. Accuracy is computed with a linear classifier (logistic regression) trained on multi-modal inputs (M+S+T) or uni-modal inputs (M, S or T).

| ($\beta$, Aggregation) | Proposed objective | | | | Mixture bound | | | |
|---|---|---|---|---|---|---|---|---|
| | M+S+T | M | S | T | M+S+T | M | S | T |
| (0.1, PoE+) | 0.983 (0.006) | 0.919 (0.001) | 0.561 (0.048) | 0.988 (0.014) | 0.992 (0.002) | 0.979 (0.002) | 0.846 (0.004) | **1.000 (0.000)** |
| (0.1, SumPooling) | 0.982 (0.004) | 0.965 (0.002) | 0.692 (0.047) | 0.999 (0.001) | 0.994 (0.000) | 0.981 (0.002) | 0.863 (0.005) | **1.000 (0.000)** |
| (1.0, PoE+) | 0.978 (0.002) | 0.934 (0.001) | 0.624 (0.040) | 0.999 (0.001) | **0.998 (0.000)** | 0.981 (0.000) | 0.851 (0.000) | **1.000 (0.000)** |
| (1.0, SumPooling) | 0.981 (0.000) | 0.962 (0.000) | 0.704 (0.014) | 0.992 (0.008) | 0.994 (0.000) | **0.983 (0.000)** | **0.866 (0.002)** | **1.000 (0.000)** |
| (4.0, PoE+) | 0.981 (0.006) | 0.943 (0.007) | 0.630 (0.008) | 0.993 (0.001) | **0.998 (0.000)** | 0.981 (0.000) | 0.846 (0.001) | **1.000 (0.000)** |
| (4.0, SumPooling) | 0.984 (0.004) | 0.963 (0.001) | 0.681 (0.009) | 0.995 (0.000) | 0.992 (0.002) | 0.980 (0.001) | 0.856 (0.001) | **1.000 (0.000)** |

## M Encoder Model architectures

### M.1 Linear models

Table 15: Encoder architectures for Gaussian models.

(a) Modality-specific encoding functions $h_s(x_s)$. Latent dimension $D = 30$, modality dimension $D_s \sim \mathcal{U}(30, 60)$.

| MoE/PoE | SumPooling/SelfAttention |
|---|---|
| Input: $D_s$ | Input: $D_s$ |
| Dense $D_s \times 512$, ReLU | Dense $D_s \times 256$, ReLU |
| Dense $512 \times 512$, ReLU | Dense $256 \times 256$, ReLU |
| Dense $512 \times 60$ | Dense $256 \times 60$ |

(b) Model for outer aggregation function $\rho_\vartheta$ for SumPooling and SelfAttention schemes.

| Outer Aggregation |
|---|
| Input: 256 |
| Dense $256 \times 256$, ReLU |
| Dense $256 \times 256$, ReLU |
| Dense $256 \times 60$ |

(c) Inner aggregation function $\chi_\vartheta$.

| SumPooling | SelfAttention |
|---|---|
| Input: 256 | Input: 256 |
| Dense $256 \times 256$, ReLU | Dense $256 \times 256$, ReLU |
| Dense $256 \times 256$, ReLU | Dense $256 \times 256$ |
| Dense $256 \times 256$ | |

(d) Transformer parameters.

| SelfAttention (1 Layer) |
|---|
| Input: 256 |
| Heads: 4 |
| Attention size: 256 |
| Hidden size FFN: 256 |

### M.2 Linear models with private latent variables

Table 16: Encoder architectures for Gaussian models with private latent variables.

(a) Modality-specific encoding functions $h_s(x_s)$. All private and shared latent variables are of dimension 10. Modality dimension $D_s \sim \mathcal{U}(30, 60)$.

| PoE ($h_s^{\text{shared}}$ and $h_s^{\text{private}}$) | SumPooling/SelfAttention |
|---|---|
| Input: $D_s$ | Input: $D_s$ |
| Dense $D_s \times 512$, ReLU | Dense $D_s \times 128$, ReLU |
| Dense $512 \times 512$, ReLU | Dense $128 \times 128$, ReLU |
| Dense $512 \times 10$ | Dense $128 \times 10$ |

(b) Model for outer aggregation function $\rho_\vartheta$ for SumPooling scheme.

| Outer Aggregation ($\rho_\vartheta$) |
|---|
| Input: 128 |
| Dense $128 \times 128$, ReLU |
| Dense $128 \times 128$, ReLU |
| Dense $128 \times 10$ |

(c) Inner aggregation functions.

| SumPooling ($\chi_{0,\vartheta}$, $\chi_{1,\vartheta}$, $\chi_{2,\vartheta}$) | SelfAttention ($\chi_{1,\vartheta}$, $\chi_{2,\vartheta}$) |
|---|---|
| Input: 128 | Input: 128 |
| Dense $128 \times 128$, ReLU | Dense $128 \times 128$, ReLU |
| Dense $128 \times 128$, ReLU | Dense $128 \times 128$ |
| Dense $128 \times 128$ | |

(d) Transformer parameters.

| SelfAttention (1 Layer) |
|---|
| Input: 128 |
| Heads: 4 |
| Attention size: 128 |
| Hidden size FFN: 128 |

## M.3 Nonlinear model with auxiliary label

Table 17: Encoder architectures for nonlinear model with auxiliary label.

(a) Modality-specific encoding functions $h_s(x_s)$. Modality dimension $D_1 = 2$ (continuous modality) and $D_2 = 5$ (label). Embedding dimension $D_E = 4$ for PoE and MoE and $D_E = 128$ otherwise.

| Modality-specific encoders |
|---|
| Input: $D_s$ |
| Dense $D_s \times 128$, ReLU |
| Dense $128 \times 128$, ReLU |
| Dense $128 \times D_E$ |

(b) Model for outer aggregation function $\rho_\vartheta$ for SumPooling and SelfAttention schemes and mixtures thereof. Output dimension is $D_0 = 25$ for mixture densities and $D_O = 4$ otherwise.

| Outer Aggregation |
|---|
| Input: 128 |
| Dense $128 \times 128$, ReLU |
| Dense $128 \times 128$, ReLU |
| Dense $128 \times D_O$ |

(c) Inner aggregation function $\chi_\vartheta$.

| SumPooling | SelfAttention |
|---|---|
| Input: 128 | Input: 128 |
| Dense $128 \times 128$, ReLU | Dense $128 \times 128$, ReLU |
| Dense $128 \times 128$, ReLU | Dense $128 \times 128$ |
| Dense $128 \times 128$ | |

(d) Transformer parameters.

| SelfAttention |
|---|
| Input: 128 |
| Heads: 4 |
| Attention size: 128 |
| Hidden size FFN: 128 |

## M.4 Nonlinear model with five modalities

Table 18: Encoder architectures for a nonlinear model with five modalities.

(a) Modality-specific encoding functions $h_s(x_s)$. Modality dimensions $D_s = 25$. Latent dimension $D = 25$

| MoE/PoE | SumPooling/SelfAttention |
|---|---|
| Input: $D_s$ | Input: $D_s$ |
| Dense $D_s \times 512$, ReLU | Dense $D_s \times 256$, ReLU |
| Dense $512 \times 512$, ReLU | Dense $256 \times 256$, ReLU |
| Dense $512 \times 50$ | Dense $256 \times 256$ |

(b) Model for outer aggregation function $\rho_\vartheta$ for SumPooling and SelfAttention schemes and mixtures thereof. Output dimension is $D_0 = 50$ for mixture densities and $D_O = 25$ otherwise.

| Outer Aggregation |
|---|
| Input: 256 |
| Dense $256 \times 256$, ReLU |
| Dense $256 \times 256$, ReLU |
| Dense $256 \times D_O$ |

(c) Inner aggregation function $\chi_\vartheta$.

| SumPooling | SelfAttention |
|---|---|
| Input: 256 | Input: 256 |
| Dense $256 \times 256$, ReLU | Dense $256 \times 256$, ReLU |
| Dense $256 \times 256$, ReLU | Dense $\times 256$ |
| Dense $256 \times 256$ | |

(d) Transformer parameters.

| SelfAttention |
|---|
| Input: 256 |
| Heads: 4 |
| Attention size: 256 |
| Hidden size FFN: 256 |

## M.5 MNIST-SVHN-Text

For SVHN and Text, we use 2d- or 1d-convolutional layers, respectively, denoted as $\mathrm{Conv}(f, k, s)$ for feature dimension $f$, kernel-size $k$, and stride $s$. We denote transposed convolutions as tConv. We use the neural network architectures as implemented in Flax Heek et al. (2023).

Table 19: Encoder architectures for MNIST-SVHN-Text.

(a) MNIST-specific encoding functions $h_s(x_s)$. Modality dimensions $D_s = 28 \times 28$. The embedding dimension is $D_E = 2D$ for PoE/MoE and $D_E = 256$ for SumPooling/SelfAttention. For PoE+/MoE+, we add four times a Dense layer of size 256 with ReLU layer before the last linear layer.

| MoE/PoE/SumPooling/SelfAttention |
| --- |
| Input: $D_s$, |
| Dense $D_s \times 400$, ReLU |
| Dense $400 \times 400$, ReLU |
| Dense $400 \times D_E$ |

(b) SVHN-specific encoding functions $h_s(x_s)$. Modality dimensions $D_s = 3 \times 32 \times 32$. The embedding dimension is $D_E = 2D$ for PoE/MoE and $D_E = 256$ for SumPooling/SelfAttention. For PoE+/MoE+, we add four times a Dense layer of size 256 with ReLU layer before the last linear layer.

| MoE/PoE/SumPooling/SelfAttention |
| --- |
| Input: $D_s$ |
| Conv(32, 4, 2), ReLU |
| Conv(64, 4, 2), ReLU |
| Conv(64, 4, 2), ReLU |
| Conv(128, 4, 2), ReLU, Flatten |
| Dense $2048 \times D_E$ |

(c) Text-specific encoding functions $h_s(x_s)$. Modality dimensions $D_s = 8 \times 71$. Embedding dimension is $D_E = 2D$ for PoE/MoE and $D_E = 256$ for permutation-invariant models (SumPooling/SelfAttention) and $D_E = 128$ for permutation-equivariant models (SumPooling/SelfAttention). For PoE+/MoE+, we add four times a Dense layer of size 256 with ReLU layer before the last linear layer.

| MoE/PoE/SumPooling/SelfAttention |
| --- |
| Input: $D_s$ |
| Conv(128, 1, 1), ReLU |
| Conv(128, 4, 2), ReLU |
| Conv(128, 4, 2), ReLU, Flatten |
| Dense $128 \times D_E$ |

(d) Model for outer aggregation function $\rho_\vartheta$ for SumPooling and SelfAttention schemes. Output dimension is $D_0 = 2D = 80$ for models with shared latent variables only and $D_0 = 10 + 10$ for models with private and shared latent variables. $D_E = 256$ for permutation-invariant and $D_I = 128$ for permutation-invariant models.

| Outer Aggregation |
| --- |
| Input: $D_E$ |
| Dense $D_E \times D_E$, LReLU |
| Dense $D_E \times D_E$, LReLU |
| Dense $D_E \times D_O$ |

(e) Inner aggregation function $\chi_\vartheta$ for permutation-invariant models ($D_E = 256$) and permutaion-equivariant models ($D_E = 128$).

| SumPooling | SelfAttention |
| --- | --- |
| Input: $D_E$ | Input: $D_E$ |
| Dense $D_E \times D_E$, LReLU | Dense $D_E \times D_E$, LReLU |
| Dense $D_E \times D_E$, LReLU | Dense $\times D_E$ |
| Dense $D_E \times D_E$ | |

(f) Transformer parameters for permutation-invariant models. $D_E = 256$ for permutation-invariant and $D_I = 128$ for permutation-invariant models.

| SelfAttention (2 Layers) |
| --- |
| Input: $D_E$ |
| Heads: 4 |
| Attention size: $D_E$ |
| Hidden size FFN: $D_E$ |

# N    MNIST-SVHN-Text Decoder Model architectures

For models with private latent variables, we concatenate the shared and private latent variables. We use a Laplace likelihood as the decoding distribution for MNIST and SVHN, where the decoder function learns both its mean as a function of the latent and a constant log-standard deviation at each pixel. Following previous works (Shi et al., 2019; Sutter et al., 2021), we re-weight the log-likelihoods for different modalities relative to their dimensions.

Table 20: Decoder architectures for MNIST-SVHN-Text.

(a) MNIST decoder. $D_I = 40$ for models with shared latent variables only, and $D_I = 10 + 10$ otherwise.

| MNIST |
| --- |
| Input: $D_I$ |
| Dense $40 \times 400$, ReLU |
| Dense $400 \times 400$, ReLU |
| Dense $400 \times D_s$, Sigmoid |

(b) SVHN decoder. $D_I = 40$ for models with shared latent variables only, and $D_I = 10 + 10$ otherwise.

| SVHN |
| --- |
| Input: $D_I$ |
| Dense $D_I \times 128$, ReLU |
| tConv(64, 4, 3), ReLU |
| tConv(64, 4, 2), ReLU |
| tConv(32, 4, 2), ReLU |
| tConv(3, 4, 2) |

(c) Text decoder. $D_I = 40$ for models with shared latent variables only, and $D_I = 10 + 10$ otherwise.

| Text |
| --- |
| Input: $D_I$ |
| Dense $D_I \times 128$, ReLU |
| tConv(128, 4, 3), ReLU |
| tConv(128, 4, 2), ReLU |
| tConv(71, 1, 1) |

## O   Compute resources and existing assets

A reference implementation is available at `https://github.com/marcelah/MaskedMultimodalVAE`. Our computations were performed on shared HPC systems. All experiments except Section 5.3 were run on a CPU server using one or two CPU cores. The experiments in Section 5.3 were run on a GPU server using one NVIDIA A100.

Our implementation is based on JAX (Bradbury et al., 2018) and Flax (Heek et al., 2023). We compute the mean correlation coefficient (MCC) between true and inferred latent variables following Khemakhem et al. (2020b), as in `https://github.com/ilkhem/icebeem` and follow the data and model generation from Khemakhem et al. (2020a), `https://github.com/ilkhem/iVAE` in Section 5.2, as well as `https://github.com/hanmenghan/CPM_Nets` from Zhang et al. (2019) for generating the missingness mechanism. In our MNIST-SVHN-Text experiments, we use code from Sutter et al. (2021), `https://github.com/thomassutter/MoPoE`.

