# OpenReview forum: "Learning multi-modal generative models with permutation-invariant encoders and tighter variational objectives"
_TMLR — Accepted by TMLR_

### Review · Reviewer_S2Lb · 2024-06-17

**Summary Of Contributions:**

The authors propose a new variational objective for fitting Multimodal VAEs. In contrast to existing mixture-based bounds, the proposed objective addresses a gap that arises in the mixture-based bounds due to modality subsampling. This is tackled by deriving a variational objective that resembles a lower-bound to the marginal log-likelihood and that, subject to some conditions, can be shown to be equal to the marginal log-likelihood. Moreover, the authors propose learnable permutation-invariant approaches to parametrise the variational distribution, which mitigates the exponential growth of variational distributions needed due to the subsampling of the modalities. The method is evaluated on some synthetic and realistic multimodal datasets and is shown to achieve better log-likelihood and other related quantities.

**Audience:**

Yes

**Broader Impact Concerns:**

None. The method presents a technical contribution for multimodal VAE training and the broader impacts are no different from a standard VAE model.

**Claims And Evidence:**

No

**Requested Changes:**

Please see the weaknesses above. The important changes/questions are under "main comments/questions". But, I hope the other comments may help improve the overall clarity/correctess of the paper and should be easy to correct.

**Strengths And Weaknesses:**

## Strengths

* The paper is on an important task of fitting VAEs on data from multiple modalities (e.g. text/images). The authors address a previously-identified gap in the mixture-based variational objectives for fitting multimodal VAEs, by proposing a new objective that is motivated by its ability to reduce this gap. (Although this motivation is slightly burried in the presentation and needs a bit more work to make it concrete, see weaknesses.)
* Noting that some previous approaches used permutation-invariant variational distributions, the authors proposed a few more approaches that aim to induce this invariance. The approaches themselves are not particularly new, but seem new in this context.
* The evaluation seems to support the authors' hypothesis that the proposed approach (using the proposed objective and variational distributions) achieves higher log-likelihood than some existing approaches.

## Weaknesses

### Main comments/questions

* The proposed objective $\mathcal{L}(x, \theta, \phi, \beta)$ in Section 2 is introduced without derivation. Although the derivation is straightforward, including the derivation where the objective is introduced would clarify that the objective is not truly a lower-bound on the log-likelihood. Consequently, I think calling the objective throughout the paper (and in the title) an "approximate bound" can be misleading as it invites potentially erroneous mis-uses of it.
* A key contribution of the paper, as stated by the authors, is a _tighter_ variational bound/objective. But, until Remark 5 (end of Section 2.1), it is not clearly explained which existing objective is the proposed objective tighter than. (Except for lightly touching upon it with one sentence in the Contributions paragraph in Page 2.) This leaves the reader wondering what is it that the paper solves until the end of Section 2.1. The background information provided in Remark 5 should be discussed with the rest of the background in Section 1, and the advantages of the objective should be discussed in more detail.
* Assuming that the variational distribution exactly approximates the model's posterior, Remark 5 explains that their objective is tighter than the existing Multimodal VAE objectives, where the gap arises due to sub-sampling of the modalities. However, the case where the variational approximation is not accurate is not discussed. Moreover, the Jensen's gap, alluded to at the end of Remark 5, is also not worked out. Explaining these two points would greatly help their contribution regarding the tightness/advantages of their objective.
* Remark 3 states that maximising $\mathcal{L}_{\setminus \mathcal{S}}$ yields $q\_\phi(z \mid x\_{\mathcal{S}}) = q^{agg}\_{\phi, \setminus \mathcal{S}}(z \mid x\_\mathcal{S})$ because of equation $\texttt{Z}\_{\text{conditional}}$. Firstly, the text should describe what quantity/parameters are you maximising $\mathcal{L}\_{\setminus \mathcal{S}}$ _with respect to_. (Is it $\phi$?) Secondly, if you maximise the expected $\mathcal{L}\_{\setminus \mathcal{S}}$ in equation $\texttt{Z}\_{\text{conditional}}$ w.r.t. the parameters of the variational distribution $\phi$, the optimal $q\_\phi(z \mid x\_{\mathcal{S}})$ may not be equal to $q^{agg}\_{\phi, \setminus \mathcal{S}}(z \mid x\_\mathcal{S})$. This is because the parameters $\phi$ appear in both terms of equation $\texttt{Z}\_{\text{conditional}}$, and hence maximisation of the equation w.r.t. $\phi$ will likely strike some balance between the two terms in the equation, i.e. $q\_\phi(z \mid x\_{\mathcal{S}}) \not= q^{agg}\_{\phi, \setminus \mathcal{S}}(z \mid x\_\mathcal{S})$.
* The discussion below equation 10 in Section 3.2, suggests that due to factorisation of the model $p\_\theta(z', \tilde z\_{\mathcal{S}} \mid x\_\mathcal{S}) p\_\theta(\tilde z\_{\setminus \mathcal{S}} \mid z',  \tilde z\_{\mathcal{S}})$, the variational distribution $q\_\phi(z', \tilde z\_{\mathcal{S}}, \tilde z\_{\mathcal{\setminus S}} \mid x\_{\mathcal{S}})$ should also similarly factorise as $q\_\phi(z', \tilde z\_{\mathcal{S}} \mid x\_{\mathcal{S}}) q\_\phi(\tilde z\_{\setminus \mathcal{S}} \mid z', \tilde z\_{\mathcal{S}})$. But, although it may be a useful inductive bias and/or simplification, this may not necessarily be true if we aim to accurately approximate the posterior $p\_\theta(z', \tilde z\_{\mathcal{S}}, \tilde z\_{\mathcal{\setminus S}} \mid x\_{\mathcal{S}})$. For example, in case $q\_\phi(z', \tilde z\_{\mathcal{S}} \mid x\_{\mathcal{S}}) \not= p\_\theta(z', \tilde z\_{\mathcal{S}} \mid x\_\mathcal{S})$, the factorisation $q\_\phi(z', \tilde z\_{\mathcal{S}} \mid x\_{\mathcal{S}}) q\_\phi(\tilde z\_{\setminus \mathcal{S}} \mid z', \tilde z\_{\mathcal{S}}, x\_{\mathcal{S}})$ may be able to better approximate the model posterior, because conditioning $q\_\phi(\tilde z\_{\setminus \mathcal{S}} \mid z', \tilde z\_{\mathcal{S}}, x\_{\mathcal{S}})$ on $x\_{\mathcal{S}}$ allows the distribution to some extent compensate for the inaccuracies (or loss of information) in $q\_\phi(z', \tilde z\_{\mathcal{S}} \mid x\_{\mathcal{S}})$.
* As explained in the paragraph preceding Examples 3-5 (after equation 11), the distribution $p\_\theta(\tilde z\_{\setminus \mathcal{S}} \mid z',  \tilde z\_{\mathcal{S}})$ is often analytically tractable. So why do Examples 3-5 not use it? Specifically, Example 3 instead uses $\prod\_{s \in \mathcal{M}\setminus\mathcal{S}} p\_{\theta}(\tilde z\_s)$ and Examples 4-5 instead use $\prod\_{s \in \mathcal{M}\setminus\mathcal{S}} p\_{\theta}(\tilde z\_s \mid z')$. Is this a typo?
* The authors provided permutation-equivariant versions of PoE, SumPooling, and Self-Attention, but did not discuss permutation-equivariant versions of MoE or MoPoE. Is there a reason why this may not be possible with these aggregation schemes, are they incompatible with permutation-equivariance? Would be good to discuss this.
* The evaluations use quantities such as "Full reconstruction", "Full rates", "Cross reconstruction", and "Cross rates", but the effect of these metrics going up/down is not clearly motivated or explained in the discussion of the results, thus it is unclear how important are they for multimodal VAEs and their contribution. Some interpretation of these metrics and their importance to multimodal VAEs would be helpful.
* In the MNIST-SVHN-Text experiments, the total number of modalities is 3. You could thus parametrise all possible permutations of modalities (8 of them in total) using a separate encoder (one per permutation). This would serve as a strong baseline to verify how good the permutation-invariant parametrisations really are.
* In Section 5.3, it seems that using the proposed objective improves cross-generation of the SVHN modality (Figure 2) but decreases cross-reconstruction (Figure 3). Why is that?

### Presentation clarity

I have also found the presentation of the paper slightly lacking in clarity and coherence (in addition to some of the points discussed above). Below are some examples:

* Relevance of the discussion about VampPrior in Remark 3 is not clear. VampPrior is not used in the paper and its use in this setting may be tricky. This is because the goal is to approximate $p\_\theta(z \mid x\_{\mathcal{S}})$. Meaning if we wanted to use VampPrior $p^{\text{Vamp}}(z \mid \mathcal{D})$ of the form $\frac{1}{K}\sum\_{k=1}^K q\_\phi(z \mid u\_k)$ to approximate $p\_\theta(z \mid x\_{\mathcal{S}})$ we would have to learn $K$ pseudo-inputs $u\_k$ for each $x\_{\mathcal{S}}$ for all $\mathcal{S} \subset \mathcal{M}$ and $\mathcal{x}$, which would scale exponentially in the number of modalities $\mathcal{M}$. So it is unclear how the discussion helps the paper.
* Remark 4 presents some hypotheses that are not experimentally validated or reflected upon in the rest of the paper, so it is unclear how relevant it is.
* Remarks 9, 10, 11, and 13 present some results/discussion that seem out-of-context. Why are they relevant to the paper?
* In Table 2, the numbers in parentheses are not explained. Are they standard deviations/errors, how many independent runs?
* The tables in the evaluations were a little hard to decipher, this is because "ours" is used to denote the proposed bound, but the proposed aggregation schemes (SumPooling/SelfAttention) weren't highlighted. Denoting the proposed bound as "our bound" or "proposed bound" and appending the proposed scheme names with "(proposed)" may help.

### Typos

There is a bunch of (minor) typos in the paper. Here's a list of some:

* Below Eq 2., the mutual information equation is missing $\mathrm{d}x\mathrm{d}y$.
* Equation $\texttt{X}\_{\text{conditional}}$ should have a minus before the term $\int q\_\phi(z \mid x) \log \frac{q\_\phi(z \mid x\_{\mathcal{S}})}{p\_\theta(z \mid x\_{\mathcal{S}})}\mathrm{d}z$ instead of a plus.
* The text after Corollary 2 should be $\mathcal{L}\_{\setminus \mathcal{S}}$ instead of $\mathcal{L}\_{\setminus \mathcal{L}}$.
* Reference links to Corollary 19 (Appendix) should be links to Corollary 2 (main text).
* The VampPrior in equation 5 is missing $1/K$.
* Page 7, missing $\mathcal{S}$ in the text after the fifth equation on the page, as in "... for any $\mathcal{S}$ ...".
* Corollary 8, "minimizing $\mathcal{L}$" should be "maximizing ...".
* Remark 9, "minimizing $\mathcal{L}_{\mathcal{S}}^{\text{Mix}}$" should be "maximizing ...".
* Page 9, The equation for MoPoE variational distribution should have $q\_\phi^{\text{PoE}}$ inside the summation instead of $q\_\phi^{\text{MoE}}$.
* Start of Section 5 states that the proposed bound will be called "masked" in the following figures, but all of the figures call it "ours", with the exception of Figure 3.
* In Table 1 the wrong results are highlighted as best in the Full-rates column. It should be the cell with 1.02.
* Page 33, the second equality following lower-bound derivation for $I\_{q\_\phi}(X\_{\setminus \mathcal{S}}, Z\_{\mathcal{M}} \mid X\_{\mathcal{S}})$ is missing a $\mathrm{d}x\_{\setminus \mathcal{S}}$ after the $\log p\_\theta(x\_{\setminus \mathcal{S}} \mid z)$ term.
* Page 33, the first equality following the upper-bound derivation for $I\_{q\_\phi}(X\_{\setminus \mathcal{S}}, Z\_{\mathcal{M}} \mid X\_{\mathcal{S}})$ is missing a $q\_\phi(z \mid x)$ after the $p_d(x\_{\setminus \mathcal{S}} \mid x\_\mathcal{S})$ term. (Also, the term $p_d(x\_{\setminus \mathcal{S}} \mid x\_\setminus)$ should be $p\_d(x\_{\setminus \mathcal{S}} \mid x\_\mathcal{S})$.)
* In Remark 22, the sign before $\beta \log p\_\theta(z)$ should be a plus (in two places).

---

> ### Author Response · Authors · 2024-07-22
>
> We thank the reviewer for taking the time to review our manuscript and for the helpful comments and feedback. We believe that incorporating the constructive comments and questions from the reviewer will improve our manuscript.
>
>
>
> >The proposed objective $\mathcal{L}(x, \theta, \phi, \beta)$ in Section 2 is introduced without derivation. Although the derivation is straightforward, including the derivation where the objective is introduced would clarify that the objective is not truly a lower-bound on the log-likelihood. Consequently, I think calling the objective throughout the paper (and in the title) an "approximate bound" can be misleading as it invites potentially erroneous mis-uses of it.
>
>
> We agree that the introduction of the variational objective was not well motivated in our initial submission, and its properties (including the fact that it is not a true lower bound on the log likelihood) are explained much later. We have now rewritten the introduction in Section 2 to address this better and that the property of lower bounding the log-evidence hinges on the approximation that the encoding distribution is close to the true posterior. We also agree that "approximate bound" can be misleading. We, therefore, tried to avoid the term "approximate bound" in the revised version, referring to it merely as a variational objective.
>
>
> >A key contribution of the paper, as stated by the authors, is a tighter variational bound/objective. But, until Remark 5 (end of Section 2.1), it is not clearly explained which existing objective is the proposed objective tighter than. (Except for lightly touching upon it with one sentence in the Contributions paragraph in Page 2.) This leaves the reader wondering what is it that the paper solves until the end of Section 2.1. The background information provided in Remark 5 should be discussed with the rest of the background in Section 1, and the advantages of the objective should be discussed in more detail.
>
> This is a very good point. Indeed, we mentioned the gap in the mixture-based bound only in Remark 5. We have now moved this to the background in Section 1, along with some rationale for this gap that results from trying to reconstruct or predict all modalities from incomplete information where some modalities are missing. At the beginning of Section 2, we provide now more details on how our objective is to avoid this issue and the resulting advantages. We hope this improves the presentation clarity.

---

> ### Author Response · Authors · 2024-07-22
>
> >Assuming that the variational distribution exactly approximates the model's posterior, Remark 5 explains that their objective is tighter than the existing Multimodal VAE objectives, where the gap arises due to sub-sampling of the modalities. However, the case where the variational approximation is not accurate is not discussed. Moreover, the Jensen's gap, alluded to at the end of Remark 5, is also not worked out. Explaining these two points would greatly help their contribution regarding the tightness/advantages of their objective.
>
>
> We thank the reader for asking to clarify the Jensen gap that we mentioned in our initial submission. We realized there was an error on our side, and the Jensen gap does not apply as stated in the same form as in the mentioned any-order methods. While a Jensen gap arises naturally if we consider a model where the probability of the multi-modal data $p_\theta(x|\mathcal{S})=p_\theta(x_\mathcal{S})p_\theta(x_{\setminus \mathcal{S}}|x_\mathcal{S})$
> depends on the choice of the modality mask $\mathcal{S}$, e.g. by directly specifying the model 'autoregressively' over two steps. Indeed, optimizing a training objective that randomises the model log-likelihood over the different masks, i.e.
> $$ \int \rho(\mathcal{S})[\log p_{\theta}(x_{\mathcal{S}})+ \log p_\theta(x_{\setminus \mathcal{S}}|x_\mathcal{S})] d \mathcal{S}
> =\int \rho(\mathcal{S}) \log p_\theta(x|\mathcal{S}) d \mathcal{S}
> \leq \log \int \rho(\mathcal{S})  p_\theta(x|\mathcal{S}) d \mathcal{S} = \log p_\theta(x)  $$
> is a lower bound via Jensen's inequality on the log-likelihood of
> $$p_\theta(x) = \int \rho(\mathcal{S})  p_\theta(x|\mathcal{S}) d \mathcal{S}, $$
> which is the marginal likelihood of a latent variable model wherein the mask $\mathcal{S}$ is a latent variable. However, in our case with a latent variable $Z$, the generative model for the multi-modal data $x$ does not depend on the choice of $\mathcal{S}$, i.e. $p_\theta(x)=p_\theta(x_{\mathcal{S}})p_\theta(x_{\setminus \mathcal{S}} |x_{\mathcal{S}})$ for any $\mathcal{S}$ so that the Jensen gap from above does not apply.
>
> We would also like to clarify that the gap in the mixture-based bound given by the conditional data entropy terms is in addition to gaps resulting from the variational approximation not being optimal.
>
> Empirically, we found that our variational objective is generally higher than the mixture-based bound. However, we refrained from reporting this as they are not directly comparable.
>
>
>
>
> >Remark 3 states that maximising $L_{\setminus \mathcal{S}}$ yields $q_{\phi}(z \mid x_{\mathcal{S}})= q_{\phi, \setminus \mathcal{S}}^{agg}(z \mid x_\mathcal{S})$ because of equation $Z_{\text{conditional}}$. Firstly, the text should describe what quantity/parameters are you maximising $L_{\setminus \mathcal{S}}$ with respect to. (Is it $\phi$?) Secondly, if you maximise the expected $L_{\setminus \mathcal{S}}$ in equation $Z_{\text{conditional}}$ w.r.t. the parameters of the variational distribution $\phi$, the optimal $q_\phi(z \mid x_{\mathcal{S}})$ may not be equal to $q_{\phi, \setminus \mathcal{S}}^{agg}(z \mid x_\mathcal{S})$. This is because the parameters $\phi$ appear in both terms of equation $Z_{\text{conditional}}$, and hence maximisation of the equation w.r.t. $\phi$ will likely strike some balance between the two terms in the equation, i.e. $q_\phi(z \mid x_{\mathcal{S}}) \not= q_{\phi, \setminus \mathcal{S}}^{agg}(z \mid x_\mathcal{S})$.
>
> This is an important point and our revision now provides a more detailed discussion in Remark 3 on the role of $q_\phi(z|x_\mathcal{S})$ that features both bounds. First, we maximize the conditional objective $L_{\setminus \mathcal{S}}$ with respect to both $\theta$ and $\phi$. Second, it is indeed the case that the variational parameters $\phi$ affect both objectives via $q_\phi(z|x_\mathcal{S})$ and the optimal $q_\phi(z|x_\mathcal{S})$ is generally attained at different values for both objectives. Our conditional objective may be slightly extended to the setting where one uses a different model for the conditional prior instead of $q_\phi(z|x_\mathcal{S})$ arising in $L_{\setminus \mathcal{S}}$ to avoid this coupling effect. While this choice may lead to higher values for the objectives, we did not explore this further and only considered our more constrained approach that encourages learning encoding models that perform well both as conditional priors and encoders. Actually, the same $q_\phi(z|x_{\mathcal{S}})$ optimizes both objectives at the same time only under idealised conditions, see Remark 3.

---

> > ### Author Response · Authors · 2024-07-22
> >
> > >The discussion below equation 10 in Section 3.2, suggests that due to factorisation of the model $p_\theta(z', \tilde z_{\mathcal{S}} \mid x_\mathcal{S}) p_\theta(\tilde z_{\setminus \mathcal{S}} \mid z', \tilde z_{\mathcal{S}})$, the variational distribution $q_\phi(z', \tilde z_{\mathcal{S}}, \tilde z_{\mathcal{\setminus S}} \mid x_{\mathcal{S}})$ should also similarly factorise as $q_\phi(z', \tilde z_{\mathcal{S}} \mid x_{\mathcal{S}}) q_\phi(\tilde z_{\setminus \mathcal{S}} \mid z', \tilde z_{\mathcal{S}})$. But, although it may be a useful inductive bias and/or simplification, this may not necessarily be true if we aim to accurately approximate the posterior $p_\theta(z', \tilde z_{\mathcal{S}}, \tilde z_{\mathcal{\setminus S}} \mid x_{\mathcal{S}})$. For example, in case $q_\phi(z', \tilde z_{\mathcal{S}} \mid x_{\mathcal{S}}) \not= p_\theta(z', \tilde z_{\mathcal{S}} \mid x_\mathcal{S})$, the factorisation $q_\phi(z', \tilde z_{\mathcal{S}} \mid x_{\mathcal{S}}) q_\phi(\tilde z_{\setminus \mathcal{S}} \mid z', \tilde z_{\mathcal{S}}, x_{\mathcal{S}})$ may be able to better approximate the model posterior, because conditioning $q_\phi(\tilde z_{\setminus \mathcal{S}} \mid z', \tilde z_{\mathcal{S}}, x_{\mathcal{S}})$ on $x_{\mathcal{S}}$ allows the distribution to some extent compensate for the inaccuracies (or loss of information) in $q_\phi(z', \tilde z_{\mathcal{S}} \mid x_{\mathcal{S}})$.
> >
> > Thanks for pointing this out. Indeed, this independence assumption $q_{\phi}(\tilde z_\mathcal{S} \mid z', \tilde z_{\setminus \mathcal{S}}, x_{\mathcal{S}} ) =q_\phi(\tilde z_\mathcal{S} \mid z', x_\mathcal{S})$ is an inductive bias that generally decreases the variational objective as it imposes a restriction on the encoding distribution that only approximates the posterior where this independence assumption holds.
> > However, this independence assumption allows us to respect the modality-specific nature of the private latent variables during encoding. In particular, this assumption allows us to encode the observations $x_{\mathcal{S}}$ via
> > $$q_{\phi}(z',\tilde z_\mathcal{S}, \tilde z_{\setminus \mathcal{S}} |x_\mathcal{S})=q_{\phi}(z'|x_\mathcal{S})p_{\theta}(\tilde z_{ \setminus \mathcal{S}}|z') q_{\phi}(\tilde z_{\mathcal{S}}|z', x_{\mathcal{S}})$$
> > so that the modality-specific information of $x_{\mathcal{S}}$ as encoded via $\tilde Z_{\mathcal{S}}$ is not impacted by the realisation $\tilde Z_{\setminus \mathcal{S}}$ of modality-specific variation from the other modalities.
> >
> >
> >
> > >As explained in the paragraph preceding Examples 3-5 (after equation 11), the distribution $p_\theta(\tilde z_{\setminus \mathcal{S}} \mid z', \tilde z_{\mathcal{S}})$ is often analytically tractable. So why do Examples 3-5 not use it? Specifically, Example 3 instead uses $\prod_{s \in \mathcal{M}\setminus\mathcal{S}} p_{\theta}(\tilde z_s)$ and Examples 4-5 instead use $\prod_{s \in \mathcal{M}\setminus\mathcal{S}} p_{\theta}(\tilde z_s \mid z')$. Is this a typo?
> >
> > Indeed, the distribution $p_\theta(\tilde z_{\setminus \mathcal{S}} \mid z', \tilde z_{\mathcal{S}})$ is often analytically tractable. However, we generally assumed that the prior over the private latent variables factorizes across the modalities (conditional on the shared latent variables) to account for the fact that, a priori, the private latent variable for each modality captures modality-specific information independent from the modality-specific information from the other modalities. The prior thus factorizes as in Examples 4-5. The factorization in Example 3 for the PoE case additionally assumes that a priori, all private latent variables are also independent of the shared latent variable, following previous works. This can be relaxed as long as the appropriate (conditional) Gaussian assumptions necessary for the PoE aggregation are satisfied. In our experiments, we assumed that all latent variables factorize a priori for all the considered aggregation schemes.
> >
> >
> > >The authors provided permutation-equivariant versions of PoE, SumPooling, and Self-Attention, but did not discuss permutation-equivariant versions of MoE or MoPoE. Is there a reason why this may not be possible with these aggregation schemes, are they incompatible with permutation-equivariance? Would be good to discuss this.
> >
> > It should principally be possible to use MoE or MoPoE for the encoding of private latent variables. However, we were not aware of previous work that uses MoE/MoPoE for private latent variables with a similar variational bound and have consequently not considered this. Recent work by Palumbo et al. (2023) uses a MoE model but, at the same time, considers a different variational bound that involves additional trainable prior distributions. We provide a brief comparison to this approach in Remark 24.

---

> > > ### Author Response · Authors · 2024-07-22
> > >
> > > >The evaluations use quantities such as "Full reconstruction", "Full rates", "Cross reconstruction", and "Cross rates", but the effect of these metrics going up/down is not clearly motivated or explained in the discussion of the results. Thus, it is unclear how important they are for multimodal VAEs and their contribution. Some interpretation of these metrics and their importance to multimodal VAEs would be helpful. [...] In Section 5.3, it seems that using the proposed objective improves cross-generation of the SVHN modality (Figure 2) but decreases cross-reconstruction (Figure 3). Why is that?
> > >
> > >
> > > Thank you for pointing out that the motivation and explanation for the rate/distortion measures has not been discussed clearly in the evaluations. We have now provided more details on these measures when we introduced the multi-modal variational objectives in Section 2. In particular, we now highlight that the mixture-based bound directly maximizes the cross reconstruction (or called cross prediction in the revised version), i.e., how well the missing modalities can be predicted as measured by the log-likelihood. This cross-prediction term does not appear in our objective, so the empirical result that these cross-prediction terms are higher (see e.g., Table 1, Figure 3, 6, 7) for the mixture-based bound across data sets and aggregation schemes and beta values seems reasonable.
> > >
> > > Note that the cross-prediction term cannot measure whether the generative model allows for the modeling of modality-specific variations, since, by construction, these should not be predictable based on other modalities. By optimizing for self-reconstruction of a modality subset and cross-prediction in a single objective, the mixture-based bound leads to "an inexact, average prediction; however, it cannot reliably predict modality-specific information" (Daunhawer et al. 2022).
> > > It is thus possible that the cross-predictions for the SVHN modalities in the mixture-based bounds are high, but predictions or generated samples are inexact, average prediction and as such do not look like representative samples. Note also that the cross prediction values in Figure 3 are averaged over all three modalities.
> > >
> > > The log-likelihood should be higher if a generative model is able to capture modality-specific information for models trained with $\beta =1$. For arbitrary $\beta$, we can take a rate-distortion perspective and look at how different models self-reconstruct all modalities (i.e. a full reconstruction term) relative to the KL-divergence of the encoding distribution of all modalities to the prior (a full rate term). This corresponds to a rate-distortion analysis of a VAE that merges all modalities into a single modality. A high full-reconstruction term is thus indicative of the encoder and decoder being able to reconstruct all modalities precisely, so that they do not produce an average prediction. Our variational objective achieves higher full-reconstruction terms (see e.g. Table 1, Figure 3, 6, 7) across data sets and aggregation schemes and beta values. Note that neither our objective, nor the mixture-based bound optimize for the full-reconstruction term directly.
> > >
> > >
> > >
> > >
> > > >In the MNIST-SVHN-Text experiments, the total number of modalities is 3. You could thus parametrise all possible permutations of modalities (8 of them in total) using a separate encoder (one per permutation). This would serve as a strong baseline to verify how good the permutation-invariant parametrisations really are.
> > >
> > >
> > > This is a valid suggestion and we are looking into this.

---

> > > > ### Author Response · Authors · 2024-07-22
> > > >
> > > > >Remark 4 presents some hypotheses that are not experimentally validated or reflected upon in the rest of the paper, so it is unclear how relevant it is.
> > > >
> > > > We agree that many of the points presented in Remark 4 have not been experimentally validated. However, we believe they may be of interest to some readers as they illustrate that similar to ELBO decompositions for uni-modal VAEs, such analyses (i) hint at shortcomings of multi-modal VAEs in an analogous way when comparing uni-modal VAEs to other uni-modal generative models and (ii) suggest that approaches developed for uni-modal VAEs that aim to address these may be of interest also for multi-modal VAEs in future work. We have added some comments at the beginning of Section 2.1 that we hope do better illustrate the relevance of the subsequent analyses and remarks.
> > > >
> > > >
> > > >
> > > > >Remarks 9, 10, 11, and 13 present some results/discussion that seem out-of-context. Why are they relevant to the paper?
> > > >
> > > > We believe that Remark 9 is relevant because it illustrates that the mixture-based bound has a different information-theoretic representation compared to our objective. In particular, while our objective bounds both a mutual information and conditional mutual information term that relate the observed modalities to two latent variables, the mixture-based bound maximizes a difference of two mutual information terms, see eq. (2) relative to a single latent variable. The mutual information terms in both objectives are bounded from below and above by different rate and distortion measures that we evaluate in the experimental section.
> > > >
> > > > Remark 10 is relevant because it shows that under idealized conditions, the optimal encoding distribution for our objective is the posterior distribution, possibly adjusted for some annealing when $\beta \neq 1$. In contrast, the optimal variational distribution for the mixture-based bound is, in general, different from the posterior distribution and involves some tilting to points that yield high cross-prediction on the training data. Among other things, this is relevant when one aims to learn identifiable models as the standard identifiability analyses concern the generative parameters such as those of the posterior distribution.
> > > >
> > > > Remark 11 illustrates the (well-known) property of self-attention, which allows for learning interactions between the unimodal encodings while pooling them, unlike a sum-pooling approach. We are happy to move it to the appendix if requested.
> > > >
> > > > Remark 12 shows that different aggregation approaches such as MoE or PoE with tempered densities are optimal aggregation schemes when one aims to minimize a weighted (forward or reverse) KL-divergence between the multi-modal encoding distributions and the uni-modal encoding distributions. However, the objective functions for learning multi-modal VAEs are usually not of this form, so that learning the aggregation function can be beneficial.
> > > >
> > > > >In Table 2, the numbers in parentheses are not explained. Are they standard deviations/errors, how many independent runs?
> > > >
> > > > Yes, the values in parentheses are standard deviations over five independent runs.
> > > >
> > > > >The tables in the evaluations were a little hard to decipher, this is because "ours" is used to denote the proposed bound, but the proposed aggregation schemes (SumPooling/SelfAttention) weren't highlighted. Denoting the proposed bound as "our bound" or "proposed bound" and appending the proposed scheme names with "(proposed)" may help.
> > > >
> > > > This is a good suggestion that we tried to adapt to some degree now. Instead of 'ours', we now write 'proposed objective' (also to avoid the term bound). Since the SumPooling/SelfAttention permutation-invariant schemes are new, to the best of our knowledge (at least for multi-modal VAEs), we so far preferred not to append 'proposed' to save space in the tables.

---

> > ### Comment · Reviewer_S2Lb · 2024-07-22
> > **Follow-up**
> >
> > > This is an important point and our revision now provides a more detailed discussion in Remark 3 on the role of $q\_{\phi}(z \mid x\_{\mathcal{S}})$ that features both bounds. First, we maximize the conditional objective $\mathcal{L}\_{\setminus \mathcal{S}}$ with respect to both $\theta$ and $\phi$. Second, it is indeed the case that the variational parameters affect both objectives via $q\_{\phi}(z \mid x\_{\mathcal{S}})$ and the optimal $q\_{\phi}(z \mid x\_{\mathcal{S}})$ is generally attained at different values for both objectives. Our conditional objective may be slightly extended to the setting where one uses a different model for the conditional prior instead of $q\_{\phi}(z \mid x\_{\mathcal{S}})$ arising in $\mathcal{L}\_{\setminus \mathcal{S}}$ to avoid this coupling effect. While this choice may lead to higher values for the objectives, we did not explore this further and only considered our more constrained approach that encourages learning encoding models that perform well both as conditional priors and encoders. Actually, the same $q\_{\phi}(z \mid x\_{\mathcal{s}})$ optimizes both objectives at the same time only under idealised conditions, see Remark 3.
> >
> > Thank you, authors, for updating the description in Remark 3, explaining the extension where $q\_{\phi}(z \mid x\_{S})$ in the conditional objective $\mathcal{L}\_{\setminus \mathcal{S}}$ is instead replaced with a different model, such that the $q\_{\phi}(z \mid x\_{S})$ in $\mathcal{L}\_{\mathcal{S}}$ and the $q\_{\phi}(z \mid x\_{S})$ in $\mathcal{L}\_{\setminus \mathcal{S}}$ are not the same. (If I understood your response correctly.)
> >
> > But, my question was about the two terms in equation $\texttt{Z}\_{\text{conditional}}$ that arise _only_ from the conditional term $\mathcal{L}\_{\setminus \mathcal{S}}$. To clarify my question, you show in $\texttt{Z}\_{\text{conditional}}$ that:
> > $$
> > \int p\_d(x\_{\setminus \mathcal{S}} \mid x\_{\mathcal{S}}) \mathcal{L}\_{\setminus \mathcal{S}}(x, \theta, \phi) \text{d} x\_{\setminus \mathcal{S}} + \mathcal{H}(p\_d(x\_{\setminus \mathcal{S}} \mid x\_{\mathcal{S}})) = - \text{KL}(q\_{\phi, \setminus \mathcal{S}}^{agg}(z \mid x\_{\mathcal{S}}) \mid\mid q\_{\phi}(z \mid x\_{\mathcal{S}})) - \int q\_{\phi, \setminus \mathcal{S}}^{agg}(z \mid x\_{\mathcal{S}})(\text{KL}(q^*(x\_{\setminus S} \mid z, x\_{\mathcal{S}}) \mid\mid p\_{\theta}(x\_{\setminus \mathcal{S}} \mid z))) \text{d} z,
> > $$
> > which you then use in Remark 3 to claim that "$\mathcal{L}\_{\setminus \mathcal{S}}$ is maximized when $q\_{\phi}(z \mid x\_{\mathcal{S}}) = \ldots = q\_{\phi, \setminus \mathcal{S}}^{agg}(z \mid x\_{\mathcal{S}})$", I'm assuming because it sets the first KL term on r.h.s., $\text{KL}(q\_{\phi, \setminus \mathcal{S}}^{agg}(z \mid x\_{\mathcal{S}}) \mid\mid q\_{\phi}(z \mid x\_{\mathcal{S}}))$, to 0. However, this ignores the second term, $\int q\_{\phi, \setminus \mathcal{S}}^{agg}(z \mid x\_{\mathcal{S}})(\text{KL}(q^*(x\_{\setminus S} \mid z, x\_{\mathcal{S}}) \mid\mid p\_{\theta}(x\_{\setminus \mathcal{S}} \mid z))) \text{d} z$, which also depends on the parameters $\phi$ via both $q\_{\phi, \setminus \mathcal{S}}^{agg}$ and $q^{\star}$. Hence, simply maximising the (expected) conditional lower-bound $\mathcal{L}\_{\setminus S}$ will not yield $q\_{\phi}(z \mid x\_{\mathcal{S}}) = q\_{\phi, \setminus \mathcal{S}}^{agg}(z \mid x\_{\mathcal{S}})$ as stated in Remark 3. You can check this by inserting $q\_{\phi, \setminus \mathcal{S}}^{agg}(z \mid x\_{\mathcal{S}}) = q\_{\phi}(z \mid x\_{\mathcal{S}})$ into the second term in $\texttt{Z}\_{\text{conditional}}$ and see that it will not generally be 0. Then, maximising the second term in $\texttt{Z}\_{\text{conditional}}$ will likely change $q\_{\phi}(z \mid x\_{\mathcal{S}})$ such that it is $q\_{\phi}(z \mid x\_{\mathcal{S}}) \not =q\_{\phi, \setminus \mathcal{S}}^{agg}(z \mid x\_{\mathcal{S}})$, thus increasing the KL in the first term. Perhaps some additional assumptions are missing that would make your statement in Remark 3 true, or am I misunderstanding something (in which case a clarification would be useful)?

---

> > > ### Author Response · Authors · 2024-07-22
> > >
> > > Thanks for pointing out that the optimality of $q_\phi(z | x_\mathcal{S})$ in $L_{\setminus \mathcal{S}}$ is not clear and requires additional assumptions. Instead of looking at $Z_{\text{conditional}}$, we can re-write (similar to [1] in the uni-modal case), the conditional expectation of $L_{\setminus \mathcal{S}}$ for any fixed $x_\mathcal{S}$ as
> > > $$ \int p_d(x_{\setminus \mathcal{S} }|x_{\mathcal{S}})  L_{\setminus \mathcal{S}}(x,\theta,\phi, 1) \text{d} x_{\setminus \mathcal{S}} =\int p_d(x_{\setminus \mathcal{S}}|x_{\mathcal{S}})  q_{\phi}(z|x)  \log p_{\theta}(x_{\setminus \mathcal{S}}|z)\text{d} z \text{d} x_{\setminus \mathcal{S}} + \int p_d(x) \mathcal{H} (q_\phi(z|x)) \text{d} x_{\setminus \mathcal{S}} + \int q_{\phi, \setminus \mathcal{S}}^{\text{agg}} (z|x_\mathcal{S})  \log q_\phi(z|x_\mathcal{S}) \text{d} z.$$
> > > Whenever $q_\phi(z | x_\mathcal{S})$ can be learned independently from $q_\phi(z |x)$, the above is maximized for
> > > $q_\phi(z |x_\mathcal{S})=q_{\phi, \setminus \mathcal{S}}^{agg}(z |x_{\mathcal{S}})$. This additional assumption is generally not satisfied for the scalable aggregation models considered here, but would hold if one parameterises the encoder for each modality subset independently.
> > >
> > > [1] Tomczak, Jakub, and Max Welling. "VAE with a VampPrior." International Conference on Artificial Intelligence and Statistics. PMLR, 2018.

---

### Review · Reviewer_y99S · 2024-07-04

**Summary Of Contributions:**

The paper derived a new and tighter variational lower bound for multi-modal VAEs (MMVAEs). It also proposed new and more flexible multi-modal aggregation schemes based on permutation-invariant NNs. A connection to identifiable VAEs is given, by seeing modality as an auxiliary variable. Both theoretical and experimental analyses are presented.

**Audience:**

Yes

**Claims And Evidence:**

Yes

**Requested Changes:**

Disclaimer: While I know well about several types of VAEs, I am unfamiliar with MMVAEs. And, as indicated above, due to the density and broadness of the paper, I failed to check many details in the paper. Thus, many of the following could be seen as suggestions to make the paper more accessible.

As indicated in the Weaknesses, *the authors could move to Appendix some of the Remarks and Examples which are noncritical for understanding the main contributions, and the Introduction could be expanded*. And I hope the following will be useful in the latter regard.

The conditional independence involved in the generative model of MMVAEs is satisfied when the latent variable Z captures all the unobserved factors shared by the modalities. I think it would be beneficial to mention this intuition for newcomers.

I am not sure why in the first place the previous work would want to consider the bound (1) rather than your (3)? I mean, your (3), which bounds each modality, is much more intuitive to me than (1). Is the scalability problem you mentioned at the beginning of p4 the main reason? Anyway, it would be helpful to touch this question earlier.

I am not sure if it is convincing enough to introduce permutation-invariance simply as a generalization of MoE/PoE. On the one hand, I am not sure if it is still a generalization in multi-modal cases, which is what we really consider. On the other hand, I think permutation-invariance and MoE/PoE contain very different information about the underlying mechanism, i.e., how the modalities work together to generate the observed data.

Could you expand on the reason (a) on p14 why your bound is beneficial for learning an identifiable VAE? A related note is that it would be useful to make it clear that identifiability is a preliminary to estimation, thus your bound or aggregation schemes do not aid the identifiability, rather, given the model which is already identifiable, the aggregation schemes give more flexible parametrization, and the bound gives a better approximation. I know the authors understand this but I think it would be better to spell this out for readers.

I believe it is unnecessary to introduce $\rho$ as a pdf, because we have finite modalities. It is simply a pmf, and this is much easier for readers.

It seems that the $Z_S$ in (2) is not only “to emphasize that Z is conditional on $X_S$”. For example, if we just see $Z_S$ as another symbol of Z, then $H(X|Z_S)$ in (2) cannot be reduced back to the 1st term of (1).

### Minor
At least in one place, you wrote “multi-modal” as “multi-model”.

**Strengths And Weaknesses:**

### Strengths

The paper is solidly written and covers various related work and ideas.

The theoretical derivations are serious.

The connection to identifiable VAEs is interesting.

### Weaknesses

The paper is very dense and could use a gentler introduction.

Many of the Remarks and Examples are related work and ideas, which, while interesting, could be distractive.

---

> ### Author Response · Authors · 2024-07-22
>
> We thank the reviewer for taking the time to review our manuscript and for the helpful comments and feedback. We very much appreciate the reviewer's positive reception for the solidly written submission covering various related ideas with serious theoretical derivations.
> We agree with the reviewer that the paper is somewhat dense and that the connection between some remarks to the main ideas of our submission could be explained better. We tried to be clearer in our revised submission with a more detailed introduction.
>
>
> >I am not sure why in the first place the previous work would want to consider the bound (1) rather than your (3)? I mean, your (3), which bounds each modality, is much more intuitive to me than (1). Is the scalability problem you mentioned at the beginning of p4 the main reason? Anyway, it would be helpful to touch this question earlier.
>
>
> We are happy to hear that our objective (3) appears more intuitive than the mixture-based bound (1). Our initial introduction did not provide a detailed motivation for the different variational objectives and we have now significantly expanded on this when we introduce our objective in Section 2. In particular, the variational objective (3) can be seen as treating all $\mathcal{S}$ modalities as a single modality, thereby learning a latent representation for a joint compression and reconstruction of all modalities in $\mathcal{S}$. However, it does not explicitly include a cross-modal prediction term, i.e. how well the latent variable $Z_{\mathcal{S}}$ encoding $x_{\mathcal{S}}$ is able to predict the modalities not in $\mathcal{S}$. The difference between (3) and (1) is exactly this cross-modal prediction term. Instead of having a single objective with a single latent variable $Z_{\mathcal{S}}$ that should both self-reconstruct and cross-predict modalities as in (1), we consider a second variational objective with another latent variable encoding all observed modalities. This second objective can be seen as an approximation of the conditional log-likelihood and gives rise to upper and lower variational bounds on the conditional mutual information. Both objectives can be scalable as long as the used aggregation functions are scalable.
>
>
>
> >The conditional independence involved in the generative model of MMVAEs is satisfied when the latent variable Z captures all the unobserved factors shared by the modalities. I think it would be beneficial to mention this intuition for newcomers.
>
>
> It is a good suggestion to provide some intuition on the assumed structure of the generative model. We have added now that this conditional independence assumption means that the latent variable $Z$ captures all unobserved factors shared by the modalities.
>
>
> >I am not sure if it is convincing enough to introduce permutation-invariance simply as a generalization of MoE/PoE. On the one hand, I am not sure if it is still a generalization in multi-modal cases, which is what we really consider. On the other hand, I think permutation-invariance and MoE/PoE contain very different information about the underlying mechanism, i.e., how the modalities work together to generate the observed data.
>
>
> We completely agree that MoE/PoE contain very different assumptions about how different modalities should be encoded together so that they generate or reconstruct the observed data. Indeed, MoE/PoE are inductive biases that lead to different behaviours, e.g., in the POE case, each expert has veto power and has a stronger influence if more competent in terms of greater precision. While such biases are optimal for certain objectives, see Remark 13, these schemes may not be flexible enough to approximate the optimal encoding distributions in the variational objectives. Although the aggregation mechanisms of MoE/PoE are of a permutation-invariant form (after uni-modal encodings), these aggregation mechanisms can only be approximated by the used permutation-invariant architectures. For example, the PoE aggregation involves taking inverses that can only be approximated by the learned aggregation models. The considered permutation-invariant models are thus at most a generalization of MoE/PoE under universal approximation assumptions. We have added this point to the submission.

---

> > ### Author Response · Authors · 2024-07-22
> >
> > >Could you expand on the reason (a) on p14 why your bound is beneficial for learning an identifiable VAE? A related note is that it would be useful to make it clear that identifiability is a preliminary to estimation, thus your bound or aggregation schemes do not aid the identifiability, rather, given the model which is already identifiable, the aggregation schemes give more flexible parametrization, and the bound gives a better approximation. I know the authors understand this but I think it would be better to spell this out for readers.
> >
> > This is a good point to spell out more clearly to the reader in the revised version. Indeed, identifiability considered here concerns parameters of the multi-modal posterior distribution and the conditional generative distribution. It is thus preliminary to estimation and only concerns the generative model and not the inference approach. However, both the multi-modal posterior distribution and the conditional generative distribution are intractable. In practice, we thus replace them with approximations. We believe that our inference approach is beneficial for this type of identifiability when making these variational approximations because (a) unlike some other variational objectives, the posterior is the optimal variational distribution with $L_{\setminus \mathcal{S}}(x)$ being an approximation of a lower bound on $\log p_{\theta}(x_{\setminus \mathcal{S}}|x_\mathcal{S})$, see Remark 10, and (b) the trainable aggregation schemes can be more flexible for approximating the optimal encoding distribution.
> >
> >
> > >It seems that the $Z_S$ in (2) is not only “to emphasize that Z is conditional on $X_S$”. For example, if we just see $Z_S$ as another symbol of Z, then $H(X|Z_S)$ in (2) cannot be reduced back to the 1st term of (1).
> >
> >
> > We are thankful to the reviewer for pointing out issues with our initial equation (2). The mutual information terms are actually only upper or lower variational bounds thereof. We have adjusted our discussion and added more details in Remark 9 and Appendix C.

---

### Review · Reviewer_tUXM · 2024-07-11

**Summary Of Contributions:**

This paper proposes a tighter bound a new aggregation scheme for VAE-based models that are infused with Mixture of Experts (MoE) or Product of Experts (PoE). The paper starts by proposing a new variational abound very similar to previous work by  Daunhawer et al., 2022 and Shi et al., 2019. From my understanding, it involves dividing the modalities two 2 subsets $X_S$ and $X_{/S}$ and optimizing the sum of two different ELBO objectives to target $p_d(X_S)$ and $p_d(X_{/S}|X_S)$ respectively (weighted by the prior $p(S)$). The second objective is different than the standard ELBO objective as it uses a the inference distribution $q(z|X_{S})$ rather than $q(z|X_{\S})$. Intuitively, this makes sense given the latent variable for some modes should contain information to roughly generate the $x$ for other modalities. The authors then provide provide various interpretations of the given bound. The paper then proposes two different variational inference distribution to use: MoE and PoE. The authors additionally propose to use the encoder architecture that is permutation invariant (e.g. sum-pooling or transformer-based encoder) which makes sense given that $x$ modalities are permutation invariant. The proposed model is weakly identifiable given some conditions. Finally the authors do a series of experiments where the they compare the propose bound to the mixture-based bound proposed by  Daunhawer et al., 2022 and Shi et al., 2019. The results do indeed look promising.

**Audience:**

Yes

**Broader Impact Concerns:**

This is mainly a theoretical paper and to the best of my understanding raises no major social impact.

**Claims And Evidence:**

Yes

**Requested Changes:**

- Sorry if I missed this but it seems like the paper assumes you have access to the ground truth conditional distribution $p(x_{/S}|x_S)$. Is this is a reasonable data assumption in multi-modal settings?

**Strengths And Weaknesses:**

**Strengths**

- *Clarity*: The paper reads well overall. I really appreciated the Remarks[4-6] providing different perspectives on the objective function. The paper is also very thorough and well-motivated. Furthermore, I think the authors do relatively a good job of explaining the limitations of their approach.

- *Soundness*: The new objective makes sense (Though I do think the paper can be improved by providing more details and intuitions for why the bound is tighter than the mixture-based). The fact that the objective $L(X_{\S})$ ends up being an approximate lower bound on $\log p(X_{\S}|X_S)$ is not super obvious mathematically at first so it's quite interesting! The use of permutation invariant architecture and distributions also makes complete sense.

- *Experiments*: The authors did a nice set of experiments to compare the objectives performance against the aother bounds. The datasets are ofcourse very simple but they are enough I think to show the potential of the new bound. There could be more highlight and discussion on the individual effect of the new bound against the choice of $q$ and the use of sum-pooling encoder. Figure 1 is also quite interesting!


Overall, I think the TMLR would find the paper interesting and I'm more than happy to recommend acceptance.

**Weaknesses**

- *Novelty*: I am not very familiar with multimodal learning literature so my apologies if this is not the case, but I am very surprised to see that the permutation invariant architecture and distributions are not been used previously before in multi-modal settings, especially given that I know these type of distributions and architectures have been used lot in Amortized VI settings in a variety of applications such as meta-learning etc. Therefore, The use of PE and MoE and the sum-pooling encoders by themselves I do not find very novel. The new bound however I think is a very valuable contribution and can be highlighted more in the paper.

---

> ### Author Response · Authors · 2024-07-22
>
> We thank the reviewer for taking the time to review our manuscript and for the helpful comments and feedback. We are happy to hear that the reviewer found the paper thorough, well motivated (with different perspectives in Remarks [4-6]), and with a promising set of small-scale experiments. We tried to address the reviewer's comments, particularly by expanding the motivation and intuition for the two considered variational objectives.
>
> >The new objective makes sense (Though I do think the paper can be improved by providing more details and intuitions for why the bound is tighter than the mixture-based).
>
> We agree that it would be helpful to provide more details and intuitions as to why the suggested variational objective avoids the gap relative to the log-likelihood that occurs in the mixture-based bound. We have now provided more details on this in the introduction and at the beginning of Section 2. Illustratively, in the mixture-based bound, one tries to reconstruct/predict all modalities from incomplete information using only the modalities $\mathcal{S}$, which leads to learning an inexact, average prediction. In particular, it cannot reliably predict modality-specific information that is not shared with other modality subsets, as measured by the conditional entropies  $\mathcal(p_d(X_{\setminus \mathcal{S}}|X_\mathcal{S}))$. These conditional entropy terms yield a gap that cannot be reduced even for flexible encoding distributions. In contrast, we introduce an additional variational objective with a second latent variable $Z_{\mathcal{M}}$ that encodes all observed modalities $X = X_{\mathcal{M}}$. Unlike the latent variable $Z_{\mathcal{S}}$ in the mixture-based bound that can only encode incomplete information using the modalities in $\mathcal{S}$, this second latent variable $Z_{\mathcal{M}}$ can encode modality-specific information from all observed modalities, thereby avoiding the averaging prediction in the mixture-based bound.
>
>
> >There could be more highlight and discussion on the individual effect of the new bound against the choice of $q$ and the use of sum-pooling encoder.
>
> This is a good point. We agree that disentangling the effect of the variational objective with the choice of aggregation is an important question. Overall, we find that for a given choice of $q$, our objective achieves a higher log-likelihood across the different experiments. Likewise, fixing the variational objective, we observe that the sum-pooling or self-attention encoder achieves higher multi-modal log-likelihoods compared to MoE or PoE models. While the PoE encoder has a better log-likelihood for MNIST, the joint log-likelihood was lower. Our interpretation is that while a PoE model may lead to a good generative model and reconstruction of a single modality, there is a strong inductive bias (e.g., each expert has veto power), which impacts the encoder for multiple modalities.
>
>
> >I am very surprised to see that the permutation invariant architecture and distributions are not been used previously before in multi-modal settings, especially given that I know these type of distributions and architectures have been used lot in Amortized VI settings in a variety of applications such as meta-learning etc. Therefore, The use of PE and MoE and the sum-pooling encoders by themselves I do not find very novel.
>
> We completely agree that permutation-invariant architectures and models are not novel per se and have been used in different amortized VI settings, such as meta-learning. Indeed, the permutation-invariant sum-pooling encoder is basically an encoder used in neural processes, while the self-attention encoder is used in attentive neural processes without an encoding of query-specific representations. Generally, we believe that different architectures from various Amortized VI settings may also be leveraged for the multi-modal VAE setup.
>
>
>
> >It seems like the paper assumes you have access to the ground truth conditional distribution $p(x_{\setminus \mathcal{S}}|x_{\mathcal{S}})$. Is this is a reasonable data assumption in multi-modal settings?
>
>
> We assume we have access to samples from the true joint multi-modal distribution, and thus samples from the true conditional distribution $p(x_{\setminus \mathcal{S}}|x_{\mathcal{S}})$. This is a common assumption used for multi-modal VAEs but also for other multi-modal generative models. However, this can be relaxed to missing modalities as long as the missingness is completely at random, see Section 4.3 and Section 5.2/Table 4 for experiments with non-linear identifiable models of 5 modalities in a partially observed setup with a missingness rate of 50 percent. Not all previous approaches for learning multi-modal VAEs can account for missing modalities.

---

> > ### Comment · Reviewer_tUXM · 2024-07-23
> > **Respond to Authors**
> >
> > Thank you for your response.
> >
> > - *Intuition and explanation behind the new bound*: Thank you for your explanation. I think adding this explanation was necessary  Reviewer S2Lb also pointed out.
> >
> > - And thanks for clarifying the assumption of having access  to $p(x_{/S}| x_S )$. I think the neat way that the proposed model handles the missing modalities is a good point! I would also still encourage to add some additional explanation regrading disentangling the bound vs choice of $q$, and possible adding a toy experiment where the bound is used with a simple dummy $q$ and the old bound is used with a choice of MoE/PoE $q$.
> >
> > - I must say I agree with some of the points raised by reviewer S2Lb. Mainly that the authors could do a better job of at explaining some of the metrics in the experiment Section. Assuming that the clarifications the authors provided made it to the revised version, as well as the fact that main key metric here remains the loglikelihood and sample quality, I think this is a minor point. Another good point raised by reviewer S2Lb was to permute over everything in the MNIST-SVHN-Text experiment to test how well these functions approximate the ideal permutation invariant case.

---

> > > ### Author Response · Authors · 2024-07-25
> > >
> > > Thank you for your feedback to our response! We have now considered to also use a variational density $q$ as suggested by reviewer S2Lb that trains a different encoding model for each modality subset to test the flexibility of the introduced permutation-invariant models for some of the experiments. In particular, we train different MLP heads for each modality subset, while using the same modality-specific initial encoding architectures across all the considered aggregation schemes. Our initial results suggest that our objective again achieves higher log-likelihood values compared to the mixture-based bound for these $q$, and learning different heads for each modality subset does not improve on the Sum-Pooling/Self-Attention aggregation, while being more computationally expensive to train. Regarding an "additional toy experiment where the bound is used with a simple dummy $q$ and the old bound is used with a choice of MoE/PoE $q$", we would like to clarify that our experiments (such as Table 3, first two rows with the columns under mixture bound) often contain results for the old bound with a choice of MoE/PoE and contrast it with different choices of $q$ in combination with our objective.

---

### Decision · Action_Editor_iMan · 2024-08-21

**Recommendation:** Accept as is

**Comment:**

Overall I see this as solid technical work in a research area that is relevant to the TMLR community.  There are some weaknesses in terms of clarity, but these are not sufficiently serious to warrant rejection of the paper.  The reviewers were unanimous in recommending the paper be accepted, and I see no reason to overwrite their decision.

As all the reviewers were already satisfied with the updates made during the rebuttals/discussions, I do not see the paper as having any specific issues that need to be corrected through minor revisions.  I, therefore, recommend that the paper is accepted "as is".  That said, I think their is scope to improve the clarity of the paper further and encourage the authors to work on this for the final camera-ready version.  To this end, I include some comments made by Reviewer SL2b below from their final recommendations which are not currently visible to the authors, but may be helpful in making such improvements.

> I still believe that the clarity of the paper could be better if the authors moved some discussion points that are less relevant to the main narrative into the appendix, further improved the arguments about the proposed bound, explained the evaluation results more clearly and their relevance to multi-modal VAEs, and analysed the properties of the new objective in more detail.

**Audience:**

The paper makes meaningful novel contributions in the area of multi-modal variational auto-encoders.  It is clearly in scope for TMLR and will be of interest to others working in the same area.  All reviewers and myself thus agree that it meets TMLR's audience criteria.

**Claims And Evidence:**

Though some concerns with the correctness of the work were raised in the original review of Reviewer S2Lb, these have been addressed in the response and paper update.  As such, all reviewers are in clear agreement that the paper meets TMLR's evaluation criteria on claims and evidence.  The only slight doubt remaining is in the clarity of the paper itself, with two reviewers expressing some lingering concerns about this in their final recommendation.  However, neither felt these issues were sufficiently severe to prohibit acceptance, and I concur with this view.